# Gravity waves in the winter stratosphere over the Southern Ocean: high-resolution satellite observations and 3-D spectral analysis

Neil P. Hindley[1], Corwin J. Wright[1], Nathan D. Smith[2], Lars Hoffmann[3], Laura A. Holt[4], M. Joan Alexander[4], Tracy Moffat-Griffin[5], and Nicholas J. Mitchell[1]

[1]Centre for Space, Atmospheric and Oceanic Science, University of Bath, Bath, UK
[2]Independent Researcher, Bath, UK
[3]Jülich Supercomputing Centre, Forschungszentrum Jülich, Jülich, Germany
[4]Northwest Research Associates, Boulder, Colorado, USA
[5]Atmosphere, Ice and Climate Group, British Antarctic Survey, Cambridge, UK

**Correspondence:** Neil Hindley (n.hindley@bath.ac.uk)

**Abstract.** Atmospheric gravity waves play a key role in the transfer of energy and momentum between layers of the Earth's atmosphere. However, nearly all Global Circulation Models (GCMs) seriously under-represent the momentum fluxes of gravity waves at latitudes near 60°S, which can lead to significant biases. A prominent example of this is the "cold-pole" problem, where modelled winter stratospheres are unrealistically cold. There is thus a need for large-scale measurements of gravity-wave fluxes near 60°S, and indeed globally, to test and constrain GCMs. Such measurements are notoriously difficult, because they require 3-D observations of wave properties if the fluxes are to be estimated without using significant limiting assumptions. Here we use 3-D satellite measurements of stratospheric gravity waves from NASA's AIRS/Aqua instrument. We present the first extended application of a 3-D Stockwell transform (3DST) method to determine localised gravity-wave amplitudes, wavelengths and directions of propagation around the entire region of the Southern Ocean near 60°S during austral winter 2010. We first validate our method using a synthetic wave field and two case studies of real gravity waves over the Southern Andes and the island of South Georgia. A new technique to overcome wave amplitude attenuation problems in previous methods is also presented. We then characterise large-scale gravity-wave occurrence frequencies, directional momentum fluxes and short-timescale intermittency over the entire Southern Ocean. Our results show that highest wave-occurrence frequencies, amplitudes and momentum fluxes are observed in the stratosphere over the mountains of the Southern Andes and Antarctic Peninsula. However, we find that around 60-80% of total zonal-mean momentum flux is located over the open Southern Ocean during June-August, where a large "belt" of increased wave-occurrence frequencies, amplitudes and fluxes is observed. Our results also suggest significant short-timescale variability of fluxes from both orographic and non-orographic sources in the region. A particularly striking result is a widespread convergence of gravity-wave momentum fluxes towards latitudes around 60°S from the north and south. We propose that this convergence, which is observed at nearly all longitudes during winter, could account for a significant part of the under-represented flux in GCMs at these latitudes.

# 1 Introduction

Gravity waves (GWs) are a key component in the dynamics of the Earth's atmosphere. Through the transportation and deposition of energy and momentum, these waves act as the primary coupling mechanism between atmospheric layers (e.g. Fritts and Alexander, 2003, and citations therein).

Despite their importance, the accurate representation of gravity waves in General Circulation Models (GCMs) used for numerical weather prediction and climate modelling has proved challenging. For the majority of operational GCMs, large portions of the gravity-wave spectrum are sub-gridscale phenomena and must be parameterised. Such parameterisations are also needed to accurately simulate gravity-wave generation and dissipation mechanisms. However, these parameterisations are poorly constrained by global observations of gravity-wave characteristics (Alexander et al., 2010). As a result, uncertainties in their scales, intensity, distribution and short-timescale variability remain large.

One example of a significant and long-standing bias that is common to nearly all GCMs occurs in the winter and spring-time Antarctic stratosphere. There, the modelled southern stratospheric polar vortex consistently breaks up too late in spring compared to observations. Simulated winds are around $10\,\mathrm{ms^{-1}}$ too strong, polar temperatures around $5-10\,\mathrm{K}$ too low, and the break-up of the polar vortex occurs some 2-3 weeks later than observations show. These biases collectively form the well-known "cold-pole problem" in GCMs (e.g. Butchart et al., 2011; McLandress et al., 2012).

The cold-pole problem has significant implications. In particular, it undermines the ability of GCMs to make accurate multi-year predictions of winds in the southern hemisphere. These are essential for climate-change projections as they provide the basis for dynamic transport of trace gases in most Chemistry Climate Models (CCMs) (e.g. McLandress et al., 2010, 2011). For example, inaccurate projections of future change in winds over the Southern Ocean, which is the major region of additional heat and carbon uptake in the global ocean (Froelicher et al., 2015), can result in unrealistically low predictions of the Antarctic ozone, which is a principal driver of recent Antarctic climate change (Garcia et al., 2017). Consequently, these biases have been identified as a serious impediment to progress in understanding the dynamics of the stratosphere and to developing GCMs.

During austral winter, observations have revealed the southern hemisphere stratosphere to be home to some of the most intense gravity wave activity on Earth (see Hindley et al., 2015, and citations therein). It has been hypothesised that biases like the cold-pole problem arise due to an under-representation of the momentum fluxes of these waves in models at latitudes around $60°$S (Butchart et al., 2011; McLandress et al., 2012; Geller et al., 2013; Garfinkel and Oman, 2018). Determining the true sources of these waves that are "missing" in models, such that they can be accurately simulated or parameterised, is thus an important step to resolving the "cold-pole problem" and other biases in GCMs.

Satellite observations are a useful way of obtaining the necessary gravity-wave measurements to do this. Their global coverage and all-weather capability allows them to detect and measure waves in regions inaccessible to ground-based techniques, such as over the open ocean. However, each satellite remote sensing instrument or technique is sensitive to only a portion of the gravity-wave spectrum, referred to as its "observational filter" (e.g. Preusse et al., 2002; Alexander et al., 2010; Alexander and Barnet, 2007, and citations therein). Due to satellite geometry and spectral weighting functions, limb-sounding satellite instruments may typically have good vertical resolutions of a few kilometres, but relatively poor horizontal resolutions

($\lambda_H \lesssim 400\,\text{km}$) for gravity waves. Alternatively, nadir-sounding instruments may have good horizontal resolutions of a few tens of kilometres, but relatively poor vertical resolutions ($\lambda_Z \gtrsim 15\,\text{km}$).

Further, satellite observations of atmospheric gravity waves are usually limited to 1-dimensional (1-D) or 2-dimensional (2-D) measurements. These are usually either a series of vertical profiles from limb-sounding instruments (e.g. Alexander et al., 2008; Ern et al., 2011; Hindley et al., 2015) or continuous cross-track scans from nadir-sounding imagers (e.g. Alexander and Barnet, 2007; Hoffmann et al., 2014). In some cases, instruments can be combined to infer 3-dimensional (3-D) wave properties (e.g. Faber et al., 2013; Alexander, 2015; Wright et al., 2016), but these are limited to regions of measurement overlap or high sampling density.

This is a major limitation, since spatially-localised measurements of the full 3-D wavevector and wave-packet amplitude are required to accurately determine the gravity-wave momentum fluxes needed to constrain gravity-wave parametrisations in GCMs (Alexander et al., 2010). Several methods for estimating momentum fluxes have been applied over the last two decades. These methods include using adjacent vertical profiles from limb-sounding observations to infer upper-bound horizontal wavelengths (e.g. Ern et al., 2004; Alexander et al., 2008; Faber et al., 2013; Alexander, 2015), or using a priori information to infer vertical wavelengths for mountain waves in nadir-sounding observations (e.g. Alexander et al., 2009), but global measurements of gravity-wave momentum fluxes from single-instrument measurements of the full 3-D wavevector have only become possible recently (e.g. Ern et al., 2017; Wright et al., 2017).

Here, we apply a 3-D spectral analysis technique to 3-D satellite observations of stratospheric gravity waves. In Sect. 2 we describe how gravity-wave perturbations are extracted from 3-D AIRS satellite observations and discuss the noise and resolution limits of our data. In Sect. 3 we describe our 3-D Stockwell transform method for obtaining measurements of 3-D gravity-wave properties and validate it using a synthetic wave field. A new approach that mitigates unwanted wave-amplitude attenuation is presented. In Sect. 4 we apply our method to real gravity wave measurements in two case studies: one over the Southern Andes and another over the isolated mountainous island of South Georgia in the Southern Ocean. In Sect. 5, our analysis is then extended to satellite observations of stratospheric gravity waves over the entire Southern Ocean around $60°\text{S}$ during austral winter 2010 to measure wave amplitudes, wavelengths, occurrence frequencies, directional momentum fluxes and short-timescale intermittency. Our key results are discussed in Sect. 6, and we present our conclusions in 7.

## 2   3-D AIRS temperature measurements

The Atmospheric Infrared Sounder (AIRS, Aumann et al., 2003) is a nadir-sounding multi-spectral imager on board NASA's Aqua satellite. Launched in 2002, Aqua is part of the "A-Train" satellite constellation, orbiting at a height of around $700\,\text{km}$ in a sun-synchronous near-polar orbit. The AIRS instrument images the atmosphere in 2378 spectral channels in a 90-pixel wide ($\sim 1800\,\text{km}$) cross-track swath between $\pm 49°$ from the nadir. This continuous swath is archived into 240 "granules" per day, each of which is 135 pixels in the ($\sim 2400\,\text{km}$) along-track direction, corresponding to roughly six minutes of observations per granule.

Hoffmann and Alexander (2009) developed a dedicated high-resolution temperature retrieval for AIRS to support gravity-wave studies. This retrieval is three-dimensional and capable of resolving wave features in both the vertical and the horizontal with superior resolution over the operational retrieval scheme. While no single technique can yet measure the full gravity-wave spectrum (e.g. Alexander and Barnet, 2007), this 3-D dataset presents an opportunity to make global measurements of gravity-wave properties in 3-D. The observational filter (e.g. Preusse et al., 2002; Alexander et al., 2010) of the 3-D AIRS retrieval is sensitive gravity waves with relatively long vertical wavelengths ($\lambda_Z \gtrsim 10-15\,\mathrm{km}$) and relatively short horizontal wavelengths ($\lambda_H \lesssim 500 - 1000\,\mathrm{km}$). This spectral portion of gravity waves are associated with high momentum fluxes via the relation in Eqn. 8, derived in Ern et al. (2004).

In the retrieval of Hoffmann and Alexander (2009), infrared radiances from the 4.3 $\mu$m and 15 $\mu$m channels are, at every pixel in the AIRS granule, used to retrieve a vertical profile of stratospheric temperature from the surface to an altitude of $z = 90\,\mathrm{km}$ in steps of 3 km below $z = 60\,\mathrm{km}$ and 5 km above. Retrieved temperatures are most reliable between $z = 20 - 60\,\mathrm{km}$ where the retrieval noise is lowest (Hoffmann and Alexander, 2009), although noise increases considerably above around $z = 55\,\mathrm{km}$ and below $z = 25\,\mathrm{km}$. The key feature of this retrieval is that wave features in the vertical are also directly resolved, with a vertical resolution that can be as low as $\sim 7\,\mathrm{km}$. An additional advantage is that this retrieval retains the full horizontal resolution of AIRS and does not combine blocks of $3 \times 3$ footprints for a cloud-clearing procedure as used in the operational AIRS retrieval. Further information on the high-resolution AIRS retrieval can be found in Hoffmann and Alexander (2009), Meyer and Hoffmann (2014), Sato et al. (2016), Ern et al. (2017) and Meyer et al. (2018).

The nighttime retrieval has advantages over the daytime retrieval since the effects of non-local thermodynamic equilibrium can be neglected, meaning both the 4.3 $\mu$m and 15 $\mu$m channels can be used. This can result in improved vertical resolution and/or reduced error due to noise. Ern et al. (2017) used only descending node (i.e. local nighttime) measurements in their analysis for this reason. In our study however, we find that noise levels and resolution in the daytime retrieval (shown by the dashed lines in Fig. 2a), which only utilises the 15 $\mu$m channels, is still quite reasonable. We find that realistic 3-D wave structures are clearly visible in daytime data in numerous examples (although not shown here), although perhaps not so clearly as might have been seen in nighttime data. Hence in the subsequent analyses presented here, we choose to include data from both the daytime and nighttime retrievals. We apply a relatively high noise (or confidence) threshold (see Sect. 2.2) appropriate for both daytime and nighttime observations. Since we consider monthly timescales in this study, we find that the increased measurement coverage outweighs potential errors introduced through the inclusion of daytime data. Further, since we focus on mid to high latitudes during winter in this study, there are likely to be far more observations made during nighttime conditions than during daytime.

## 2.1  AIRS observations of stratospheric gravity waves

To separate gravity-wave temperature perturbations from signals of planetary waves and large-scale temperature gradients, the raw 3-D AIRS temperature fields are detrended using a 4[th]-order cross-track polynomial (e.g. Wu, 2004; Alexander and Barnet, 2007; Hoffmann et al., 2016; Wright et al., 2017). The shortest observable wavelength (Nyquist-limited) is around 30 to 40 km,

due to the horizontal spacing which varies between 14 to 20 km, depending on scan-angle from the nadir. Since we regrid our data in the next step, these values are later standardised to be independent of scan angle.

Once each cross-track row has been detrended, the residual gravity-wave perturbations are regridded on to a regular distance grid in $x$, $y$ and $z$ (cross-track, along-track and vertical directions respectively) using linear interpolation. This regular grid is needed for accurate spectral analysis. The retrieval of Hoffmann and Alexander (2009) produces temperature measurements on the same horizontal grid as standard AIRS data ($135 \times 90$ pixel granules, 240 granules per day), but at 17 vertical levels between $z = 0 - 90$ km, with a spacing of 3 km below $z = 60$ km. Our regular horizontal grid is chosen to be of the same size, but regularly-spaced so as not to invalidate our wavelength measurements in later spectral analysis. The horizontal spacing of our regular grid is 18.2 km and 19.6 km in the along-track and cross-track directions respectively. Our chosen vertical regular distance grid is over the range $z = 10 - 70$ km in steps of 3 km, close to the original grid of Hoffmann and Alexander (2009), but centred around $z = 40$ km. To focus on perturbations between $z \approx 20 - 60$ km, where retrieval noise is lowest (Hoffmann and Alexander, 2009), each vertical column of the measurement volume is multiplied by a Tukey window that is equal to 1 within these altitudes with a smooth half-bell taper towards zero below 20 km and above 60 km. This step means we are less likely to be sensitive to vertical wavelengths greater than 40 km. To reduce the effect of horizontal pixel-scale noise, we also apply a horizontal $3 \times 3$-pixel boxcar filter to all height levels.

Figure 1 shows three examples of AIRS stratospheric gravity-wave temperature perturbations in the southern hemisphere during June-July 2010. In each panel, detrended temperature perturbations are shown along the AIRS scan-track at an altitude of $z = 40$ km for around (a) 0530 UTC on 1st June, (b) 0330 UTC on 5th July and (c) 1940-2120 UTC on 28th June 2010. The respective granule numbers are (a) $052 - 055$, (b) $033 - 036$ and (c) $197 - 200$ and $213 - 216$. Panel (c) shows two consecutive overpasses on the same day, separated in time by around 95 minutes.

Large gravity-wave temperature perturbations exceeding $\pm 6$ K are observed over the Southern Andes in Fig. 1a. Phase fronts are aligned broadly north-south, following the extent of the mountain ranges at the southern tip of South America. Due to their shape and location, it is reasonable to assume that these waves are very likely to be orographic mountain waves that result from flow over topography near the surface. Assuming that these waves propagate into the prevailing wind, the slight inclination of the phase fronts would suggest a significant additional southward component of wave momentum flux. Gravity-wave temperature perturbations around $\pm 4$ K are also observed over the Antarctic Peninsula, where a slight northward inclination is apparent. This region has been identified as an intense "hot spot" of gravity-wave activity (Hoffmann et al., 2013; Hindley et al., 2015; Hoffmann et al., 2016) and has been the subject of numerous studies in recent years (e.g. Alexander and Teitelbaum, 2007, 2011; de la Torre et al., 2014; Wright et al., 2017).

Figure 1b shows gravity-wave temperature perturbations over the isolated mountainous island of South Georgia (54°S, 36°W) in the Southern Ocean, located around 2000 km east of the Antarctic Peninsula. Near to the island, temperature perturbations exceed $\pm 5$ K with phase fronts arranged in a series of chevron-like patterns, indicative of mountain waves from an isolated point source (e.g. Vosper, 2015). The temperature perturbations in Figs. 1(a-b) are shown at one vertical layer of our 3-D volume of AIRS measurements. The regions outlined by the black and white dashed lines around the Southern Andes, Antarctic Peninsula and South Georgia in panels (a) and (b) are analysed in detail as case studies in Sects. 4.1 and 4.2.

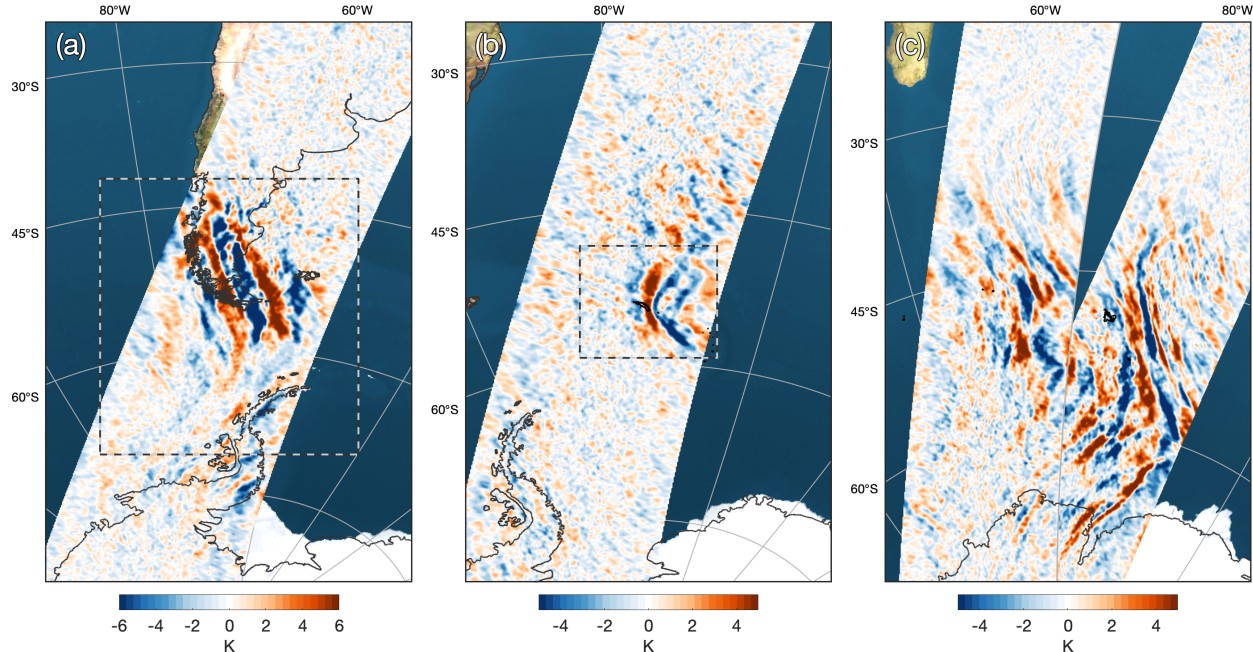

**Figure 1.** Temperature perturbation measurements along the AIRS scan-track at an altitude of $z = 40\,$km during overpasses on (a) 1$^{st}$ June over South America and Antarctic Peninsula, (b) 5$^{th}$ July over the island of South Georgia and (c) 28$^{th}$ June 2010 over the Southern Ocean. Panel (c) shows two consecutive overpasses on the same day. This is one vertical layer in a 3-D volume of AIRS temperature measurements between $z = 20 - 60\,$km using the retrieval of Hoffmann and Alexander (2009). Dashed lines in (a) and (b) indicate the regions considered in the 3-D case studies in Figures 4 and 5.

Figure 1c shows measurements from two consecutive AIRS overpasses over the Southern Ocean. The Kerguelen (49°S, 69°W) Islands and Heard Islands (-53°S, 73.5°W) are located near the centre of the image, with the island of Madagascar visible in the top left and the Antarctic coastline at the bottom. Very large-scale gravity-wave perturbations up to around ±4 K are visible from around 75-35°S, with distinct regions of southward and northward propagation (assuming westward propagation into the prevailing westerly winds) visible to the north and south of around 55-60°S respectively. There appears to be some consistency of phase fronts between successive overpasses where they overlap, suggesting that these waves have periods significantly longer than the orbital period of the Aqua satellite ($\sim$ 95 minutes), likely several hours.

The source of the waves in panel (c) is hard to define, but it seems likely that they have a non-orographic origin. Their size and spatial pattern make the relatively small islands of Kerguelen and Heard unlikely to be orographic sources in this case. One probable source mechanism is that these are waves excited as a result of spontaneous adjustment processes around the edge of the southern stratospheric polar vortex (e.g. O'Sullivan and Dunkerton, 1995; Sato and Yoshiki, 2008). The scale of these waves, and the fact that their vertical extent is large enough to be visible to AIRS, is quite striking. However, we find that events such as these are not atypical. In this study, dozens of wave events very similar to these were observed visually in

AIRS measurements at nearly all longitudes over the Southern Ocean during June-August 2010. Interestingly, events such as these seemed to be observed more frequently over the southern Atlantic and southern Indian Ocean sections than the southern Pacific.

## 2.2 Determining retrieval noise and observational filter for 3-D AIRS gravity-wave measurements

Virtually all satellite measurements are subject to errors that may arise from retrieval noise. In the case of AIRS temperature retrievals, measurement noise in observed radiance spectra is mapped into retrieval noise in retrieved temperatures. Sources of noise may thus include thermal noise within the instrument itself, effects of non-local thermal equilibrium due to sunlight during daytime, or significant deviations in true atmospheric conditions from a priori and smoothing constraints applied in the retrieval process.

Hoffmann and Alexander (2009) estimated the error due to noise in the 3-D AIRS temperature retrieval to be around $1.4 - 2.1$ K for midlatitude atmospheric conditions between altitudes of $z = 20 - 60$ km. To examine the effects of different atmospheric conditions at different latitudes and seasons on retrieval error noise, the noise analysis of Hoffmann and Alexander (2009) was repeated for daytime and nighttime retrievals in the tropics, at midlatitudes and during polar winter and summer. This noise analysis follows standard optimal estimation retrieval theory (Rodgers, 2000), where the measurement covariance

matrix that characterises the noise of the AIRS radiance observations is mapped into temperature errors by means of the gain matrix calculated in the retrieval process (see Hoffmann and Alexander, 2009, their Sect. 4.4). Noise estimates for the individual AIRS channels used in the error analysis have been taken from version 6 of the AIRS channel property files available from the AIRS instrument team at NASA.

Figure 2a shows estimated retrieval noise in 3-D AIRS temperatures against altitude. Noise is generally lowest between

$z = 25 - 50$ km for all latitudes and seasons, where most error values lie in the range 1.2 to 1.7 K. Retrievals performed during polar winter at altitudes around $z = 40$ km have slightly lower noise error values of around 0.8 K and 1.2 K for the daytime and nighttime retrievals respectively. Here, we focus on the months of June to August, where around 70 to 75% of measurements poleward of $30°$S took place under nighttime conditions, so a value closer towards 1.2 K is probably more realistic for this study.

In practice we choose a slightly higher noise threshold of 1.5 K for measured wave amplitudes around $z = 40$ km in this study. We apply this threshold to results of our large-scale spectral analysis of 3-D AIRS gravity-wave temperature perturbations in Sect. 5. This value was chosen so as to increase confidence in our measured gravity wave properties as distinct from the impact of noise. A summary of further steps we have taken to mitigate the impact of retrieval noise in our gravity-wave measurements provided in Appendix B.

### 2.2.1 AIRS spectral resolution and observational filter

AIRS is a nadir-sounding instrument, and derived radiance measurements are made with deep vertical weighting functions which affect the vertical resolution of retrieved measurements. Figure 2b shows the estimated vertical resolution of 3-D AIRS temperatures against altitude for different atmospheric conditions, again using the approach of Hoffmann and Alexander

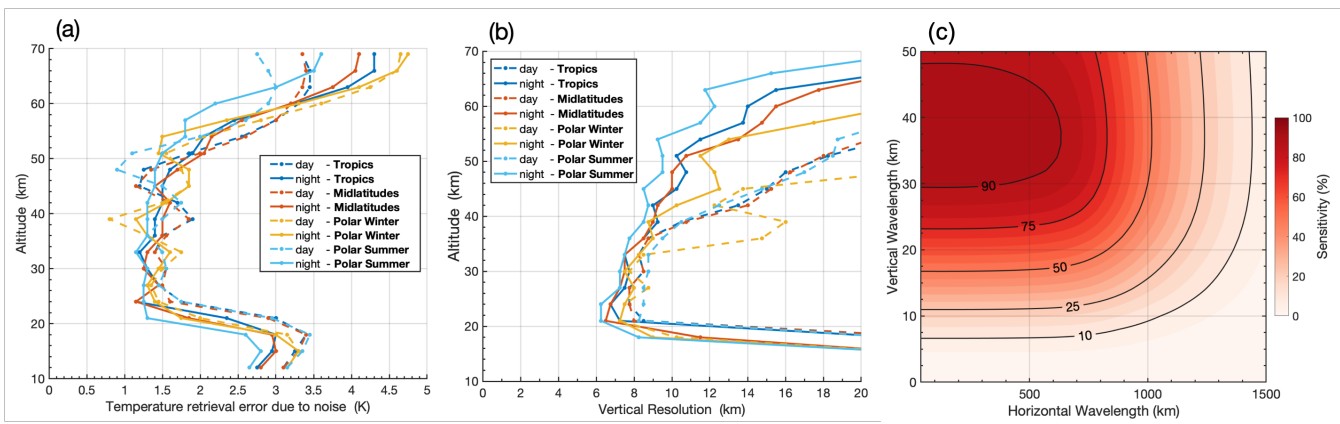

**Figure 2.** Estimated temperature retrieval errors (a) and vertical resolutions (b) for the 3-D AIRS temperature measurements against altitude for different atmospheric conditions, based on latitude and season. Dashed and solid lines indicate the daytime and nighttime retrievals respectively. Panel (c) shows estimated sensitivity of the detrended 3-D AIRS temperature perturbations to waves with different vertical and horizontal (across-track) wavelengths for observations made under average atmospheric conditions at an altitude of $z = 40\,\mathrm{km}$. For details, see text in Sect. 2.2.

(2009). In the altitude range $z = 20 - 50\,\mathrm{km}$, the vertical resolution of nighttime retrieval varies between around 6 and 13 km, whereas resolution of the daytime retrieval varies between around 9 and 17 km. Generally, the nighttime retrieval (solid lines) has improved vertical resolution over the daytime retrieval (dashed lines). This is due to the inclusion of measurements from both the $4.3\,\mu\mathrm{m}$ and $15\,\mu\mathrm{m}$ channels in the retrieval (Hoffmann and Alexander, 2009). The largest improvement between day

and night is seen during polar winter at altitudes between around $z = 34 - 42\,\mathrm{km}$. As mentioned above, around 70 to 75% of measurements used in our study took place during nighttime conditions, so our results here are less affected by the poorer daytime resolution.

In this study, we focus our investigation at altitudes around $z = 40\,\mathrm{km}$. This is a convenient height, since it lies in the centre of the usable height range and represents the point at which the greatest range of vertical wavelengths can be measured using

spectral analysis methods due to edge truncation and "cone of influence" effects (e.g. Woods and Smith, 2010).

The average vertical resolution of both the daytime and nighttime retrievals globally under all atmospheric conditions at $z = 40\,\mathrm{km}$ is $\sim 10\,\mathrm{km}$. Using this value, and the horizontal across-track resolution of the $4^{\mathrm{th}}$-order polynomial detrending as shown in Fig. 5 of Hoffmann et al. (2014) we are able to estimate the approximate sensitivity of our detrended AIRS temperature perturbation measurements to waves with different spectral characteristics, as shown in Fig. 2c. This sensitivity map can be

considered as an approximate observational filter for our gravity-wave retrieval. To find the values for the vertical in Fig. 2c, synthetic sinusoidal waves of various wavelengths, centred at $z = 40\,\mathrm{km}$, were created between altitudes of $z = 10 - 70\,\mathrm{km}$. These waves were localised to the usable altitude window of $20 - 60\,\mathrm{km}$ using same the Tukey window approach as described in Sect. 2.1, then convolved with a Gaussian of full-width-at-half-maximum (FWHM) of 10 km to simulate the AIRS retrieval at $z = 40\,\mathrm{km}$. The reduction in the amplitude of these synthetic waves was used to measure sensitivity. This approach is not

perfect, as different atmospheric conditions and the change in vertical resolution with height will further impact our sensitivity, but it serves as a first-order representation of the observational filter of our 3-D AIRS measurements.

Sensitivity is close to 100% for waves with wavelengths between 35 to 45 km in the vertical and less than around 500 km in the horizontal, and around 50% for wavelengths greater than 17 km in the vertical and less than 1000 km in the horizontal. The majority of our measured wavelengths in the results of this study fall within this 50% sensitivity region (as we would expect), so it is likely that measured wave amplitudes in this study may be a factor of 2 or more higher in reality than they appear in AIRS measurements. We do not apply any correction factor for this here however, as such a factor would place enormous weight on the accuracy of the determination of such a correction factor, any error in which could lead to large and confusing discrepancies in momentum flux values. Our results are thus presented 'as measured', subject to the observational filter of the AIRS instrument.

The sensitivity assessment results here are broadly in line with the sensitivity assessments of Hoffmann and Alexander (2009, their Fig. 6) and Ern et al. (2017, their supplementary Fig. S3a), except that here we include the impact of the limited observational window of $z = 20 - 60$ km, which reduces sensitivity to waves with vertical wavelengths longer than around 45 km, as can be seen in the top left of Fig. 2c.

## 3    Measuring 3-D gravity-wave properties with the Stockwell transform

In order to estimate gravity-wave momentum fluxes from AIRS temperature perturbations, spectral analysis tools are required. For temperature perturbation measurements, these tools must provide spatially-localised measurements of wave-packet amplitudes and full 3-D wavevectors in order for momentum fluxes to be estimated (Ern et al., 2004).

Fourier analysis can reveal what frequencies (or wavenumbers) are present in a given dataset, but it cannot tell us when (or where) those frequencies occurred. In the atmosphere, gravity waves are transient phenomena. In order to accurately measure their characteristics, it is essential to be able to localise the time or place at which different wave packets are observed, in order to accurately localise and measure their spectral characteristics individually.

Ern et al. (2017) applied the "S3D" method of Lehmann et al. (2012) to do this, which involves least-squares sine-fitting within small-scale cubes to localise wave features. The S3D method is effective, but it is not without its limitations. Large cubes will provide better spectral resolution, but poorer spatial localisation and vice versa. Smaller cubes also make the method rather more computationally intensive. Further, since the sine-fitting method assumes wave homogeneity over the entire cube, this will become increasingly unlikely for large cubes.

### 3.1    The $N$-dimensional Stockwell transform

Here we present an alternative solution: a 3-D extension of the Stockwell transform (hereafter S-transform, Stockwell et al., 1996; Stockwell, 1999). The S-transform is a widely-used spectral analysis technique for the measurement and localisation of frequencies (or wavenumbers) in a timeseries (or distance profile) and their corresponding amplitudes. This capability makes the S-transform well suited to gravity-wave analysis from a variety of geophysical datasets (e.g. Fritts et al., 1998; Stockwell

and Lowe, 2001; Alexander and Barnet, 2007; Alexander et al., 2008; Stockwell et al., 2011; McDonald, 2012; Wright and Gille, 2013; Alexander, 2015; Sato et al., 2016; Hindley et al., 2016; Wright et al., 2017; Hu et al., 2019a). The S-transform has also been used in a variety of other fields, such as the planetary (Wright, 2012), engineering (Kuyuk, 2015) and biomedical sciences (e.g. Goodyear et al., 2004; Brown et al., 2010; Yan et al., 2015). Since it can make use of fast (discrete) Fourier Transform (DFT) algorithms, the S-transform (including higher-dimensional versions) can be relatively fast to compute. This makes it an attractive choice for large-scale data analysis in the geosciences, where many tens or hundreds of thousands of transforms may be needed.

It is usually easier to describe higher-dimensional S-transforms analytically when generalised for $N$-dimensions. For an $N$-dimensional function $h(\boldsymbol{x})$, where $\boldsymbol{x} = (x_1, x_2, \cdots x_N)$ is a column vector describing an $N$-dimensional coordinate system, the generalised form of the $N$-dimensional S-transform (NDST) $S(\boldsymbol{\tau}, \boldsymbol{f})$, using the definition of Stockwell et al. (1996), can be written as

$$
S(\boldsymbol{\tau}, \boldsymbol{f}) = \int_{-\infty}^{\infty} h(\boldsymbol{x})\, w(\boldsymbol{x} - \boldsymbol{\tau}\,;\boldsymbol{f})\, e^{-i2\pi \boldsymbol{f} \cdot \boldsymbol{x}}\, \mathrm{d}\boldsymbol{x}. \tag{1}
$$

where $\boldsymbol{\tau} = (\tau_1, \tau_2, \cdots, \tau_N)$ and $\boldsymbol{f} = (f_1, f_2, \cdots, f_n)$ are column vectors denoting translation and spatial frequency (inverse of wavelength) in the $x_1, x_2, \ldots x_N$ directions, and $\boldsymbol{f} \cdot \boldsymbol{x}$ denotes the scalar product[1]. Here, angular wavenumbers (more commonly used in the atmospheric sciences) are related to spatial frequency shown here as $k_n = 2\pi f_n$. The function $w(\boldsymbol{x} - \boldsymbol{\tau}\,;\boldsymbol{f})$ is a Gaussian apodizing function, referred to as the "voice Gaussian" (Stockwell et al., 1996), and is given by

$$
w(\boldsymbol{x} - \boldsymbol{\tau}, \boldsymbol{f}) = \frac{1}{(2\pi)^{N/2}} \prod_{n=1}^{N} \frac{|f_n|}{c_n}\, e^{\frac{-(x_n - \tau_n)^2 f_n^2}{2c_n^2}} \tag{2}
$$

where $c_n$ is a positive scalar scaling parameter in each dimension $n$ that can be used to tune the spectral-spatial localisation capabilities of the NDST for each dimension independently (Fritts et al., 1998; Pinnegar and Mansinha, 2003; Hindley et al., 2016). Here, for the 3DST, we find that setting $(c_x, c_y, c_z) = (0.25, 0.25, 0.25)$ provides a good compromise between spatial and spectral localisation. Further discussion of the effects of altering $c_n$ can be found in Hindley et al. (2016).

A two-dimensional S-transform (2DST) was applied in Stockwell et al. (2011) to analyse a mesospheric bore wave over Antarctica in airglow observations. Later, Hindley et al. (2016) developed a new application of the 2DST to measure stratospheric gravity-wave properties in 2-D AIRS brightness temperature measurements. The authors of Hindley et al. then extended their 2DST method to 3-D, and an early version of the software was used to apply the 3DST to 3-D AIRS temperature measurements in a regional gravity-wave study over the southern Andes and Antarctic Peninsula in Wright et al. (2017). Recently, Hu et al. (2019a) and Hu et al. (2019b) used the 2DST to investigate mesospheric gravity-waves and solitary waves in airglow measurements from the day/night band of the Visible Infrared Imaging Radiometer Suite (VIIRS) instrument. Other studies

---

[1]Note that a phase shift of $e^{i2\pi \boldsymbol{\tau} \cdot \boldsymbol{f}}$ arises between our application of S-transform later in Sect. 3.2 and the Stockwell et al. (1996) definition given here in Eqn. 1. This phase shift arises from our approach of deriving the analytic signal to compute the S-transform. It does not affect our results here, but is worth noting for future applications. In simple terms, the analytical approach of Stockwell et al. (1996) in the spectral domain centres each Gaussian window around zero and shifts the spectrum $H(\alpha)$, whereas our method keeps $H(\alpha)$ centred around zero and shifts each Gaussian window.

currently in progress include the application of the 2DST to the study of atmospheric waves around blocking systems, and an application of the 3DST to investigations of wave structures over small islands and hurricanes. The S-transform application described here uses a new generalised software suite that currently supports the computation of 1-, 2-, 3- and 4-dimensional S-transforms, following the same algorithmic approach for each, which is described in Sect. 3.2.

5 ## 3.2 Computing the $N$-dimensional S-transform

In our application, the computation of the S-transform given in Eqn. 1 is performed in several steps. It is usually more computationally efficient to compute the S-transform using Fourier-domain multiplication operations rather than spatial-domain convolutions. To do this we must find $W(\boldsymbol{\alpha} - \boldsymbol{f}, \boldsymbol{f})$, which is the appropriately-shifted Fourier analogue of the voice Gaussian in Eqn. 2, given as

$$W(\boldsymbol{\alpha} - \boldsymbol{f}, \boldsymbol{f}) = \prod_{n=1}^{N} e^{-2\pi^2 c_n^2 \frac{(\alpha_n - f_n)^2}{f_n^2}} \tag{3}$$

where $\boldsymbol{\alpha} = (\alpha_1, \alpha_2, \cdots, \alpha_N)$ denotes translation in the Fourier domain.

For any real $N$-dimensional input data $h(\boldsymbol{x})$ that might contain wave-like perturbations, such as 3-D AIRS temperature perturbations or a synthetic wave field, the following steps are performed:

1. The $N$-dimensional Fourier transform of the input data $H(\boldsymbol{\alpha}) = \text{FFT}\{h(\boldsymbol{x})\}$ is computed.

2. The analytic signal $H(\boldsymbol{\alpha}) \to H_a(\boldsymbol{\alpha})$ is determined (Hilbert transform). This means that, for all coefficients of $H(\boldsymbol{\alpha})$ in a complex conjugate pair, one of the pair is doubled and the other is set to zero. All coefficients not in a complex conjugate pair are left unchanged.

3. $H_a(\boldsymbol{\alpha})$ is multiplied by the Fourier-domain "voice Gaussian" $W_n(\alpha_n - f_n, f_n)$ in Eqn. 3 for a specific frequency $f_n$.

4. The inverse Fourier transform $\text{FFT}^{-1}\{H_a(\boldsymbol{\alpha}) \times W_n(\alpha_n - f_n, f_n)\}$ is taken and the result is inserted into the respective

   rows and columns of $S(\boldsymbol{\tau}, \boldsymbol{f})$ to give the S-transform result for the specified frequency $f_n$.

5. Steps (3) and (4) are repeated for every frequency considered in the S-transform.

For 3-D input data, the resulting S-transform $S(\boldsymbol{\tau}, \boldsymbol{f})$ is a 6-D complex-valued object. In our application, translation in the spatial domain $\boldsymbol{\tau}$ is analogous to the input spatial coordinate system $\boldsymbol{x}$, such that

$$S(\boldsymbol{\tau}, \boldsymbol{f}) = S(\boldsymbol{x}, \boldsymbol{f}) \tag{4}$$

$$\equiv S(x, y, z, f_x, f_y, f_z)$$

To obtain a more manageable output, we follow the method of Hindley et al. (2016) to "collapse" the 6-D S-transform object $S(\boldsymbol{x}, \boldsymbol{f})$ down to 3-D objects that contain the dominant wave amplitudes and spectral characteristics at each location in $\boldsymbol{x}$. This is achieved by finding the coefficients of $|S(\boldsymbol{x}, \boldsymbol{f})|$ with the largest absolute spectral amplitudes in $\boldsymbol{f}$ for each location

in $x$. The value of each complex coefficient denotes the dominant amplitude and phase at each location in $x$, which is stored in the complex 3-D object $\mathcal{A}(x)$. The locations of the coefficients of $\mathcal{A}(x)$ within $S(x, f)$ thus correspond to the dominant corresponding spatial frequencies for each location in $x$. This gives us three more 3-D objects $\mathcal{F}_x(x)$, $\mathcal{F}_y(x)$, $\mathcal{F}_z(x)$ that contain these dominant spatial frequencies. The result is four 3-D objects $\mathcal{A}(x, y, z)$, $\mathcal{F}_x(x, y, z)$, $\mathcal{F}_y(x, y, z)$, $\mathcal{F}_z(x, y, z)$ that

are the same size as the input data $h(x)$. These objects contain the dominant measured amplitudes, phases and frequencies at each location $x, y, z$ in the input data.

## 3.3 Improved frequency selection in the S-transform

The S-transform is normally computed for a pre-specified range of frequencies. As can be seen from the steps in Sect. 3.2 however, the computational cost of our S-transform application is almost directly proportional to the number of frequencies

considered.

For 1- and 2-D S-transforms, this cost is not generally significant for modern computing systems. However for the 3DST, in particular for large-scale climatological studies of 3-D AIRS measurements, several tens of thousands of frequency combinations are typically applied for each granule. Many of these frequencies may not even be present in the data. This has a large computational cost, such that global-scale studies of multiple months or years becomes impractical.

Here we apply a new method to reduce the computational cost of the S-transform by considering only the dominant frequencies present in the input data. Steps (1) and (2) listed in Sect. 3.2 are performed as usual. Then, the coefficients of the Fourier transform $H(\alpha)$ are sorted into descending order with respect to their absolute spectral amplitude. Excluding the zeroth frequencies, for which the S-transform is not defined, the top $\eta$ frequencies with the largest absolute spectral coefficients are selected. For the 3DST applied here, we set $\eta = 1000$ such that the dominant 1000 frequencies are selected. We can also set

limits such that only the dominant $\eta$ frequencies within a particular frequency range can be considered. Steps (3), (4) and (5) are then performed as usual but only for these selected frequencies.

We find that for 3-D AIRS temperature perturbation measurements, setting $\eta = 1000$ reduces computation time by up to a factor of 10, while the resulting "collapsed" objects $\mathcal{A}(x, y, z)$, $\mathcal{F}_x(x, y, z)$, $\mathcal{F}_y(x, y, z)$, $\mathcal{F}_z(x, y, z)$ that contain the dominant S-transform amplitudes and frequencies are nearly identical to those computed using full frequency ranges. For more or less

complex wavefields, $\eta$ can be increased or decreased as required.

## 3.4 Improved wave-packet amplitude measurements in the S-transform

The S-transform is known to attenuate the amplitudes of highly-localised wave packets. For a wave packet with wavelength $\lambda$ and peak amplitude $a$, the peak S-transform-measured amplitude evaluated for $\lambda$ will always be less than $a$ due to the interaction of the wave packet's amplitude envelope and the "voice Gaussian" Eqn. 2. The more localised the wave packet (that

is, the fewer wave cycles), the larger the amplitude attenuation will be. For gravity-wave studies this can be a limitation, since gravity waves are very often observed in localised packets. In 1-D this attenuation effect is usually negligible (Wright, 2010), however for higher dimensions it can be quite significant.

Amplitude attenuation for higher dimensional S-transforms has been shown in Stockwell et al. (2011), Hindley et al. (2016), Wright et al. (2017) and others. Hindley et al. (2016) compensated somewhat for amplitude attenuation in the 2DST through the use of a scaling parameter and a new elliptic-Bessel window, but this approach still had limitations. A "boost factor" was applied in Wright et al. (2017) which also mitigated this effect to a reasonable degree, but its determination was somewhat arbitrary.

### 3.4.1 Estimating S-transform amplitude attenuation for wave packets

If the exact analytical form of the wave packet is known, the S-transform attenuation can be predicted exactly. To show this, we consider an $N$-dimensional wave packet $h(\boldsymbol{x})$, which consists of a cosinusoidal wave $\cos(\boldsymbol{k} \cdot \boldsymbol{x} + \theta)$ with angular wavenumber $\boldsymbol{k}$, amplitude $a$ and phase $\theta$, enclosed within a Gaussian amplitude envelope:

$$h(\boldsymbol{x}) = a\cos(\boldsymbol{k} \cdot \boldsymbol{x} + \theta) \prod_{n=1}^{N} e^{\frac{-(x_n - \tau_n)^2}{2s_n^2}} \tag{5}$$

Here, $\boldsymbol{k} = (k_1, k_2 \cdots k_N)$ and $\boldsymbol{s} = (s_1, s_2 \cdots s_N)$ are column vectors describing angular wavenumbers $k_n$ and standard deviations $s_n$ in an $N$-dimensional coordinate system $\boldsymbol{x} = (x_1, x_2, \cdots, x_N)$. If we compute the $N$-dimensional S-transform of this wave packet (that is, insert Eqn. 5 into Eqn. 1) and evaluate for wavelength $\boldsymbol{\lambda} = 2\pi / \boldsymbol{k}$, the peak measured amplitude $a_{\text{out}}$ will be given by

$$a_{\text{out}} = a \prod_{n=1}^{N} \left( \frac{s_n^2}{s_n^2 + c_n^2 \lambda_n^2} \right)^{\frac{1}{2}} \tag{6}$$

where $c_n$ is the scaling parameter for the S-transform described in Sect. 3.1. The full derivation of Eqn. 6 is included in Appendix A. We can see from Eqn. 6 that the full measured attenuation is the product of the terms for each dimension $n$, and it is phase invariant. If the standard deviation $s_n \gg \lambda_n$ (i.e. the wave packet is very large containing many wave cycles), the terms in the product approach unity, and the attenuation is virtually non-existent such that $a_{\text{out}} \to a$.

In reality, a typical gravity wave packet that we might observe in AIRS measurements might have wavelength $\lambda_n \approx s_n$, that is, perhaps one wave cycle per standard deviation of an approximately Gaussian amplitude envelope. In this case, the peak measured amplitude (taking $c_n = 1$) would be roughly 70%, 50% and 35% of the input value for the 1-, 2- and 3-dimensional S-transforms. By setting $c_n = 0.25$ for each dimension, we can improve this to 89%, 80% and 72% respectively, but this is at the expense of some spectral resolution. However, the inclination of a wave packet can mean that the wavelength in a particular direction can be very long (i.e. $\lambda_n \gg s_n$), so these values represent a best case (see Sect. 3.5).

Of course, we could simply recover the analytic signal of the input data to get the pointwise absolute amplitude (i.e. compute only steps (1), (2) and (4) in Sect. 3.2), but this amplitude would have no frequency dependence. As a result, the pointwise amplitude of individual regions could be contaminated by measurement noise, or from frequencies not considered in the S-transform.

### 3.4.2 A "composite" S-transform method for improved amplitude measurements

Here we present a new solution to this problem. To estimate the original localised amplitude of the dominant wave packets in the input data, we compute the S-transform again but for only a single $N$-dimensional "composite" Gaussian window.

This composite Gaussian window $W_{\text{comp}}(\boldsymbol{\alpha} - \boldsymbol{f}, \boldsymbol{f})$ is assembled as a combination of all the "voice Gaussians" in Eqn. 3 that were applied in the original S-transform. For the 3DST, for example, this is done by concatenating all the 3-dimensional "voice Gaussians" in Eqn. 3 into an 4-dimensional object, then taking the max{ } along the 4th dimension as

$$W_{\text{comp}}(\boldsymbol{\alpha} - \boldsymbol{f}, \boldsymbol{f}) = \max_{n=4} \{ w_1 \| w_2 \| \dots \| w_J \} \tag{7}$$

where $j = 1, 2, \dots J$ denotes every frequency (or frequency combination) considered in the S-transform, since each frequency will have a unique "voice Gaussian" as given by $w_j (\boldsymbol{\alpha} - \boldsymbol{f}_j, \boldsymbol{f}_j)$. The "composite" S-transform is then computed by performing steps (1) and (2) listed in Sect. 3.2 as usual. Then, steps (3) and (4) are performed only once using the composite Gaussian window shown in Eqn. 7.

The result is a complex-valued "composite" S-transform object $S_{\text{comp}}(\boldsymbol{x})$ that is the same size as the input data. The absolute magnitude of the coefficients of $S_{\text{comp}}(\boldsymbol{x})$ contain the estimated dominant wave amplitude at each location in the input data. The key aspect of these amplitude measurements is that they are only focused around frequencies that were considered in the S-transform. If only one frequency $f$ is considered, $S_{\text{comp}}(\boldsymbol{x})$ is equivalent to $S(\boldsymbol{x}, \boldsymbol{f})|_{\boldsymbol{f}=f}$. Likewise, if all available frequencies are considered, $W_{\text{comp}}(\boldsymbol{\alpha} - \boldsymbol{f}, \boldsymbol{f})$ is equal to unity everywhere and $S_{\text{comp}}(\boldsymbol{x})$ simply resembles the analytic signal.

In the next section, we show that this new amplitude estimation method dramatically reduces attenuation effects for localised wave packets in the S-transform.

### 3.5 Testing the 3DST analysis using a synthetic wave field

Before we apply our 3DST method to 3-D AIRS temperature perturbations, we first test the method on a synthetic wave field containing known wave parameters, such that we can assess its performance in measuring these parameters. Our 3DST method is designed to be as general as possible, such that it can be applied to study wave packets in any dataset, not just the 3-D AIRS measurements considered here.

To assemble our synthetic wave field, we first define a $200 \times 200 \times 200$ element domain in a regular coordinate system $x, y, z$. We specify the units of our domain to be kilometres with 1 km spacing, but for synthetic waves this is an arbitrary choice. For consistency with the 3-D AIRS measurements analysed later in the study, we define our synthetic waves to be equivalent to temperature perturbations around a uniform background state, although again this choice is arbitrary.

Within our spatial domain, we define eight synthetic wave packets. Each wave packet consists of a cosinusoidal wave centred at a given location in our synthetic domain. All input waves have a central (peak) wave amplitude equal to 2 K and are numbered 1 to 8. Each wave is localised by a Gaussian amplitude envelope centred at this location with standard deviations $s_x$, $s_y$ and $s_z$ in the $x, y$ and $z$ directions respectively. These envelopes are allowed to overlap. Each wave packet has a wavelength $\lambda$, azimuth $\theta$ (measured clockwise from the positive $y$-axis) and elevation angle $\phi$ (relative to the $x, y$ plane). All input wave parameters

| Wave | 1 | 2 | 3 | 4 | 5 | 6 | 7 | 8 |
|---|---|---|---|---|---|---|---|---|
| **Input wave parameters at central location** | | | | | | | | |
| $T'$ (K) | 2 | 2 | 2 | 2 | 2 | 2 | 2 | 2 |
| $\lambda$ (km) | 19 | 25 | 10 | 47 | 27 | 60 | 15 | 12 |
| $\theta$ (deg) | 66 | -16 | -30 | -104 | 135 | 45 | -80 | 144 |
| $\phi$ (deg) | 17 | 22 | 45 | 14 | 16 | 65 | 32 | 36 |
| Centred at | 0,0,0 | -50,-50,-75 | 25,25,25 | 60,60,60 | -50,30,20 | 50,-50,30 | -20,60,-75 | -30,-35,-35 |
| $s_x, s_y, s_z$ | 21,21,21 | 42,32,21 | 11,15,15 | 42,42,42 | 42,42,42 | 47,53,64 | 32,32,32 | 13,13,13 |
| **3DST-measured wave parameters at central location** | | | | | | | | |
| $T'_{\text{Standard}}$ (K) | 1.38 | 1.37 | 1.55 | 1.05 | 1.62 | 0.91 | 1.54 | 1.59 |
| $T'_{\text{New}}$ (K) | 1.95 | 1.86 | 1.91 | 1.56 | 1.98 | 1.46 | 1.90 | 1.98 |
| $\lambda$ (km) | 18 | 24 | 10 | 41 | 27 | 49 | 15 | 11 |
| $\theta$ (deg) | 61 | -16 | -35 | -117 | 135 | 45 | -75 | 143 |
| $\phi$ (deg) | 21 | 29 | 43 | 24 | 16 | 47 | 32 | 36 |

**Table 1.** Input and measured synthetic wave parameters for the test of our 3-D Stockwell transform (3DST) method on a synthetic wave field. The specified wave parameters are wave amplitude $T'$ (maximum temperature perturbation), wavelength $\lambda$, azimuth $\theta$, elevation angle $\phi$, central location in our specified coordinate system (Centred at:) and the standard deviations of the Gaussian envelope for each wave packet $s_x$, $s_y$ and $s_z$ in the $x$, $y$ and $z$ directions. $T'_{\text{Standard}}$ and $T'_{\text{New}}$ denote the 3DST-measured wave amplitude using the "standard" method and "new" composite S-transform method described in Sect. 3.4.1.

for the eight synthetic wave packets are shown in the top section of Table 1. Figure 3a shows our synthetic wave field, with wave packets 1 to 8 numbered accordingly. Temperature perturbations are evaluated at $\pm 1$ K on 3-D isosurfaces, while black arrows indicate the direction of phase propagation for each wave packet.

Our synthetic wave field is not designed to be especially realistic compared to AIRS observations (for this we have the real AIRS observations). It is instead designed to provide examples of a wide range of wave packets of different shapes, sizes, wavelengths and orientations such that we can assess the full performance of our 3DST application more generally. As such, we do not add simulated retrieval noise to the synthetic wave packets here for visual clarity, but in previous studies it was found that our S-transform application is sufficiently resistant to noise due to its ability to spectrally separate small-scale speckle noise from real signals Hindley et al. (2016). Even if we added such noise, we could simply ignore it by not analysing for these small-scale speckle noise features, so its inclusion is not useful here. A full description of the treatment of retrieval noise for real AIRS measurements in this study is provided in Appendix B.

We compute the 3-D S-transform of the synthetic wave field, following the method described in Sect. 3.2. The bottom section of Table 1 shows the central measured wave properties. Figure 3b shows the absolute 3DST-measured wave amplitude $|\mathcal{A}(x,y,z)|$, evaluated at isosurfaces of 1 K. For each wave packet, orange surfaces show the 1 K isosurface using the standard

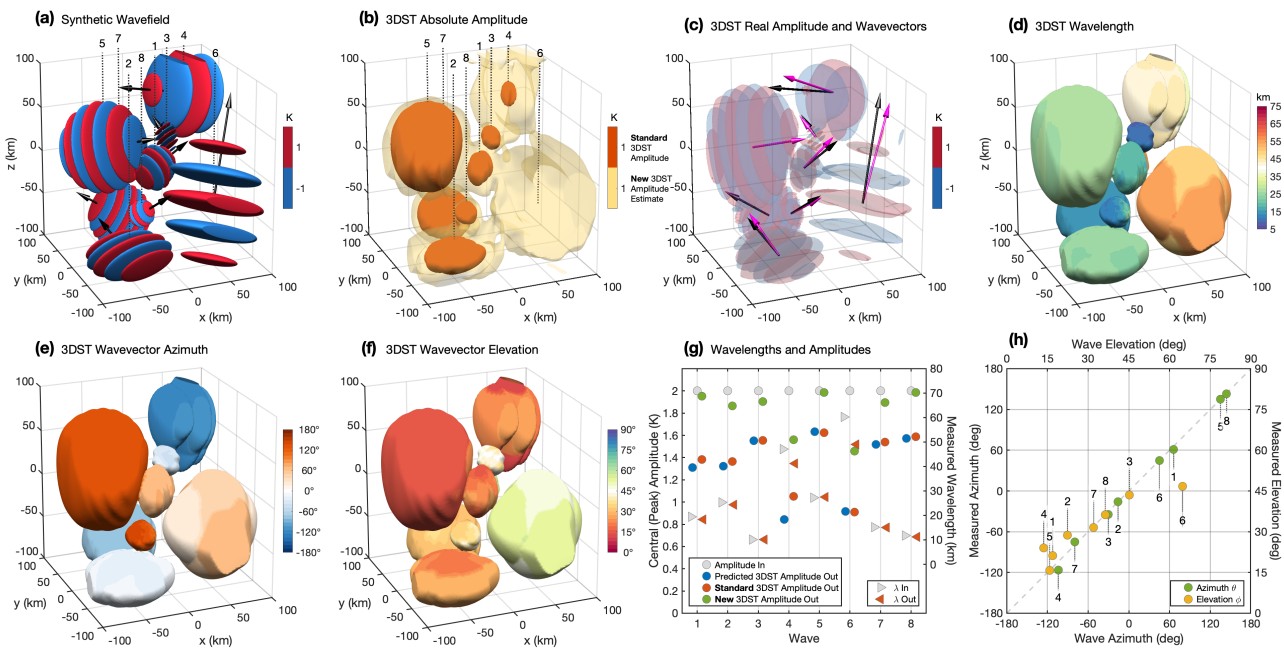

**Figure 3.** Results of 3-D Stockwell transform (3DST) analysis of a synthetic wave field. Panel (a) shows temperature perturbations of the input wave packets, with isosurfaces evaluated at $\pm 1$ K and black arrows indicating the direction of phase propagation. Absolute 3DST-measured wave amplitudes $|\mathcal{A}(x,y,z)|$ are shown in panel (b), where orange and yellow isosurfaces are drawn at 1 K for the "standard" and "new" 3DST amplitude measurement methods respectively, as described in Sect. 3.4.1. Real 3DST-measured temperature perturbations $\Re[\mathcal{A}(x,y,z)]$ from the "new" method are shown in panel (c), where black (pink) arrows indicate the true (measured) direction of phase propagation. Panels (d), (e) and (f) show 3DST-measured wavelengths, wavevector azimuths and elevations, relative to the positive $y$ axis and $x, y$ plane respectively, evaluated on 1 K isosurfaces. Coloured grey circles in panel (g) shows the central (maximum) input temperature amplitudes for each wave packet, with blue, orange and green coloured circles showing the predicted, "standard" and "new" 3DST-measured wave amplitudes at the centre of each wave packet respectively. Input and 3DST-measured wavelengths evaluated at the centre of each wave packet are shown by grey and orange triangles. Finally, panel (h) shows input versus 3DST-measured azimuth (green circles) and elevation (yellow circles) evaluated at the centre of each wave packet. The values in panels (g) and (h) are also listed in Table 1. For more details, see Sect. 3.5.

3DST amplitude measurement, while yellow surfaces (which are slightly transparent) show the 1 K isosurface using the new "composite" S-transform method described in Sect. 3.4.1.

The location of each wave packet in Fig. 3b is clearly identifiable. However, the physical extent of the 1 K isosurfaces for the standard S-transform amplitude measurements (orange) is smaller than the outer extent of the $\pm 1$ K isosurfaces shown in

5   Fig. 3a. This indicates that several of the waves have experienced amplitude attenuation in the standard S-transform method, as expected and discussed in Sect. 3.4.1. The attenuation effect is worst for synthetic waves numbered 4 and 6, where the peak

central wave amplitudes are only 1.05 K and 0.91 K respectively. For wave packet number 6, this means that no 1 K isosurface can be drawn in Fig. 3b, since no part of the measured wave packet is above 1 K.

The measurement of the absolute wave amplitude of the synthetic wave packets is much improved when the "new" composite S-transform method is used, as shown by the yellow isosurfaces shown in Fig. 3b. The outer extent ($|T'| \approx 1$ K) of each wave packet is well identified by the new method, and this outer limit closely resembles the extent of the wave packets in Fig. 3a. The peak central amplitudes are also much closer to their original values, as shown in Table 1. The edges of the yellow isosurfaces in Fig. 3b are not as smooth as those drawn for the standard absolute S-transform amplitudes. This is because amplitudes measured using the composite method do not discriminate within the range of frequencies that we have specified, which means that it is more sensitive to other overlapping wave-packets at the boundaries of each one.

Figure 3c shows the real part $\Re[\mathcal{A}(x, y, z)]$ of the measured wave amplitude using the new composite S-transform method. Taking the real part of the dominant wave amplitude at each location creates a "reconstruction" of the input wave field, which can provide a useful sanity check on the effectiveness of the 3DST. Black and pink arrows indicate the directions of the input and measured wavevectors respectively, evaluated at the centre of each wave packet. The input and measured directions are also shown in Table 1. The transparency of the isosurfaces here has been set so as to see the wavevector arrows clearly. The input and measured wavevector directions are in very good agreement, with directional errors typically less than around $\pm 10°$. Wave packets 4 and 6 are less well-measured however. This is due to the combination of their inclination and wavelength. Since they are inclined close to the horizontal and vertical directions respectively, their wavelengths in these directions are very long. Since we use the Discrete (fast) Fourier Transform (DFT) algorithms to compute the S-transform, spectral resolution is quite poor for wavelength that are large compared to the physical extent of the domain. This results in reduced spectral and directional precision when measuring wave packets such as these. The resulting directional error for wave packets 4 and 6 is still less than 20° for these two cases, so we do not believe that this will significantly affect our results in this study.

One of the key strengths of the S-transform is the ability to localise spectral characteristics at every location in $x, y, z$. Here, for the 3DST, our three dominant wavenumber objects $\mathcal{K}_x(x, y, z)$, $\mathcal{K}_y(x, y, z)$ and $\mathcal{K}_z(x, y, z)$, are used to find the 3DST-measured wavelength $\lambda = \sqrt{\mathcal{K}_x^2 + \mathcal{K}_y^2 + \mathcal{K}_z^2}$, azimuth $\theta = \tan^{-1}(\mathcal{K}_x/\mathcal{K}_y)$ relative to the positive $y$ axis and elevation $\phi = \tan^{-1}\left(\frac{\mathcal{K}_z}{\sqrt{\mathcal{K}_x^2 + \mathcal{K}_y^2}}\right)$ relative to the $x, y$ plane. These 3-D objects are then evaluated at the $|\mathcal{A}(x, y, z)| = 1$ K isosurface in Figs. 3c, 3d and 3e respectively. The colours of the isosurfaces drawn in panels (c-e) indicate the respective values at that location. These results indicate that the wavelength, azimuth and elevation of each wave packet is well-localised by our 3DST method.

Figs. 3g and 3h are graphical representations of the values listed in Table 1. Coloured circles in Fig. 3g indicate the input (grey), predicted (blue), "standard" measured (orange) and "new" measured (green) central wave amplitudes. Predicted values are calculated using the relation in Eqn. 6, since here the wavelength and size of the amplitude envelope of each wave packet is known. In most cases, the predicted amplitude attenuation is almost exactly what is measured by the standard S-transform approach. In the case of wave packet 4 however, the measured amplitude is higher than the predicted value. We suspect that this could be due to the proximity of the wave packet to the corner of the domain, which could lead to less severe attenuation. Green circles show that the "new" composite S-transform method dramatically improves the central peak amplitude measurement for

most wave packets. Once again however, the amplitude measurement of wave packets 4 and 6 is improved under the new method, but significant attenuation is still present. We suspect this is due to the poor spectral resolution in the DFT for these packets due to their inclination as discussed above, which leads to more spectral leakage into adjacent frequencies. Even when analysed with the composite S-transform method, the "voice Gaussians" for the appropriate frequencies are not large enough to counteract this. Adjusting the scaling parameter $c$ could improve this by increasing the size of the "voice Gaussians", but this could compromise our spectral resolution for other wave packets.

Grey and orange triangles in Fig. 3g show the input and measured wavelengths at the central location of each wave packet, while green and yellow circles in Fig. 3h show the input and measured azimuths and elevations respectively. There is good agreement in the input and measured wavelengths and directions for all wave packets except wave packets 4 and 6, for the same reasons discussed above.

In this section our 3DST method was applied to a synthetic wave field. We show that our 3DST method can localise and measure wave amplitudes, wavelengths and directions with either high or as-predicted accuracy. By applying a new composite S-transform method, we show that much of the amplitude attenuation inherent in the standard S-transform amplitude measurement can be significantly reduced. However, we have also highlighted some limitations that can arise from the use of DFT algorithms for wave packets at very high or very low inclinations. In the next section, our 3DST method is applied to real 3-D AIRS temperature perturbation measurements of stratospheric gravity-waves.

## 4  3-D Stockwell transform analysis of 3-D AIRS data

Now that we have tested our 3DST method on a synthetic wave field, we next apply the method to real AIRS measurements of stratospheric gravity waves. We begin with 3-D AIRS temperature perturbation measurements as described in Sect. 2.1. Background temperature variations are removed via a 4[th]-order polynomial cross-track fit and the resulting gravity-wave perturbations are regridded onto a regular distance grid in the horizontal and vertical directions. The perturbations are then localised to the usable altitude window of altitudes between $z = 20 - 60\,\mathrm{km}$, where retrieval noise is lowest.

Before we compute the 3DST, the temperature perturbations are multiplied by an exponential scale factor $\kappa(z) = e^{-\frac{z-40}{2H}}$, where $H = 7\,\mathrm{km}$ is the approximate scale height for the atmosphere, in order to temporarily remove the exponential increase of wave amplitude with decreasing atmospheric density at higher altitudes. This step effectively "normalises" wave packet amplitudes to what they would be if the same wave was observed at $z = 40\,\mathrm{km}$, at the centre of our usable height range. We found that this is an important step, since the significantly increased amplitudes of waves at higher altitudes sometimes artificially dominated the localised wave spectra over waves at lower altitudes in the 3DST. After the transform, 3DST-measured wave amplitudes are divided by $\kappa(z)$ to restore them to their true values[2].

---

[2]It should be mentioned that exponential growth with altitude cannot always be assumed in the general case, and in reality growth is highly dependent on local wind conditions and wave forcing (Kaifler et al., 2015; Kruse et al., 2016; Fritts, 2016). Here however, since here the factor $\kappa(z)$ is immediately removed from measured amplitudes after the S-transform, any deviations from exponential growth are preserved, so this assumption is very unlikely to adversely affect our results. Further, Wright et al. (2015, their Fig. 10a) showed that gravity wave potential energy measurements from satellite measurements over the southern

For each granule of AIRS measurements, the 3DST of our scaled and windowed gravity wave temperature perturbations $T'(x,y,z)$ is computed via the method described in Sect. 3.2. We set $\eta = 1000$ to analyse for the dominant 1000 frequencies in each granule. For AIRS measurements, we apply limits to these frequencies: we only analyse for wavelengths greater than 40 km in the horizontal and wavelengths greater than 6 km in the vertical. This is to ignore pixel-to-pixel scale variations which are very likely to be indistinguishable from noise, as discussed in Hindley et al. (2016). We also only analyse for negative vertical wavenumbers. This means that we make the assumption that all waves are upwardly propagating in order to break the "upwards and forwards" versus "downwards and backwards" ambiguity (e.g. Alexander et al., 2009; Wright et al., 2016; Ern et al., 2017). While we suspect the majority of waves to be upwardly propagating, we recognise that some fraction may be propagating downwards (e.g. Zhao et al., 2017). However, we do not believe this fraction is large enough to invalidate our results. Our reasoning for this is explained in Sect. 6.3. We also apply the new "composite" S-transform method as described in Sect. 3.4.1 for improved measurement of localised wave amplitudes.

As described in Sect. 3.1, the resulting 6-dimensional 3DST object for each AIRS granule, is collapsed down to 3-D objects containing the dominant wave amplitudes, phases and frequencies in the cross-track, along-track and vertical directions at each location in the granule. Using the azimuth of the along-track direction we then project these local cross-track and along-track frequencies to zonal and meridional wavenumbers $k$ and $l$. The vertical (spatial) frequencies in the 3DST method here are equivalent to vertical wavenumber $m$.

The final result of our 3DST analysis is spatially localised measurements of the dominant wave amplitude and phase in the complex object $\mathcal{A}(x,y,z)$, together with full 3-D wavevector $\mathcal{K}_x(x,y,z)$, $\mathcal{K}_y(x,y,z)$ and $\mathcal{K}_z(x,y,z)$ at each location between $z = 20 - 60$ km on our regular distance grid for each AIRS granule.

## 4.1 Case Study 1: 3-D gravity-wave measurements over the Southern Andes and Antarctic Peninsula

Before we extend our 3DST method to investigate gravity wave characteristics over the entire Southern Ocean, we first apply the method to two case studies over known hot spots of gravity wave activity.

The first case study is over the hot spot region around the Southern Andes and Antarctic Peninsula, one of the most intense regions of stratospheric gravity-wave activity on Earth (e.g. Hoffmann et al., 2013, 2016). Figure 4a shows AIRS 3-D temperature perturbations over this region during an overpass on 1$^{\text{st}}$ June 2010 (granule numbers $053 - 054$). This is the same overpass as was shown in Fig. 1a, focusing on the region outlined by the dashed lines.

These granules were analysed with the 3DST method, and the results of this analysis are shown in Figures 4(b-f). In order to display the observed and measured values in Fig. 4 clearly, the exponential scale factor $\kappa(z)$ is used. In Figs. 4a and 4b, isosurfaces are drawn at $T' \times \kappa(z) = \pm 2$ K. These isosurfaces are then coloured with the real values of $T'_{\text{AIRS}}$ and $T'_{\text{3DST}}$ respectively. Likewise, isosurfaces are drawn at $|T'_{\text{3DST}}| \times \kappa(z) = 2$ K and $|T'_{\text{3DST}}| \times \kappa(z) = 4$ K in Figs. 4(c-f). These surfaces are then coloured with the 3DST-measured absolute amplitudes, wavelengths and horizontal wavevector directions as shown. The outer isosurfaces in Figs. 4(c-f) have been made slightly transparent in order to see the internal structure. The noise

---

Andes closely follow lines of exponential growth in the stratosphere of roughly $e^{\frac{z}{2H}}$ above and below $z = 40$ km, so $\kappa(z)$ should be a reasonable factor to apply.

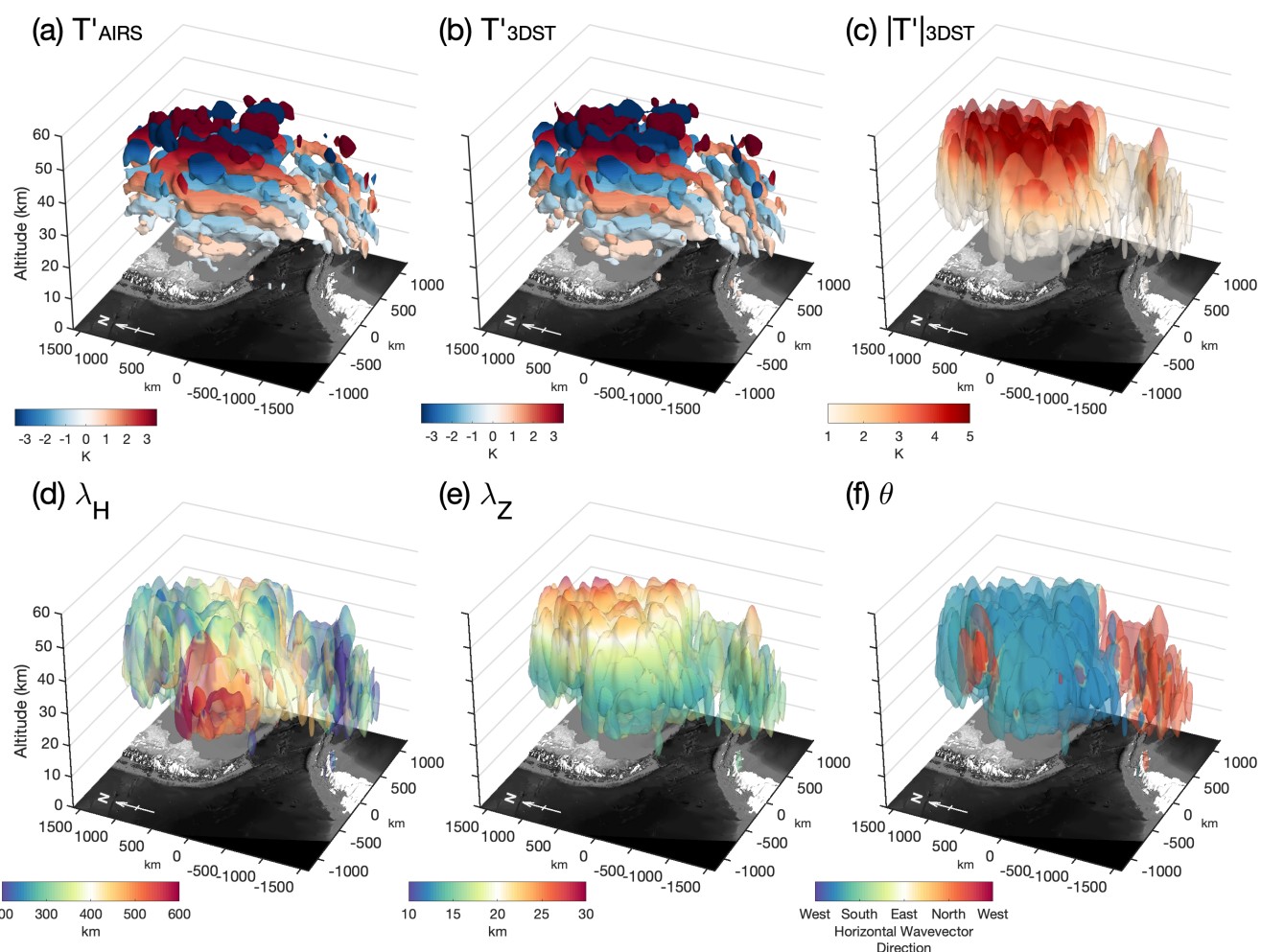

**Figure 4.** 3DST analysis of 3-D AIRS temperature measurements over Southern Andes and Antarctic Peninsula on $1^{st}$ June 2010 (granule numbers $053-054$). Panels (a) shows the observed AIRS temperature perturbations $T'_{AIRS}$. Panels (b) and (c) show the real $T'_{3DST}$ and absolute $|T'|_{3DST}$ 3DST-measured wave amplitude respectively between $z = 20 - 60$ km. Panels (d), (e) and (f) show 3DST-measured horizontal wavelength $\lambda_H$, vertical wavelength $\lambda_Z$ and azimuth $\theta$ respectively. The outer isosurfaces in panels (c-f) have been made slightly transparent in order to see the internal structure. A grayscale map of the region has been plotted underneath the isosurfaces, with topography height shown to scale and an arrow indicating the direction of North. See text for details.

threshold derived in Sect. 2.2 is not applied to these case studies such that the full range of 3DST-measured amplitudes can be seen.

Figure 4a shows several gravity wave phase fronts stacked vertically over the Southern Andes and Antarctic Peninsula. Wave amplitude on these outer isosurfaces increases with altitude, from around 1 K at $z = 25$ km to more than 3 K at $z = 55$ km. The

wave amplitude within these isosurfaces is likely to be much higher however, as can be seen in Fig. 1a. Assuming that these waves are upwardly-propagating mountain waves propagating into the prevailing westerly wind, the phase fronts over the Southern Andes appear to have a southward component, while the phase fronts over the Antarctic Peninsula appear to have northward component, as was seen in Fig. 1a, which is consistent with the results of previous studies (e.g. Wright et al., 2017). What is interesting is that at all heights shown here, the phase fronts appear to extend meridionally over the Drake Passage from the mountains to the north and south, converging towards latitudes around $60°$S.

Figures 4(b-f) show the results of our 3DST analysis of these AIRS measurements. Panels (b) and (c) show the real and absolute 3DST-measured wave amplitudes. The location, orientation and amplitudes of the measured phase fronts in Fig. 4b agree very well with those in the input AIRS measurements in Fig. 4a. The absolute wave amplitudes shown in Fig. 4c are also very close to their input values, with peak values located over the southern tip of South America. A secondary localised maximum is observed directly over the Antarctic Peninsula.

3DST-measured horizontal and vertical wavelengths are shown in Fig. 4d and 4e respectively. Measured horizontal wavelengths are generally between 300-400 km directly over the Southern Andes, while shorter horizontal wavelengths are measured over the Antarctic Peninsula, with values around 200 km. To the south west of the Southern Andes, longer horizontal wavelengths are measured with values close to 600 km. This is in good agreement with the structure observed in Fig. 1a. We can check these values for altitudes near $z = 40$ km by inspection of Fig. 1a, since both figures are plotted on a regular horizontal distance grid. The extent of the dashed region in Fig. 1a is $2400 \times 3200$ km in the zonal and meridional directions. This suggests an approximate horizontal wavelength of around 360 km for the region over the southern tip of South America, which is very similar to the values measured by the 3DST.

Vertical wavelengths generally increase with increasing altitude, from around 15 km at $z = 25$ km to around 25 km at $z = 55$ km. This is what we would expect, since mountain waves would be refracted to longer vertical wavelengths by the strong winds of the stratospheric jet (e.g. Eckermann and Preusse, 1999). By inspection of ERA5 reanalysis winds for this period, we find that zonal winds around $60°$S steadily increase from the surface to altitudes around $z = 50$ km, reaching values around 70 ms$^{-1}$. Above $z = 50$ km, wind speeds remain steady for around 5 km and then decrease to around 30 ms$^{-1}$ at altitudes around $z = 80$ km. From this we expect vertical wavelengths to be refracted up to around 25 km at altitudes around $z = 50$ km, using the relation in Eckermann and Preusse (1999, their Eqn. 1), although we note that the accuracy of modelled stratospheric parameters in reanalyses at these latitudes can be variable (Wright and Hindley, 2018).

Our vertical wavelength results are thus consistent with this, although we note that there is a significant reduction in vertical resolution for altitude above $z = 50$ km for the 3-D AIRS retrieval, as shown in Fig. 2b, so we would not expect to be able to measure a reduction in vertical wavelengths this height as zonal wind decreases. Further, as discussed in Sect. 3.5, the spectral resolution of the DFT algorithms is relatively poor for wavelengths that are quite long compared to the measurement window. On the other hand, sensitivity of the 3-D AIRS measurements to longer vertical wavelengths is very good as shown in Fig. 2c. To check our results, visual inspections and measurements of waves were performed for vertical cross-sections through the observations in Figs. 4 and 5, and good agreement was found with our 3DST-measured vertical wavelengths in each case.

One of the key strengths of our 3DST method is the ability to localise and measure wavevector directions. In the horizontal, this works very well. We can constrain the azimuth of the propagation direction of horizontal wavevectors $\theta = \tan^{-1}(k/l)$ by assuming upward propagation (i.e. negative vertical wavenumber $m$), which is very likely to be the case for the waves in Fig. 4 since they are likely to be mountain waves. Over the Southern Andes in Fig. 4f, a large wave packet is measured as propagating southwest, whereas the smaller wave packet over the Antarctic Peninsula is measured to be propagating northwest. This is in good agreement with what we can infer by inspection of Fig. 1a. A small region of northwest-propagation is observed over the mountains of the Southern Andes at altitudes between $z = 40 - 45$ km approximately. While our confidence that this small region corresponds to a real propagating gravity wave packet is probably quite low, it is in good agreement with a small structure observed in Fig. 1a at this location. This at least gives us confidence that the 3DST method is producing a fair spectral description of the input data.

## 4.2 Case Study 2: 3-D gravity-wave measurements over South Georgia Island

For our second case study, we select gravity-wave temperature perturbation measurements from an AIRS overpass of South Georgia (54°S, 36°W) on 5[th] July 2010 (granule numbers $034 - 035$). This is the same overpass as shown in Fig. 1b. South Georgia is an isolated, mountainous island in the Southern Ocean. It has mountains ranges nearly 3 km high, lies in a region of very strong tropospheric winds, and is an intense source of stratospheric gravity wave activity (e.g. Alexander and Grimsdell, 2013; Hoffmann et al., 2013, 2016; Hindley et al., 2016; Jackson et al., 2018; Garfinkel and Oman, 2018).

Figure 5a shows AIRS measured temperature perturbations over the island in the region specified by dashed lines in Fig. 1b. In the same manner as the case study over the Southern Andes in Sect. 4.1, these gravity-wave temperature perturbations were analysed using our 3DST method. The results of the analysis are shown in Figures 5(b-f). This example was selected as a more challenging case study for both the 3-D AIRS measurements and our 3DST method. The spatial features of the wave field are quite small compared to the horizontal resolution of the AIRS measurements, and consists of two possible northward and southward components located close to the south and north of the island.

Phase fronts are observed over South Georgia in Fig. 5a as a series of chevrons stacked vertically over the island. These chevrons point westwards into the prevailing wind. At least 2-3 vertical stacks of these chevrons are visible arranged one behind the other to the east of the island. This arrangement is characteristic of a mountain wave field from an isolated "point" source (e.g. Alexander et al., 2009; Vosper, 2015). These characteristic "chevron" patterns are also somewhat analogous to the wake patterns formed in the lee of ships and submarines in the oceans known as Kelvin wakes (e.g. Noblesse et al., 2016), which have been extensively modelled.

As in Fig. 4, Figs. 5(b-f) show the results of our 3DST analysis. Panels (b) and (c) show the real and absolute 3DST-measured wave amplitudes using the new "composite" S-transform method. The location, orientation and amplitudes of the measured phase fronts in Fig. 5b agree well with the input AIRS measurements in Fig. 5a. The absolute wave amplitudes shown in Fig. 5c are also very close to their input values, with peak values exceeding 5 K located directly over the island. Larger absolute wave amplitudes in the internal structure of the wave field in Fig. 5c also appear to extend downwind in the characteristic chevron-shaped pattern.

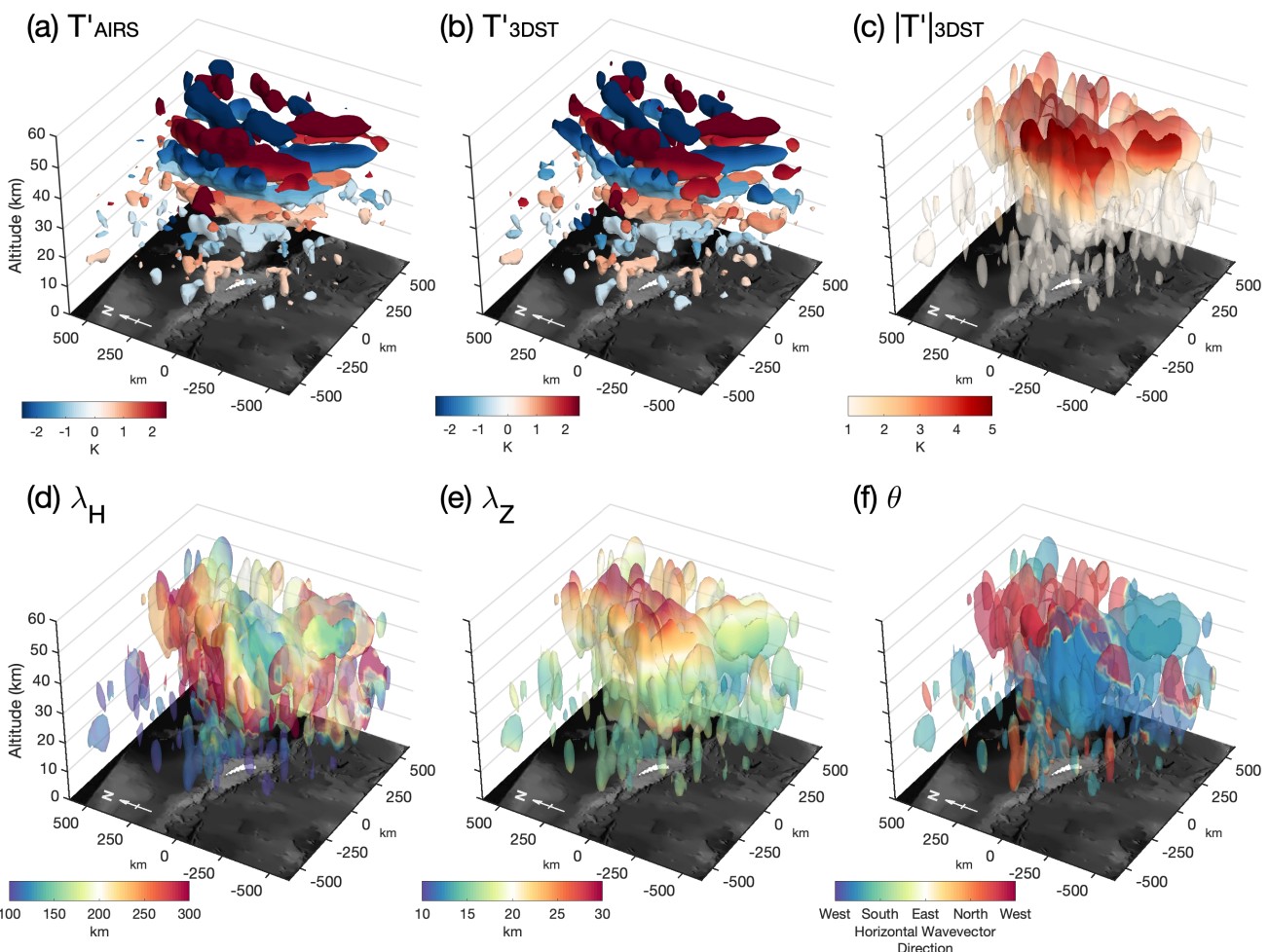

**Figure 5.** As Fig. 4, but for 3DST analysis of 3-D AIRS temperature measurements over the island of South Georgia (54°S, 36°W) on 5th July 2010 (granule numbers $034 - 035$). For details, see text in Sect. 4.2.

3DST-measured horizontal and vertical wavelengths are shown in Figs. 5d and 5e. Short horizontal wavelengths around 150 km are measured directly over the island. These values increase to around 250 km in the downwind section of the wave field, but horizontal wavelengths are larger towards the outer regions. Isolated regions of very short horizontal wavelengths less than 100 km are seen upwind (to the west) of the island. These correspond to the small speckles of uncorrelated amplitudes seen in Fig. 5a. These are almost certainly noise artefacts with small amplitudes. When the noise threshold of 1.5 K derived in Sect. 2.2 is applied, these features are mostly removed. As in Fig. 4, vertical wavelengths in Fig. 5e are observed to steadily increase with increasing altitude. The longest vertical wavelengths, up to $25 - 30$ km, are generally observed directly over the island.

Assuming upward wave propagation, which is very likely for a characteristic mountain wave field such as this, the directions of measured horizontal wavevectors can be constrained. In Fig. 5f, northwestward directions are observed to the north of the island and southwestward directions to the south. This is characteristic of a typical mountain wave pattern from an isolated source (e.g. Alexander et al., 2009). Crucially, the measured change in direction from northward to southward appears to occur directly over or in line with the island, which is in very good agreement with the wave pattern observed in Fig. 1b. The southward component in Fig. 5f appears to be slightly more dominant at lower altitudes directly over the island. Despite the relatively small physical extent of the wave field over South Georgia, the fact that we are able to accurately measure and then localise the opposing directions of measured horizontal wavevectors gives us confidence in the ability of the 3DST method to constrain the relative components of directional momentum fluxes.

## 5 Wintertime gravity-wave characteristics over the Southern Ocean

In this section, we extend the application of our 3DST method to 3 months of wintertime AIRS measurements over the entire Southern Ocean during June-August 2010. As discussed in Sect. 1, this region is important due to the long-standing "cold-pole problem", where unresolved gravity-wave drag is a leading candidate for the strong wind and temperature biases found in nearly all modern GCMs (e.g. McLandress et al., 2012; Garfinkel and Oman, 2018). Understanding the nature of gravity waves around $60°$S, and measuring their momentum fluxes, is key to solving this problem.

The 3DST analysis method is applied to each granule of 3-D AIRS temperature perturbation measurements during June, July and August 2010. The resulting 3-D measurements of wave amplitude, phase and horizontal and vertical wavevectors are then regridded from the geolocated AIRS scan-track onto a horizontal regular distance grid of $\Delta x = \Delta y = 40\,\mathrm{km}$, centred on the south pole. Since we focus our study on gravity wave properties over the Southern Ocean, the choice of this regular, orthogonal grid provides a more favourable viewing geometry and better spatial localisation of features at mid and high latitudes than would be the case if a latitude-longitude grid were used.

### 5.1 Wave-occurrence frequency

As an initial investigation, we first apply a simple method to estimate the frequency of occurrence of gravity waves over the Southern Ocean during winter. AIRS measurements are made over the same geographical region, on average, twice a day (less often in equatorial regions and more often at high latitudes). Here, we define gravity-wave occurrence frequency during each month as the fraction of AIRS scan-track pixels within each bin on our regular distance grid whose 3DST-measured absolute wave amplitude is greater than the noise threshold of 1.5 K at $z = 40\,\mathrm{km}$. Due to retrieval errors, there will be some uncertainty in this approach, but it provides a good indication of the likelihood of gravity-wave observations over specific regions, thus revealing any hot spots of wave activity.

Fig. 6 shows monthly wave occurrence frequency at $z = 40\,\mathrm{km}$ over the southern hemisphere for June-August 2010. Several key features are apparent. A familiar belt of increased gravity-wave occurrence is observed around the latitude band near $60°$S, and appears to move poleward through June-August. This is in good agreement with observations of gravity-wave potential

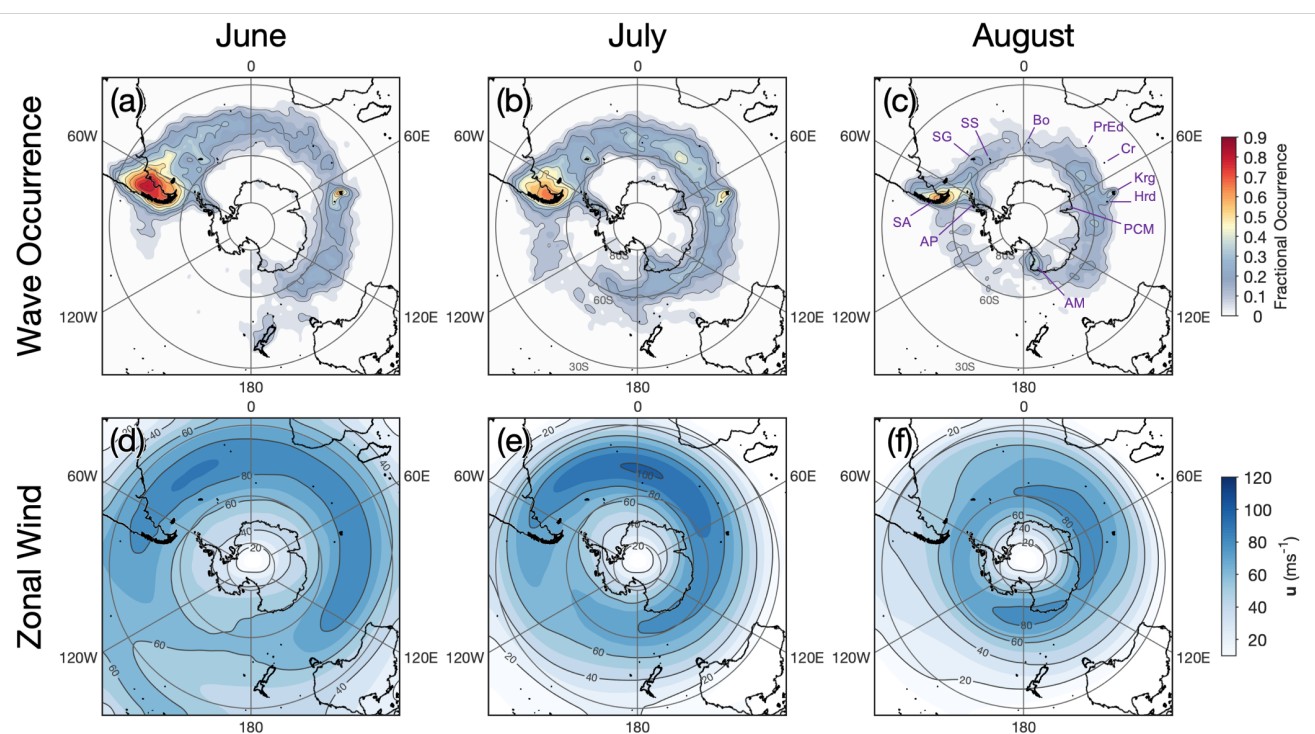

**Figure 6.** Monthly gravity wave occurrence frequency at $z = 40$ km over the southern hemisphere for June, July and August 2010 (a-c). Wave occurrence is defined here as the fraction of AIRS pixels with 3DST-measured absolute wave amplitude above a noise threshold of 1.5 K. Panels (d-f) show monthly-mean zonal wind at $z = 40$ km from ERA5 reanalysis for the same time period. For reference in later sections, the locations of the Southern Andes (SA), Antarctic Peninsula (AP), South Georgia (SG), and the South Sandwich (SS), Bouvet (Bo), Prince Edward (PrEd), Crozet (Cr), Kerguelen (Krg) and Heard (Hrd) Islands are marked in panel (c), along with the locations of the Prince Charles (PCM) and Admiralty (AM) Mountains on the continent of Antarctica.

energy in previous studies (e.g. Hendricks et al., 2014; Hindley et al., 2015). Over the Southern Andes at around 45°S, very high wave occurrence frequencies of up to 90% are observed during June, indicating that a wave was measured with an amplitude above the noise threshold almost every time the AIRS instrument passed over this region. During July and August, this drops to around 65% and 60% respectively, with the point of highest wave occurrence moving poleward towards the southern tip of
5    South America.

It is also notable that the Kerguelen Islands, Heard Island and South Georgia are located under regions of increased wave occurrence. During June, wave occurrence frequencies of up to 60% and 45% are observed over the Kerguelen and Heard Islands respectively. Over South Georgia, wave occurrence frequencies of up to 40% are observed during June and July. Over New Zealand and Tasmania, isolated regions of increased wave occurrence up to 20% are observed during June and July.

10    The spatial distribution of these regions of increased gravity wave activity are in good agreement with previous studies such as Hoffmann et al. (2013), who identified hot spots of wintertime stratospheric gravity-wave activity in the southern hemisphere

from 2-D AIRS brightness temperature measurements, using a variance-based approach to define regions of increased wave activity. Using a visual identification method, Alexander and Grimsdell (2013) identified orographic gravity wave events in AIRS brightness temperature measurements over small islands in the Southern Ocean using a series of selection criteria. During July of 2003 and 2004, they found wave occurrence frequencies of 72% and 36% (with an error of around 8%) over

the Heard and Kerguelen Islands and South Georgia respectively. This is broadly consistent with our results here, even though we use a different method and consider a different year.

Hoffmann et al. (2016) used AIRS brightness temperature measurements and a variance-threshold method to investigate several hot spots of orographic gravity wave activity in the southern hemisphere, such as mountain ranges and small islands. In a 12-year composite of measurements during April-October, they found overall "wave-event frequencies" of 59.1%, 56%,

34.4%, 13.5%, 36.3% and 44.1% over the Southern Andes, Antarctic Peninsula, Kerguelen Islands, New Zealand, Heard Island and South Georgia respectively. Despite methodological differences, the spatial distribution of localised gravity-wave hot spots in Hoffmann et al. (2016) and their relative frequencies of occurrence are in good agreement with our results here, although much of the broad belt of increased gravity-wave activity is removed when their method is applied.

Figures 6(d-f) show monthly-mean zonal wind speed $u$ at an altitude near $z = 40\,\mathrm{km}$ for June, July and August 2010 derived

from 3-hourly ERA5 (Copernicus Climate Change Service, 2017) reanalysis data from the European Centre for Medium-Range Weather Forecasts (ECMWF). Positive values of $u$ indicate a westerly (i.e. eastward) direction. The location of the edge of the stratospheric polar vortex (also known as the polar night jet) is clearly visible. Largest zonal wind speeds of up to $100\,\mathrm{ms^{-1}}$ occur at latitudes around 45°S in an arc clockwise from longitudes around 60°W to 120°E during June and July. This pattern then moves poleward towards around 60°S during August, with a reduction in zonal wind speed to peak values around $80\,\mathrm{ms^{-1}}$.

The spatial distribution of gravity-wave occurrence frequency in Figs. 6(a-c) shows some similarity to the spatial distribution of strong zonal winds in the stratosphere. This is to be expected, since increased wind speed with altitude refracts waves to longer vertical wavelengths, increasing their likelihood of detection by AIRS. Further, strong westerly winds can provide a vertical "conduit" through which westward-propagating gravity waves generated in the troposphere may ascend into the stratosphere without encountering critical layers. Measured wave occurrence frequency is largest over the Southern Andes

and the Kerguelen Islands when the central latitude of the jet is located directly over the mountains during June, but smaller when the jet moves poleward during August and the winds are weaker. Likewise, increases in wave occurrence frequency are observed over the Antarctic Peninsula and the Admiralty mountains when the central latitude of the jet is directly over them during August.

## 5.2   Amplitudes and wavelengths

Figure 7 shows monthly-mean measured wave amplitudes (a-c), horizontal wavelengths (d-f) and vertical wavelengths (g-i) from our 3DST analysis of AIRS temperature perturbations for June, July and August 2010 at an altitude of $z = 40\,\mathrm{km}$. The monthly-mean measured wave amplitudes shown are calculated using all 3DST-measured wave amplitudes, not just those that exceeded the noise threshold of 1.5 K.

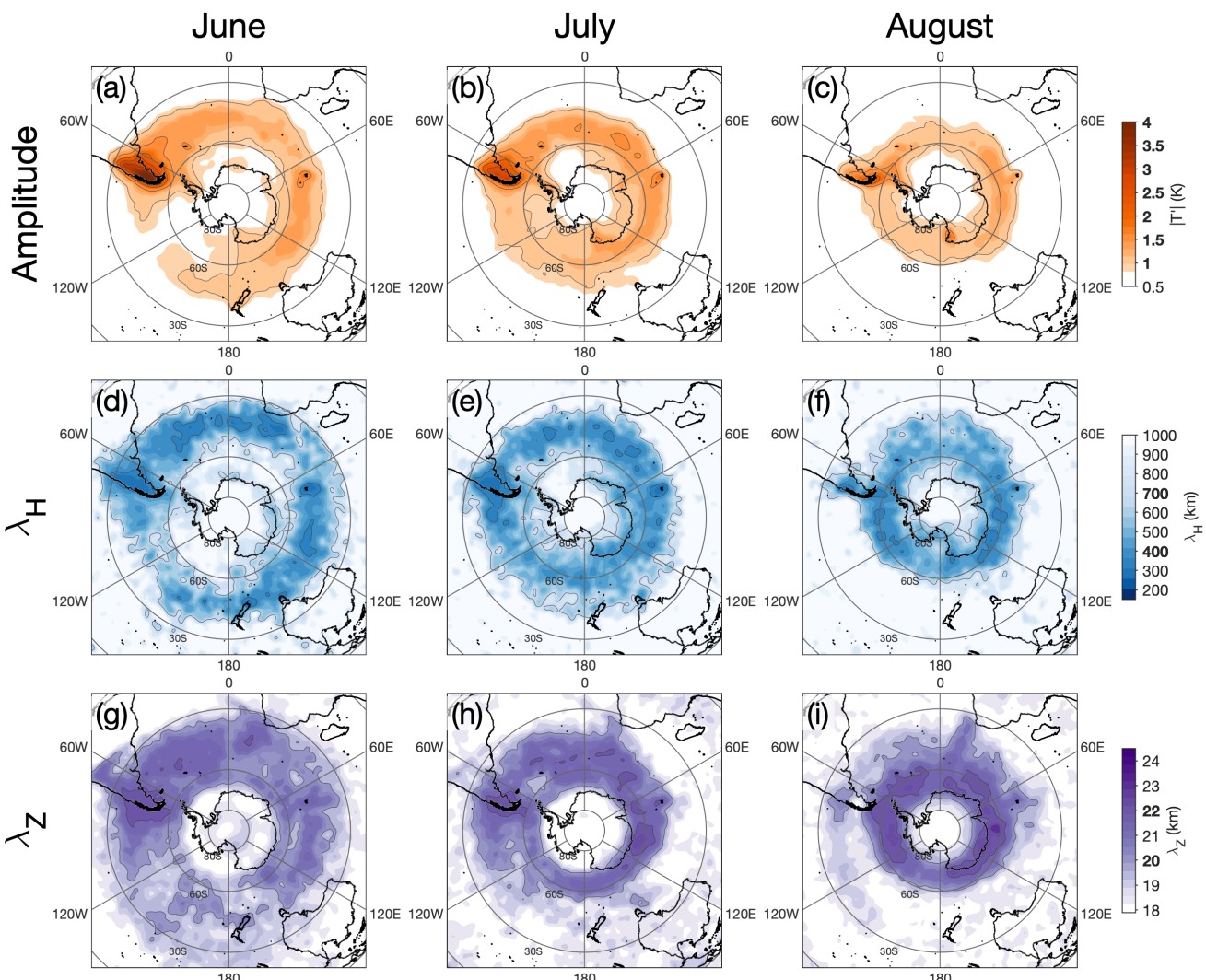

**Figure 7.** Monthly-mean gravity wave amplitude (a,b,c), horizontal wavelength (d,e,f) and vertical wavelength (g,h,i) at $z = 40$ km over the Southern Ocean during June, July and August 2010 from 3-D AIRS measurements. To guide the eye, thin grey contours are drawn at the values marked in **bold** on the colour bars.

As before, increased gravity-wave activity in a belt around $60°$S is observed during winter in Figs. 7(a-c), with monthly-mean wave amplitudes greater than $1 - 1.5$ K most of the way around the Antarctic continent. The Southern Andes dominates, with peak monthly-mean wave amplitudes of around 4, 3 and 2 K during June, July and August respectively. Other localised regions of increased wave amplitude are also visible even at this monthly-mean level, such over the mountains of the Antarctic
5    Peninsula, the Admiralty Mountains (located at around $170°$W) and small islands such as South Georgia, Heard and Kerguelen Islands. As is apparent in Fig. 6, there is a suggestion that during June, the "belt" of increased gravity-wave activity appears

to form a "tail" that maximises over the Southern Andes and Antarctic Peninsula then decreases as it extends far eastward, particularly during June and July. This has been seen in previous studies (e.g. Alexander et al., 2008; Yan et al., 2010; Hendricks et al., 2014; Hindley et al., 2015), and could suggest a connection between wave activity over the Southern Andes and, at least, some of the central and western portions of the observed belt of gravity-wave activity around $60°$S. A possible mechanism for this could be downwind propagation of gravity waves from the mountains either directly (Sato et al., 2012) or through the generation of secondary gravity waves with non-zero phase speeds as a result of wave breaking or intermittency due to variability in the stratospheric wind over the mountains (Woods and Smith, 2010; Bossert et al., 2017; Vadas and Becker, 2018). Increased gravity-wave activity over the Southern Ocean at all longitudes around $60°$S has also been attributed to waves generated by spontaneous adjustment mechanisms resulting from baroclinic instability around the vortex edge (Hendricks et al., 2014; Hindley et al., 2015; Plougonven and Zhang, 2014) or convection within fronts (Jewtoukoff et al., 2015; Plougonven et al., 2015; Holt et al., 2017). During August, non-orographic processes could begin to be more dominant over any connection between gravity-wave activity over the Southern Andes and over the Southern Ocean, leading to the more zonally uniform distribution of gravity-wave activity within this latitude band that we observe in Fig. 7c.

It is worth mentioning that our measured wave amplitudes are somewhat larger than those measured by Ern et al. (2017), who used the S3D method to analyse 3-D AIRS temperature perturbations globally at an altitude of $z = 36$ km for January 2009. This difference is likely to arise for several reasons, such as the known effect of wave amplitude averaging over the cube-size used in the S3D method, or that their analysis was performed at an altitude of $z = 36$ km, where we would expect a $\sim 25\%$ reduction in wave amplitude due to the exponential trend in wave amplitude with atmospheric density.

An interesting observation is that over regions where we would expect to observe lower gravity-wave activity during winter (i.e. far from $60°$S) in Figs. 7(a-c), monthly-averaged 3DST-measured absolute wave amplitudes were well below the noise threshold of $1.5$ K derived in Sect. 2.2. This suggests that the noise threshold of $1.5$ K should be considered as more of a "confidence" threshold rather than a strict noise floor, since amplitudes well below this threshold are frequently observed in a geographical distribution that appears realistic.

Figs. 7(d-f) and 7(g-i) show monthly-mean measured horizontal $\lambda_H$ and vertical $\lambda_Z$ wavelengths respectively. Unlike Figs. 7(a-c), these means are calculated only for horizontal and vertical wavelength values where the 3DST-measured wave-packet amplitude was greater than the threshold of $1.5$ K.

Horizontal wavelengths between $300 - 600$ km in Figs. 7(d-f) are observed in the same spatial distribution as the observed belt of increased gravity wave activity around $60°$S in Figs. 7(a-c). Regions of shorter horizontal wavelengths between around $300 - 400$ km are observed over the Southern Andes, Kerguelen Islands and the southern tip of New Zealand during June and July. During August, a region of similarly short horizontal wavelengths is observed over the Drake Passage between the Southern Andes and Antarctic Peninsula. By inspection of the observed gravity-wave perturbations in Figs. 1(a-c), these measurements of average horizontal wavelengths for strong wave events are reasonable. The wave structure in Fig. 1c, for example, suggests that large-scale and large-amplitude gravity waves with wavelengths comparable to the values shown in Figs. 7(d-f) are not limited to the immediate vicinity of orography, and could be non-orographically generated far out over open ocean. Thus, the observation of average horizontal wavelengths between around $300 - 600$ km at all longitudes around

60°S seems reasonable, within the observational filter of our data and method. From our sensitivity assessment in Fig. 2c, we can see that these wavelengths lie within our region of highest sensitivity. It is worth mentioning that measured horizontal wavelengths over small mountainous islands could be lower, but their averaged monthly-mean values could be easily high-biased by contributions from transitory wave structures with much longer wavelengths, such as in the example in Fig. 1c.

Monthly-mean vertical wavelengths of around $19 - 23$ km are observed at almost all longitudes around the Southern Ocean in Figs. 7(g-i). Relatively long vertical wavelengths are observed over the Southern Andes during June and July, and towards the central latitude of the stratospheric polar vortex during July and August. This is what would we would expect, as the strong wind of the stratospheric jet refracts waves to longer vertical wavelengths, leading to preferential likelihood of observation by AIRS, and improved amplitude measurement due to the observational filter.

The range of vertical wavelengths shown in Figs. 7(g-i) is quite small. This is likely due to a combination of several factors. Firstly, the sensitivity of our gravity-wave retrieval is less than 50% for vertical wavelengths less than around 17 km, as shown in Fig. 2c. Secondly, we may be less likely to observe waves with vertical wavelengths longer than around 30 to 35 km over open ocean since, generally speaking, a reasonably strong and stable vertical wind column and a persistent source (such as wind flowing over a mountain range, or the centre of an intense storm) are required to refract gravity waves to such long

vertical wavelengths. As a result, we measure vertical wavelengths generally within around 15 to 30 km, which when averaged over a month converge towards the fairly narrow ranges of values we see in Fig. 7.

In our study, 3DST measurements of regions with very little resolvable gravity-wave activity (e.g. regions equatorward of 30°S in Fig. 7) tend to be measured as having very long horizontal wavelengths ($\lambda_H \gg 1000$ km, i.e. a flat horizontal field). This is a result of our methodological approach for dealing with "empty" regions without clear gravity-waves signals. If there

are no clear gravity waves, and we have excluded the small-scale retrieval noise via the steps in Appendix B, the only remaining undulations that we can measure are large-scale flat features that correspond to very long horizontal wavelengths, which are not likely to be physical. This is evident in the white regions of Figs. 7(d-f), where the colour scale is saturated for horizontal wavelengths much greater than 1000 km. These regions generally have very low amplitudes, but even once our noise threshold is applied, some small patches of these regions do persist (see Fig. B2) and start to aggregate in the monthly averages. In the

vertical, artefacts from differences between vertical layers start to dominate in these empty regions, so the resulting vertical wavelength measurements are very short ($\lambda_Z < 6$ km).

Fortunately however, long horizontal wavelengths and short horizontal wavelengths correspond to very low momentum fluxes via Eqn. 8, so artefacts from these regions are not likely to significantly affect our flux results in Sect. 5.3. In future studies it may be sensible to mask out these regions of obvious low wave activity in order to make measured wavelength results

clearer, but here it is useful to understand the kind of results the 3DST produces when analysing such empty regions. In the study of Ern et al. (2017, their Fig. 2), where the S3D method was used to analyse 3-D AIRS measurements during January 2009, large regions of low wave activity are measured as having very short horizontal wavelengths, which suggests that their analysis method has a different approach for dealing with empty regions that do not contain clear gravity-wave signals. Neither approach is right or wrong, but it is important to consider these methodological differences when comparing results from

different methods for regions with very little wave activity to ensure consistency.

In future studies, we could attempt to improve the vertical resolution of our 3DST analysis method for long vertical wavelengths by using the 2-D S-transform (2DST). By computing the 2DST of adjacent vertical layers, then computing spectral co-variance between them, a phase shift between the adjacent layers could be found. This phase shift could be used to infer the vertical wavelengths of co-varying gravity-wave signals between the adjacent layers. Although this could introduce a low-bias in measured vertical wavelengths due to error in phase shift measurements as discussed in Hindley et al. (2015), the resulting improvement in resolution could lead to the technique being preferable for data sets with relatively small vertical measurement windows, such as 3-D AIRS measurements.

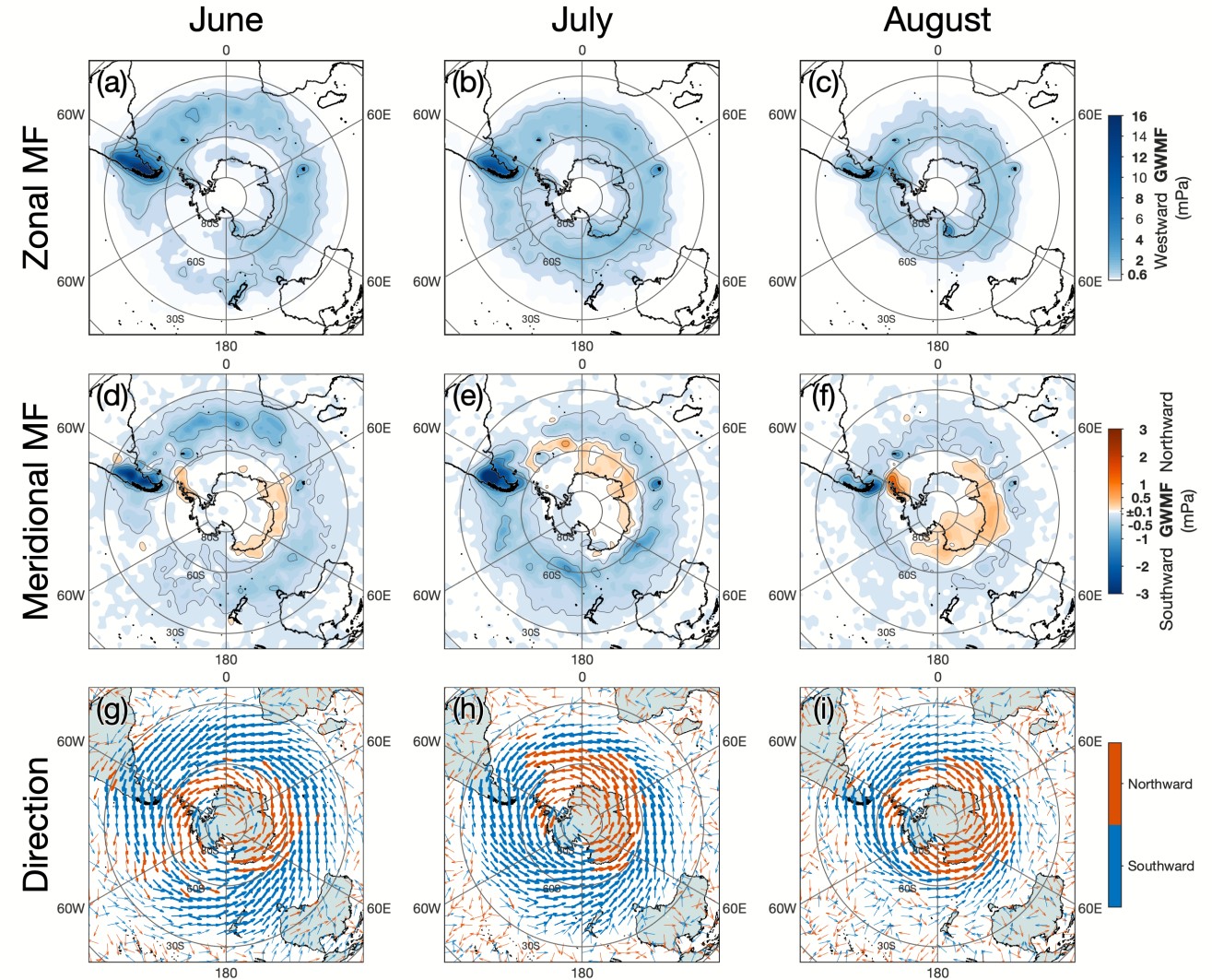

**Figure 8.** Monthly-mean zonal (a,b,c) and meridional (d,e,f) gravity wave momentum flux at $z = 40$ km during June, July and August 2010. Monthly-mean northward (southward) horizontal wavevector directions (g,h,i) are shown by orange (blue) arrows. The thicknesses of the arrows in (g,h,i) are scaled for mean wave amplitude in the corresponding months from Fig. 7(a-c).

### 5.3 Zonal and meridional momentum fluxes and horizontal wavevector directions

Using the mid-frequency approximation (Fritts and Alexander, 2003), the zonal and meridional components of gravity-wave momentum flux $MF_x$ and $MF_y$ can be estimated from measurements of wave amplitude, horizontal wavelength and vertical wavelength via the relation in Ern et al. (2004):

$$(MF_x, MF_y) = \frac{\rho}{2}\left(\frac{g}{N}\right)^2 \left(\frac{|T'|}{\bar{T}}\right)^2 \left(\frac{k}{m}, \frac{l}{m}\right) \tag{8}$$

where $\rho$ is atmospheric density, $g$ is the acceleration due to gravity, $N$ is the buoyancy frequency, $|T'|$ is our absolute 3DST-measured absolute wave amplitude, $\bar{T}$ is the background temperature, and $k = 2\pi/\lambda_x$, $l = 2\pi/\lambda_y$ and $m = 2\pi/\lambda_z$ are zonal, meridional and vertical angular wavenumbers respectively.

Figures 8(a-f) show monthly-mean zonal and meridional (d-f) gravity-wave momentum fluxes at an altitude of $z = 40\,\mathrm{km}$
for June-August 2010, determined using our 3-D AIRS measurements. The values shown only include measurements where the 3DST-measured absolute wave amplitude was greater than the noise threshold of 1.5 K.

Zonal gravity-wave momentum flux is entirely westward over the Southern Ocean during June-August, with values exceeding $\sim 0.6 - 2\,\mathrm{mPa}$ at almost all longitudes around 60°S in a similar "belt" pattern as was seen in Fig. 7. Largest fluxes are observed over the Southern Andes during June, with mean values exceeding 16 mPa. Additional regions of increased flux
are visible over the Antarctic Peninsula, South Georgia, the Kerguelen Islands, the Admiralty Mountains and New Zealand, although these values are nearly an order of magnitude smaller.

This broad belt of enhanced gravity-wave fluxes in winter is also in good agreement with results from satellite limb-sounding observations. Using lower-bound estimates of gravity-wave momentum fluxes from High Resolution Dynamics Limb Sounder (HIRDLS) and Sounding of the Atmosphere using Broadband Emission Radiometry (SABER) measurements, Ern et al. (2018)
showed a broad enhancement of GWMF at $z = 30\,\mathrm{km}$ in a characteristic belt around 60°S from around April to October in multi-year averages. Fluxes were found to be largest over the Southern Andes and Antarctic Peninsula during these months in Ern et al. (2018), as in our measurements here during June to August.

However, the belt of increased flux over the Southern Ocean shown in Ern et al. (2018) appears to be comparatively more pronounced during June to August in their study than we observe here in AIRS measurements. The observational filter of
limb-sounding instruments means that they are more sensitive to gravity waves with relatively short vertical wavelengths ($\sim 3-15\,\mathrm{km}$) and relatively long horizontal wavelengths ($\sim 500-5000\,\mathrm{km}$). This suggests that a significant part of the oceanic section of the belt of enhanced gravity wave activity at 60°S is made up of long-horizontal wavelength waves, to which AIRS is less sensitive. Over the mountains however, the comparatively strong peak in AIRS flux measurements in Fig. 8 suggests that waves here may be more likely to fall within the observational filter of AIRS, with relatively long vertical and short horizontal
scales as mentioned in Sect. 2.1.

Despite being the dominant single region of gravity-wave momentum flux at these latitudes, in our results the Southern Andes region (defined here as longitudes being between 80-55°W) contributes only $\sim 20-37\%$ of the total integrated zonal flux within the latitude band 68°S to 35°S during June-August, as shown in Table 2. This suggests that it is unlikely that any

|  | June | July | August |
|---|---|---|---|
| Southern Andes | 36.8% | 28.0% | 20.6% |
| Southern Ocean | 63.2% | 72.0% | 79.4% |
| Zonal Mean GWMF | -0.96 mPa | -0.84 mPa | -0.46 mPa |

**Table 2.** Percentage of total zonal momentum flux at $z = 40$ km in the latitude band $68°$S to $35°$S for June-August 2010. Negative flux values indicate a westward direction. The Southern Andes region (which includes the Drake Passage and the tip of the Antarctic Peninsula) is defined here as between longitudes of $80°$W-$55°$W, while the Southern Ocean region is defined as all longitudes outside of this, as shown by orange and grey dashed lines in Fig. 9c. In terms of surface area, the Southern Andes and Southern Ocean regions as defined here cover approximately 7% and 93% respectively of the total surface area in the latitude band $68°$S to $35°$S.

single source region accounts for the total integrated flux around $60°$S, and thus highlights the need to quantify the sources and relative flux contribution of non-orographic waves over the Southern Ocean.

The meridional fluxes in Figs. 8(d-f) and mean horizontal wavevectors in Figs. 8(g-i) also reveal an interesting picture. Large southward fluxes of up to 3 mPa are observed over the Southern Andes region, however large southward fluxes are also observed far out over the Southern Ocean at nearly all longitudes equatorward of around $60°$S. Poleward of around $60°$S, fluxes are predominantly northward from $60°$W to $180°$E. These results provide strong evidence of meridional "focusing" of gravity waves into the wintertime polar vortex from sources to the north and south. In particular, mean southward and northward fluxes of up to $\pm 2$ mPa are observed over the Southern Andes and Antarctic Peninsula respectively during August. This oblique propagation over the mountains was reported in simulated gravity-wave energy flux by Sato et al. (2012) and Alexander et al. (2016), observed during August 2010 in gravity-wave potential energy by Hindley et al. (2015), and also observed in meridional momentum flux and horizontal group velocity by Wright et al. (2017) during August 2014. These results suggest that this may be a persistent phenomenon that occurs annually.

Over the Southern Ocean, the convergence of horizontal wavevectors around $60°$S in panels (g-i) suggests that the oblique meridional propagation of waves into "belt" around $60°$S is a persistent feature at all longitudes. This suggests that this focusing effect applies to the majority of waves in the region and not just those from, for example, the Southern Andes or Antarctic Peninsula. These results provide additional evidence of a widespread oblique propagation effect, described in an increasing number of studies (Watanabe et al., 2008; Wu and Eckermann, 2008; Preusse et al., 2009; Sato et al., 2009; Ern et al., 2011; Sato et al., 2012; Kalisch et al., 2014; Hindley et al., 2015; Alexander et al., 2016; Ehard et al., 2017).

In particular, Wu and Eckermann (2008) used the relative difference between measured temperature variances in the ascending and descending nodes of Microwave Limb Sounder (MLS-Aura) measurements to show a zonal-mean convergence of wavevector directions around $60°$S over the Southern Ocean during June 2005 at altitudes around $z \approx 45$ km. Later, Holt et al. (2017) applied the S-transform method of Alexander and Barnet (2007) to 2-D AIRS brightness temperature measurements to show a similar convergence of wave vector directions around $60°$S over the same region during July 2005. Thus there is

a growing field of evidence that suggests that the widespread oblique propagation of waves into the southern wintertime polar vortex, an effect which is not generally considered by most operational parameterisation schemes in GCMs (Kalisch et al., 2014), could be a widespread and persistent phenomenon that should be considered in future schemes. This is discussed further in Sect. 6.

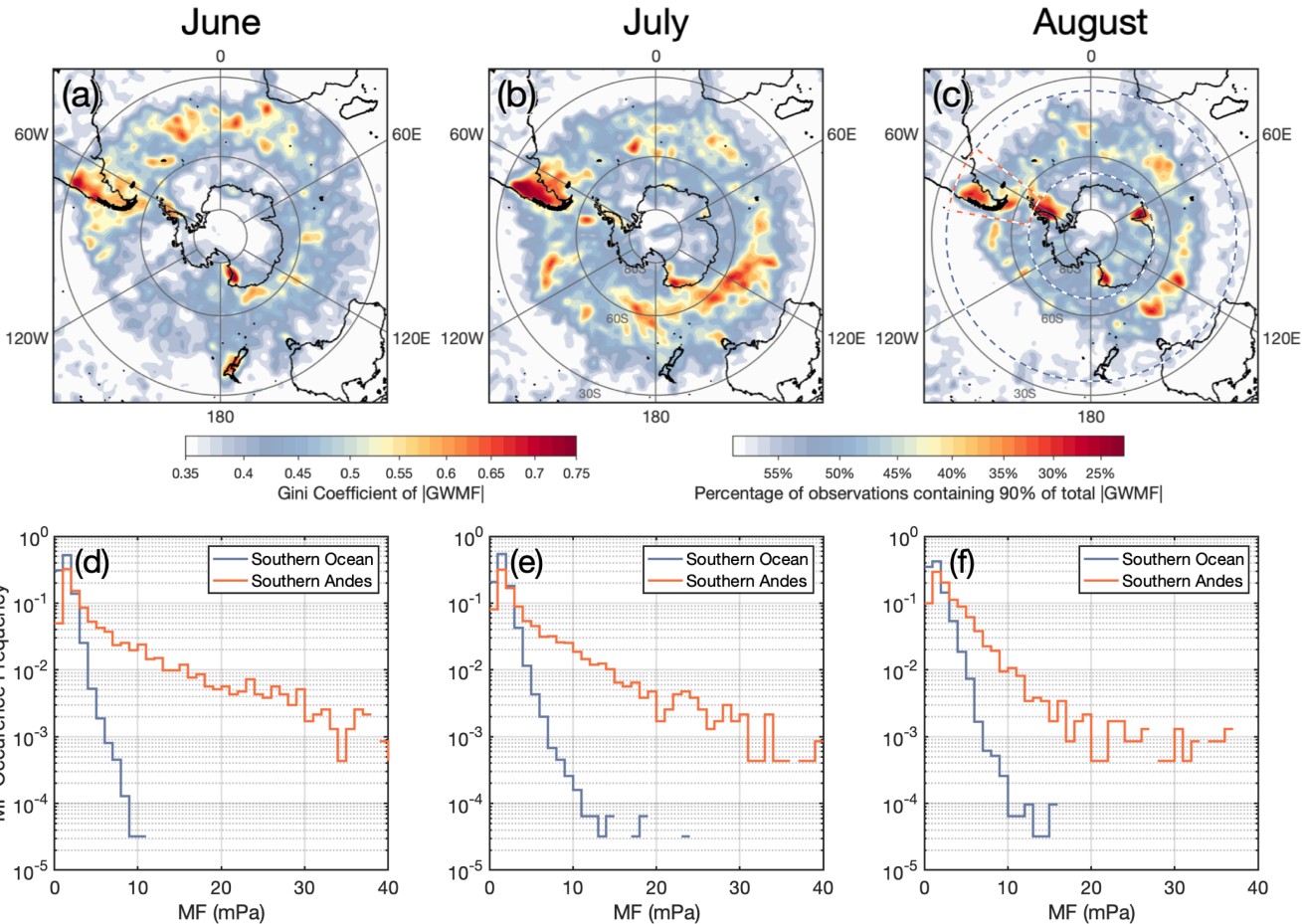

**Figure 9.** Panels (a,b,c) show the Gini coefficient of absolute gravity-wave momentum flux (GWMF) for each month of AIRS measurements at $z = 40$ km for June, July and August 2010. One colour bar shows the value of the Gini coefficient while the other shows the corresponding fraction of wave events attributable to 90% of the total monthly absolute GWMF. The same colour scale and limits are used such that both properties directly correspond. Orange and grey dashed lines in (c) show regions defined over the southern Andes and Southern Ocean, as used in Table 2. Probability density functions (PDFs) of GWMF at $z = 40$ km over these regions for the corresponding months are shown in panels (d,e,f).

## 5.4 Intermittency

So far in this study, we have presented our measured gravity-wave properties around the Southern Ocean as monthly-means. However, gravity waves are transient phenomena and their sources in the stratosphere can be highly variable on timescales from days to hours. Characterising the short-timescale variability (or intermittency) of gravity waves for different geographical regions and different conditions is key to developing more realistic parameterisations (Alexander et al., 2010).

A common metric that can be used to quantify the intermittency of gravity-wave activity is the Gini coefficient (Gini, 1912). The Gini coefficient is a scalar measure between zero and 1 of the unevenness of a distribution. For example, if a group of eight people divide a cake into eight pieces shared evenly between them, the Gini coefficient of the group is zero. If the entire cake is given to one lucky person, the Gini coefficient of the group is 1. In the context of gravity waves, by computing the Gini coefficient of gravity-wave momentum flux over a fixed time interval, we can determine whether the total measured flux was distributed evenly between many small-amplitude waves, or concentrated in a smaller number of large-amplitude waves events.

Previous studies (e.g. Hertzog et al., 2008, 2012; Plougonven et al., 2013; Wright et al., 2013, 2017) have used the Gini coefficient to characterise gravity-wave intermittency in a variety of data sets, such as observations made by satellites, super-pressure balloons, radiosondes and in models. Typically, orographic wave sources have been found to generally exhibit higher intermittency over long timescales than non-orographic sources (Plougonven et al., 2013; Wright et al., 2013).

Here, we compute the Gini coefficient $g_i$ of AIRS measurements of absolute gravity-wave momentum flux $|\text{MF}| = \sqrt{\text{MF}_x^2 + \text{MF}_y^2}$ in each grid cell of our regular distance grid over the southern hemisphere for June, July and August 2010 at an altitude of $z = 40\,\text{km}$, as shown in Fig. 9.

Several regions of high Gini coefficient values (i.e. high intermittency) are revealed. Intermittency is high ($g_i \gtrsim 0.7$) over the mountains of the Southern Andes and Antarctic Peninsula, particularly during July and August. A Gini coefficient of 0.7 corresponds to 90% of the total monthly momentum flux being attributed to less than 25% of all observed waves. By comparing these results to wave occurrence frequencies in Fig. 6 and measured wave amplitudes in Fig. 7, we find that although waves over these mountains seem to be very frequent, and they have (on average) relatively high amplitudes, most of the total monthly momentum flux is still attributed to relatively few, very large-amplitude wave events. Such events could be due to the variability in favourable large-scale wind conditions. Strong westerly winds blowing over the mountains at surface level, coupled with a smooth "conduit" of steadily increasing westerlies rising into the stratosphere that remains in place for several hours could allow a very intense mountain wave field to form, as discussed in Sect. 5.1. These results suggest that when these favourable conditions do occur, the intensity of the quasi-stationary mountain wave field that forms over the Southern Andes could be extremely high.

Regions of high intermittency are also observed over the Prince Charles Mountains and the Admiralty Mountains, located at the coast of the Antarctic continent at around 70°E and 170°E respectively. While some increased wave amplitudes and momentum fluxes were observed over these Antarctic mountain ranges in Figs. 7 and 8, these results suggest that, particularly during June and August, the total monthly flux was unevenly distributed into relatively few, high-flux wave events, something

that was hidden in the monthly-mean analysis. Furthermore, Fig. 6 indicates that waves were only observed in around 20-30% of observations over these mountains, suggesting that even fewer individual wave events are responsible for much of the total flux from these mountains during each month.

New Zealand is also located under a region of high intermittency during June. However, as with the Antarctic mountain ranges above, only relatively low wave occurrence frequencies ($\sim 20\%$) and monthly-mean wave amplitudes were measured over the region. This suggests that high-flux wave events over New Zealand are observed here in very rare but intense episodes, perhaps only during a few days each month. This is likely due to the relative likelihood of strong and favourable wind conditions that allow mountain waves to propagate into the stratosphere, or that refract such waves to the long wavelengths visible to AIRS. In contrast, intermittency was relatively low over the Kerguelen islands during June and July, despite larger wave occurrence frequencies ($\sim 45\%$) and significant momentum fluxes.

Over the Southern Ocean, intermittency is generally lower. This is consistent with previous studies (Hertzog et al., 2008; Plougonven et al., 2013; Jewtoukoff et al., 2015; Plougonven et al., 2017), but several localised regions of high intermittency are observed, particularly around the southern Atlantic and southern Indian Oceans during June and July-August respectively. These are likely attributable to individual high-flux wave events, such as the example shown in Fig. 1c. Given their location, it is likely that they correspond to intense non-orographic gravity-waves from sources such as storms, jets, fronts or spontaneous adjustments around the vortex edge.

An interesting observation in Fig. 9 is the relatively low intermittency observed over the island of Kerguelen in all three months. Normally, we would expect orographic sources to exhibit higher intermittency than non-orographic sources. Inspection of Fig. 6 indicates that Kerguelen lies under a region of relatively high wave occurrence, indicating that waves were observed over the island in more than half of all AIRS observations. This suggests that contrary to what we might expect from an isolated mountain-wave source, Kerguelen may be a remarkably persistent source of stratospheric gravity-waves during winter, within the AIRS observational filter.

Figures 9(d,e,f) show probability density functions (PDFs) of gravity-wave momentum flux at $z = 40\,\mathrm{km}$ for June-August 2010. PDFs for regions over the southern Andes and the Southern Ocean are shown by grey and orange lines respectively, using the same boundaries as described in Table 2, which is illustrated by the dashed grey and orange lines in Fig. 9c.

Increased intermittency is observed for the region over the southern Andes (orange line) than for the Southern Ocean (grey line), with an increased tendency towards large ($> 10\,\mathrm{mPa}$) but infrequent GWMF values during all three months. GWMF over the Southern Ocean is concentrated almost entirely at values below around $10\,\mathrm{mPa}$ during all three months, but the fraction of high GWMF values over the southern Andes reduces from June through to August as the distribution changes shape. This suggests a reduction in rare, large-amplitude mountain wave events towards the end of winter and into spring, likely due to a reduced frequency of favourable wind conditions over mountains that permit the upward propagation of these waves into the stratosphere.

The spatial distribution of our intermittency results are also in good general agreement with the results of Plougonven et al. (2013), who used the Gini coefficient to characterise intermittency of absolute momentum flux in stratospheric balloon observations from the Vorcore campaign around the Southern Ocean. They found increased regions of intermittency over the

mountains of the Antarctic Peninsula, with generally lower values over the Southern Ocean, although we note that the Vorcore campaign took place later in the year between October and December, whereas we focus on June, July and August here. A region of increased intermittency of momentum flux was also observed over the Southern Ocean by Wright et al. (2013) using limb-viewing satellite measurements, but they were unable to closely localise individual regions of increased intermittency.

## 6 Discussion

### 6.1 Oblique horizontal propagation of waves towards latitudes around 60°S

As discussed in Sect. 5.3, in Fig. 8(g-i) we showed evidence of a convergence of meridional components of horizontal wavevectors into the latitude band around 60°S. Interestingly, this oblique propagation phenomenon appears to occur at nearly all longtiudes around the Antarctic continent. Despite this, such oblique propagation is not generally considered in gravity wave parameterisation schemes for GCMs, which usually assume vertical-only propagation and deposition of horizontal momentum (Alexander et al., 2010; Kalisch et al., 2014). Our results suggest that the integrated zonal flux in models that is not being directed into the 60°S band by the lack of consideration of oblique propagation could be a significant factor in the missing wave flux in the region. This missing westward flux in GCMs could make the vortex too stable and thus less prone to break-up, hence an important factor for the cold-pole problem.

Kalisch et al. (2014) used the Gravity Wave Regional Or Global Ray Tracer (GROGRAT, Marks and Eckermann, 1995; Eckermann and Marks, 1997) to assess differences in resultant gravity wave drag distributions for two sets of gravity waves: one set allowed to propagate obliquely (GWO) and the other only allowed to propagate vertically (GWV). During austral winter, they found a poleward shift in peak deposition of zonal momentum in the southern hemisphere stratosphere for GWO waves compared to GWV waves. This result was coupled with a large increase in the poleward meridional drag that maximised over the location of the southern wintertime polar vortex, suggesting a meridional "focussing" effect in to the zonal maximum as observed in several other studies (Wu and Eckermann, 2008; Sato et al., 2012; Hindley et al., 2015; Wright et al., 2017). Their simulated results suggest that the widespread oblique propagation of these waves could result in a significant increase in wave drag around 60°S. Our observational results confirm that this widespread and persistent oblique propagation towards latitudes around 60°S does occur for the waves observed in AIRS measurements. As a result, the northward and southward meridional components of gravity wave momentum flux shown in Fig. 8 suggest that additional quantities of momentum could be transported latitudinally and deposited into mean flow closer to the centre of the stratospheric polar vortex around 60°S. This could be an important element of the solution to the missing momentum around 60°S.

### 6.2 The role of small islands

A persistent region of higher wave amplitudes, shorter horizontal wavelengths and longer vertical wavelengths is observed directly over and slightly downwind of the Kerguelen and Heard Islands during all months in Fig. 7. This corresponds to significant westward ($\gtrsim 2$ mPa) and southward ($\sim$0.5-1 mPa) momentum fluxes measured directly over the islands in Fig. 8.

These features are highly indicative of a strong and persistent mountain wave field over the islands, and the fact that they are visible even in a monthly mean is quite striking, suggesting these islands could be very significant wave sources in the region. This is in line with the results of previous studies, such as Alexander and Grimsdell (2013); Hoffmann et al. (2016). The relatively low flux intermittency observed over the Kerguelen Islands in Fig. 9 further suggests that mountain wave fields over these islands are persistent and/or relatively frequently occurring.

Enhanced wave amplitudes and fluxes are also observed over South Georgia island in Figs. 7 and 8, together with a tendency to a northward-southward divide in meridional momentum flux directly over the island during July in Fig. 8e, possibly suggesting a persistence of chevron-shaped wave fields similar to the case study shown in Fig. 5. While Hoffmann et al. (2013) identified the Kerguelen Islands and South Georgia as wave sources during austral winter, this is the first time their relative contribution to monthly mean momentum fluxes has been shown in the context of other sources.

The contribution of small islands to the "missing momentum" problem during winter around the latitude belt of $60°$S could be significant (McLandress et al., 2012). Garfinkel and Oman (2018) found that, for several GCMs, springtime stratospheric temperature biases could be reduced by up to $30 - 50\%$ by increasing the topographic variance from the islands by a factor of 5. They also found however that further increases did not lead to much more significant improvement, suggesting that fluxes from small islands may only form part of the solution.

## 6.3 Possible regions of downwardly-propagating waves

Even with these new 3-D observations, there remains an ambiguity between waves travelling "upwards and forwards" or "downwards and backwards", which cannot be resolved without time-varying measurements at shorter time intervals than successive AIRS overpasses, or a priori information such as supplementary wind fields from reanalyses (Alexander et al., 2009). In this study, we collapse this ambiguity by only analysing for negative vertical wavenumbers (i.e. $m < 0$) in the 3DST. This makes the assumption that all measured waves are propagating upwards.

We suspect that the majority of gravity waves detected around $z = 40$ km in our region of study during winter are indeed propagating upwards, although we acknowledge that we may also be observing a significant portion of downwardly propagating waves (e.g. Kaifler et al., 2017). Using stratospheric lidar measurements at McMurdo, Antarctica, Zhao et al. (2017) showed that upwardly propagating waves accounted for 70% of identified waves during winter, with a tendency towards longer vertical wavelengths, which might increase their likelihood of detection by AIRS.

Using the difference in variance between the eastward and westward parts of 2-D AIRS radiance measurements in several channels, Gong et al. (2012) were able to show that stratospheric gravity waves were overwhelmingly westward-inclined during July 2005 over the Southern Ocean. They implied that this must indicate westward and upward propagation, since if they were propagating downwards and eastwards they would almost certainly have been filtered out by strong zonal winds in the stratospheric polar night jet.

Therefore, if the majority of waves observed at around $z = 40$ km around $60°$S during winter are travelling westward (that is, $k < 0$) as found by Gong et al. (2012), and the assumption of upward propagation ($m < 0$) in our study was wrong to a significant degree, we would expect to see regions of net zero or eastward propagation in our results in Fig. 8(g-i). This is

because by attributing the wrong sign to $m$, we would reverse the sign of $k$ and $l$ (e.g. Wright et al., 2016). The fact that we do not observe large regions of net zero or eastward propagation in our results around $60°$S (see Fig. 8), where the vast majority of waves were detected (see Fig. 6), suggests that our assumption of upward propagation is largely valid. Equatorward of around $30°$S in Fig. 8(g-i), we observe essentially random mean wave directions where there are very few wave detections above our
noise threshold.

In a future study, it could be interesting to instead use supplementary background wind information to collapse the ambiguity around upwards or downwards propagation. By assuming that waves always propagate against the background flow, we could constrain the relative fractions of upwardly and downwardly propagating waves. In this study however, we wish to focus on accurately characterising the horizontal directionality of measured waves, so we compromise on an "upwards-downwards"
ambiguity to achieve this.

## 6.4   Comparison with resolved waves in GCMs

Preusse et al. (2014) used ray-tracing of resolved waves in ECMWF operational analyses to determine the lowest traceable altitude (LTA) of stratospheric gravity waves observed around the southern wintertime polar vortex. They found that waves measured at $z = 25$ km were traceable all the way back to the ground (LTA $\approx 0$ km) over the Southern Andes and Antarctic
Peninsula during June-August (their Fig. 5). This was strongly indicative of an orographic origin. Out over the Southern Ocean however, waves were only typically traced back to altitudes around $z = 5 - 15$ km, indicative of non-orographic processes. They found reasonable correlations between convective precipitation and higher LTAs, though the two were not always geographically co-located with each other. They suggested this could be due to the oblique propagation of gravity waves from their sources to their LTA position, which implies the meridional focusing of gravity waves into the latitude band of $60°$S during
austral winter. Our observational results in Fig. 8(g-i) would seem to support this hypothesis, although our results go further to suggest that this oblique focusing also occurs at altitudes around $z = 40$ km, which is significantly higher than the altitudes considered in Preusse et al. (2014).

Three other interesting features of Preusse et al. (2014, their Fig. 5) have relevance here. Particularly during August, they showed that the vast majority of resolved waves around the coast of Antarctica clockwise from around $60°$E to $60°$W had LTAs
close to zero, strongly suggesting orographic sources. This could imply that regions of zonal and meridional flux we observe around the Antarctic coast in Fig. 8(a-f) could be orographic in origin. Indeed, Hoffmann et al. (2013, their Figs. 6 and 7) identified several regions around the eastern coast of Antarctica as wave hot spots. Since it is such a large area, the integrated flux over this region with a strong northward component directed towards $60°$S could present a significant contribution to the "missing momentum" at these latitudes. This is interesting as the extended coastal regions of Antarctica have not generally
been considered as likely candidates for this.

Additionally, there seems to be very little evidence of waves being traced back to low LTAs over small islands in Preusse et al. (2014). This is almost certainly due to these islands being subgrid-scale, and thus their flux contributions will not be present. In our AIRS analysis, the fluxes from these small islands are resolvable, and we can see significant effects even on

monthly timescales in Fig. 8(a-f). Backward ray-tracing of our 3-D wave observations in future studies may be able to confirm to what extent small islands are the sources of the observed waves over open ocean, compared to other processes.

Finally, over the Southern Ocean between longitudes of roughly 45°W clockwise to 15°E, Preusse et al. (2014) showed regions with very high LTAs of 10 to 15 km (suggesting a non-orographic source) but with relatively little precipitation. In our Figs. 6 and 7, we observe increased wave occurrence frequencies and amplitudes extending eastwards out over this region and away from the southern Andes, particularly during June. Since mountain-wave structures generally form directly over the mountains and do not usually extend so far downwind, it could be that some of the observed gravity-wave activity in this immediate downwind region in our results and those of Preusse et al. (2014) could be the result of a broad spectrum of secondary gravity waves. These secondary waves, which can have a variety of phase speeds and directions, could be generated as these very large mountain waves ascend and break over the mountains as they encounter critical wind layers in the middle and upper stratosphere (Woods and Smith, 2010; Bossert et al., 2017; Vadas et al., 2018; Vadas and Becker, 2018). It is however very difficult to distinguish between different non-orographic sources here, so we are not able to speculate further.

Further eastwards of the prime meridian around the Antarctic continent, we expect that waves from non-orographic sources such as storms, jets and fronts (Plougonven and Zhang, 2014) and spontaneous adjustment processes around the polar night jet are likely to become more dominant. Future work involving ray-tracing these observed waves back to their sources will help to constrain the relative contributions of these non-orographic wave sources, in addition to the role of orographic waves from small islands and oblique propagation from continental mountain ranges in the region.

## 7    Summary and Conclusions

In this study we have investigated stratospheric gravity waves and their momentum fluxes at the crucial latitudes near 60°S, where nearly all general circulation models significantly under-represent these fluxes in parameterisations. These gravity-wave measurements involved the first extended application of a 3-D Stockwell Transform (3DST) method to measure the amplitudes, wavelengths, fluxes and intermittency of gravity waves in high-resolution 3-D satellite observations over the Southern Ocean. We have:

1. Developed a 3DST method to measure localised amplitudes, spectral characteristics and directions of wave packets in 3-D data sets. We have tested and validated our 3DST method on synthetic wave packets and developed a new method to improve the wave amplitude measurement for localised wave packets in higher-dimensional S-transforms.

2. Applied the 3DST method to 3-D AIRS satellite observations of real gravity waves in two new case studies: one over the Southern Andes and Antarctic Peninsula and another over the isolated mountainous island of South Georgia in the Southern Ocean.

3. Applied the 3DST method to 3-D gravity-wave measurements in the stratosphere over the entire Southern Ocean during June-August 2010 to produce the first 3-D satellite wintertime study of gravity-wave amplitudes, wavelengths, directional momentum fluxes and intermittency at latitudes near 60°S.

We find that monthly wave occurrence frequencies are highest over the Southern Andes ($\sim 80\%$) during June 2010 with average wave amplitudes exceeding 4 K over the region. Additional increased wave occurrence frequencies are found over the Antarctic Peninsula ($\sim 40\%$) and the islands of Kerguelen ($\sim 50\%$), Heard ($\sim 45\%$) and South Georgia ($\sim 40\%$). An extended region of increased wave occurrence frequency ($\sim 20-30\%$) and wave amplitude ($1-1.5$ K) extends in a long arc from the Southern Andes eastward to longitudes sometimes further than $180°$E during June and July. During August, a more zonally-symmetric "belt" of gravity wave activity is observed at latitudes around $60°$S. Increased wave activity over this large oceanic region is likely to correspond to non-orographic processes such as storms, jets, fronts and spontaneous (geostrophic) adjustment around the edge of the stratospheric polar vortex.

Monthly-mean zonal momentum fluxes are westward and largest (several tens of mPa) over the southernmost part of South America, with values exceeding 16 mPa during June. Over the Southern Ocean, monthly-mean fluxes are closer to around $1-2$ mPa. However, the much larger area of the Southern Ocean means that it contains between $65-80\%$ of the total zonal flux in the latitude band. This indicates that non-orographic waves over the ocean make a very significant contribution to the total zonal flux at these latitudes.

Measured fluxes over several regions at southern high latitudes are also found to be highly intermittent. Over the Southern Andes we find that around 90% of the total monthly momentum flux is attributable to less than 25% of wave events, despite high wave occurrence frequencies and monthly-mean wave amplitudes. Over New Zealand, the Antarctic Peninsula, the Prince Charles Mountains and the Admiralty Mountains on the coast of Antarctica we find that around 90% of the total monthly momentum flux is attributable to around $25-45\%$ of wave events during June and August. Over the Southern Ocean, some isolated regions also exhibit intermittency that is much higher than expected, which suggests that significant parts of the total non-orographic flux also could be attributable to relatively rare but intense wave events.

Finally, analysis of the propagation directions of measured horizontal wavevectors and monthly-mean meridional momentum fluxes reveal a widespread convergence of gravity waves towards latitudes near $60°$S during winter. This indicates significant additional propagation of momentum flux into this region, and is observed at almost all longitudes. We conclude that this additional flux, originating from sources to the north and south including non-orographic wave sources over the Southern Ocean and orographic sources in mountainous Antarctic coastal regions, could make a significant contribution to the "missing" flux around $60°$S reported in GCMs (e.g. McLandress et al., 2012).

These results highlight the powerful insights into the characteristics of stratospheric gravity waves over the Southern Ocean near $60°$S that are provided by the 3DST analysis of 3-D satellite observations presented here. With these measurements we can guide development in GCMs to reduce biases and improve predictability over longer timescales, ultimately contributing to better forecasts of weather and climate.

## Appendix A: Quantifying the amplitude attenuation of specified wave packets in higher-dimensional Stockwell transforms

In Sect. 3.1 the effect of the amplitude attenuation effect for the amplitudes of localised wave packets in the S-transform was shown and discussed. To our knowledge, this attenuation effect is not something that is usually considered in applications of the S-transform in the geosciences. This is likely due to the effect being largely negligible for the 1DST, the most common application of the S-transform. In Sect. 3.4.1 we described and applied a "composite" S-transform method to overcome this problem by localising and measuring wave amplitude indiscriminately for a specified range of frequencies, thus dramatically reducing wave amplitude attenuation for any given frequency.

However, as more studies start to consider multi-dimensional data sets, it is useful to try to quantify the attenuation effect for wave packets in higher-dimensional S-transforms. Here, we make an estimate of the expected amplitude attenuation for different wave packets of varying sizes and orientations in the $N$-dimensional S-transform. To do this, we define an $N$-dimensional wave packet then compute its $N$-dimensional S-transform. What we want to see is how variables relating to the size and shape of the wave packet propagate through the S-transform function, and how they might affect the measured amplitude as a result. We also seek to demonstrate that the attenuation effect is inherent to the S-transform itself, and not an artefact of our application.

### A1 A generalised expression for the amplitude attenuation of Gaussian wave packets in the S-transform

We begin by defining an $N$-dimensional wave packet $h(\boldsymbol{x})$, which consists of a cosinusoidal wave $\cos(\boldsymbol{k}^{\top}\boldsymbol{x}+\theta)$ with angular wavenumber $\boldsymbol{k}$, amplitude $a$ and phase $\theta$, enclosed within a Gaussian amplitude envelope. Here, $\boldsymbol{x}=(x_1,x_2,\cdots,x_N)$ and $\boldsymbol{k}=(k_1,k_2\cdots k_N)$ are column vectors describing an $N$-dimensional spatial coordinate system and wavenumbers in that coordinate system respectively. As shown in Eqn. 5, the $N$-D wave packet $h(\boldsymbol{x})$ is thus given by:

$$h(\boldsymbol{x}) = a\cos(\boldsymbol{k}^{\top}\boldsymbol{x}+\theta)\prod_{n=1}^{N} e^{-\frac{\frac{1}{2}(x_n-\tau_n)^2}{s_n^2}}, \tag{A1}$$

where $s_n = (s_1, s_2 \cdots s_N)$ is the standard deviation of the Gaussian envelope in each dimension.

To compute the amplitude attenuation on the input amplitude $a$, we take the $N$-dimensional Stockwell transform of our wave packet. Here we take the Fourier transform of $h(\boldsymbol{x})$ and apply the spectral-domain form of the NDST (Stockwell et al., 1996, their Eqn. 9), not the spatial-domain form in Eqn. 1.

The S-transform, in very general terms, applies Gaussian windows to the analytic signal of the input data, which allows it to make measurements of the absolute wave amplitude at each location in $\boldsymbol{x}$. Following the method of Stockwell et al. (1996); Stockwell (2007), the magnitude of the analytic signal can be found by considering only one of the complex conjugate parts of $h(\boldsymbol{x})$ and doubling the result. This means we need only analyse one of the complex conjugate parts here, so we split $h(\boldsymbol{x})$ into its complex conjugate parts as

$$h(\boldsymbol{x}) = \frac{a}{2}\left(e^{i(\boldsymbol{k}^{\top}\boldsymbol{x}+\theta)} + e^{-i(\boldsymbol{k}^{\top}\boldsymbol{x}+\theta)}\right)\prod_{n=1}^{N} e^{-\frac{\frac{1}{2}(x_n-\tau_n)^2}{s_n^2}} \tag{A2}$$

and consider only one part $h^+(\boldsymbol{x})$, given by

$$h^+(\boldsymbol{x}) = \frac{a\,e^{i\theta}}{2}\left(e^{i\boldsymbol{k}^\top\boldsymbol{x}}\right)\prod_{n=1}^{N}e^{-\frac{\frac{1}{2}(x_n-\tau_n)^2}{s_n^2}} \tag{A3}$$

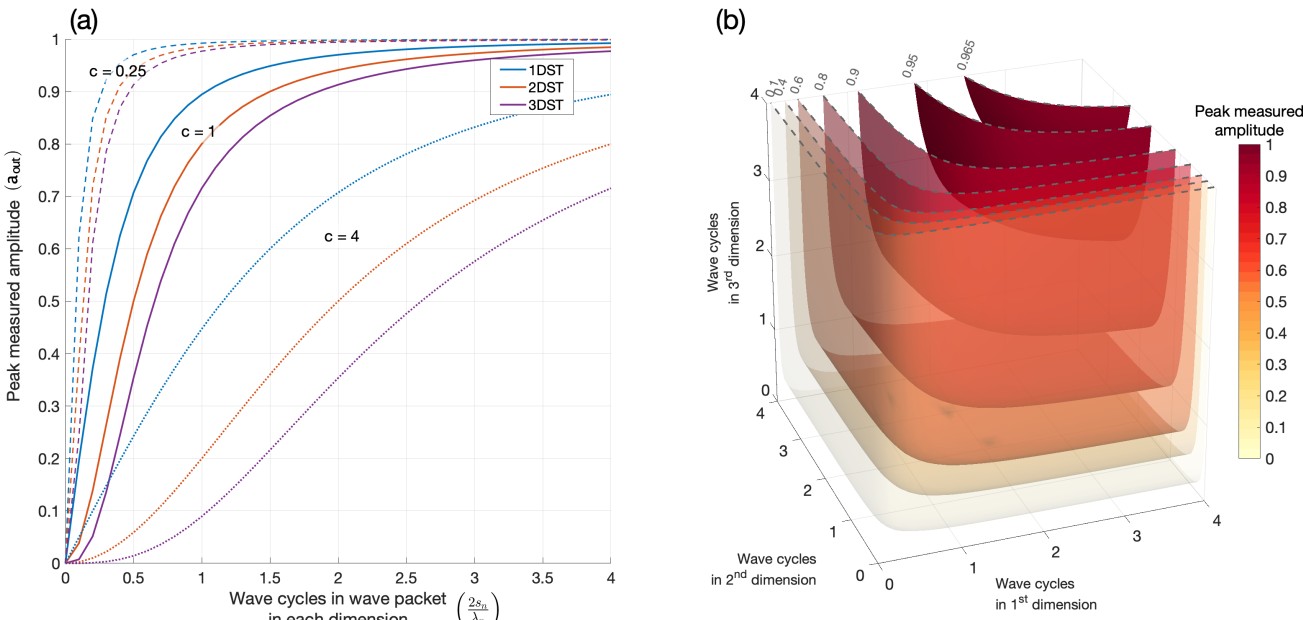

**Figure A1.** (a) Theoretical peak measurable amplitude $a_{\mathrm{out}}$ in each dimension $n$ for an $N$-dimensional Gaussian wave packet with peak amplitude $a = 1$, wavelength $\lambda_n$ and standard deviation $s_n$ in the 1-, 2- and 3-dimensional Stockwell transforms (with scale factor $c_n$). The $x$-axis shows the number of wavecycles per two standard deviations $\beta_n = \frac{2s_n}{\lambda_n}$ in each dimension for any given wave packet. Panel (b) shows the peak measured amplitude for wave packets as isosurfaces in 3-D for the 3DST with $c_n = 0.25$. To summarise, the more wave cycles in a wave packet in each dimension, the better the amplitude measurement. Adjusting the scaling parameter $c_n$ can also improve results.

We next find an expression for the Fourier transform $H^+(\boldsymbol{\alpha})$ of our wave packet, where $\boldsymbol{\alpha} = (\alpha_1, \alpha_2 \cdots \alpha_N)$ is a column vector denoting wavenumbers in the Fourier domain for each dimension $n$:

5  $$H^+(\boldsymbol{\alpha}) = \int\limits_{-\infty}^{+\infty} h^+(\boldsymbol{x})\,e^{-i\boldsymbol{\alpha}^\top\boldsymbol{x}}\,d\boldsymbol{x}$$

$$= \frac{a\,e^{i\theta}}{2}\prod_{n=1}^{N}\int\limits_{-\infty}^{+\infty}e^{-\frac{\frac{1}{2}(x_n-\tau_n)^2}{s_n^2}}\,e^{i(k_n-\alpha_n)x_n}\,dx_n \tag{A4}$$

We then let $y_n = (x_n - \tau_n)$ and $dy_n = dx_n$ to give

$$H^+(\boldsymbol{\alpha}) = \frac{a\,e^{i\theta}}{2} \prod_{n=1}^{N} \int_{-\infty}^{+\infty} e^{-\frac{1}{2}\frac{y_n^2}{s_n^2}} e^{i(k_n - \alpha_n)y_n} e^{i(k_n-\alpha_n)\tau_n}\,dy_n$$

$$= \frac{a\,e^{i\theta}}{2} \prod_{n=1}^{N} \int_{-\infty}^{+\infty} e^{-\frac{1}{2}\left(\frac{y_n}{s_n} - i(k_n-\alpha_n)s_n\right)^2} e^{-\frac{1}{2}(k_n-\alpha_n)^2 s_n^2} e^{i(k_n-\alpha_n)\tau_n}\,dy_n \tag{A5}$$

which we simplify further by letting $v_n = \left(\frac{y_n}{s_n} - i(k_n - \alpha_n)s_n\right)$ and $dv_n = \frac{dy_n}{s_n}$ to give

$$H^+(\boldsymbol{\alpha}) = \frac{a\,e^{i\theta}}{2} \prod_{n=1}^{N} \int_{-\infty}^{+\infty} s_n e^{-\frac{1}{2}v_n^2}\,dv_n\, e^{-\frac{1}{2}(k_n-\alpha_n)^2 s_n^2} e^{i(k_n-\alpha_n)\tau_n}$$

$$= \frac{a\,e^{i\theta}}{2}(2\pi)^{\frac{N}{2}} \prod_{n=1}^{N} s_n\, e^{-\frac{1}{2}(k_n-\alpha_n)^2 s_n^2} e^{i(k_n-\alpha_n)\tau_n} \tag{A6}$$

We now compute the $N$-dimensional Stockwell transform $S^+(\hat{\boldsymbol{\tau}}, \hat{\boldsymbol{k}})$ for this complex conjugate part, using the Fourier domain definition in Stockwell et al. (1996), given as

$$S^+(\hat{\boldsymbol{\tau}}, \hat{\boldsymbol{k}}) = \frac{1}{(2\pi)^N} \int_{-\infty}^{+\infty} H^+(\boldsymbol{\alpha}) W(\boldsymbol{\alpha}; \hat{\boldsymbol{k}}, \boldsymbol{\sigma}) e^{i\boldsymbol{\alpha}^\top \hat{\boldsymbol{\tau}}}\,d\boldsymbol{\alpha} \tag{A7}$$

where

$$W(\boldsymbol{\alpha}; \hat{\boldsymbol{k}}, \boldsymbol{\sigma}) = \prod_{n=1}^{N} e^{\frac{-\frac{1}{2}(\alpha_n - \hat{k}_n)^2}{\sigma_n^2}} \tag{A8}$$

is the Gaussian apodizing function in the Fourier domain $\boldsymbol{\alpha}$ with standard deviation $\sigma_n = \frac{k_n}{c_n}$ in each dimension $n$ and $c_n$ is the scaling parameter discussed in Sect. 3.1. Often referred to as the "voice Gaussian", this function is scaled with wavenumber $k_n$ to produce the specific spatial-spectral localisation capabilities of the S-transform (Stockwell et al., 1996; Stockwell, 2007).

Here, $\hat{\boldsymbol{k}}$ would normally denote a range of wavenumbers in each dimension, but since we wish to evaluate the S-transform for only the specific wavenumber of our input wave packet to compute the amplitude attenuation, we can simplify Eqn. A7 by setting $\hat{\boldsymbol{k}} = \boldsymbol{k}$. Likewise, we set $\hat{\boldsymbol{\tau}} = \boldsymbol{\tau}$ to evaluate the S-transform at the centre of the wave packet envelope in Eqn. A1, where the magnitude of the S-transform is maximised and should be the closest to the input amplitude $a$. Inserting $H^+(\boldsymbol{\alpha})$ from Eqn. A6 then yields

$$S^+(\boldsymbol{\tau}, \boldsymbol{k}) = \frac{1}{(2\pi)^N} \int_{-\infty}^{+\infty} \left[ \frac{a\,e^{i\theta}}{2}(2\pi)^{\frac{N}{2}} \prod_{n=1}^{N} s_n e^{-\frac{1}{2}(k_n-\alpha_n)^2 s_n^2} e^{i(k_n-\alpha_n)\tau_n} \right] e^{\frac{-\frac{1}{2}(\alpha_n - k_n)^2}{\sigma_n^2}} e^{i\alpha_n \tau_n}\,d\boldsymbol{\alpha} \tag{A9}$$

which we can simplify as

$$
\begin{aligned}
S^+(\boldsymbol{\tau}, \boldsymbol{k}) &= \frac{a\, e^{i\theta}}{2(2\pi)^{\frac{N}{2}}} \prod_{n=1}^{N} s_n\, e^{ik_n\tau_n} \int\limits_{-\infty}^{+\infty} e^{-\frac{1}{2}(k_n - \alpha_n)^2 \left( s_n^2 + \frac{1}{\sigma_n^2} \right)}\, d\alpha_n \\
&= \frac{a\, e^{i\theta}}{2(2\pi)^{\frac{N}{2}}} \prod_{n=1}^{N} s_n\, e^{ik_n\tau_n} \left( \frac{2\pi\sigma_n^2}{s_n^2\sigma_n^2 + 1} \right)^{\frac{1}{2}} \\
&= \frac{a}{2} \prod_{n=1}^{N} \frac{s_n\sigma_n}{\sqrt{s_n^2\sigma_n^2 + 1}}\, e^{i(k_n\tau_n + \theta)}
\end{aligned}
\tag{A10}
$$

Noting that the relevant part of the complex conjugate pair in Eqn. A2 contains half of total spectral energy, we can double the magnitude of our result in Eqn. A10 to recover the peak measured amplitude $a_{\text{out}}$ as

$$
a_{\text{out}} = 2 \left| S^+(\boldsymbol{\tau}, \boldsymbol{k}) \right| = a \prod_{n=1}^{N} \frac{s_n\sigma_n}{\sqrt{s_n^2\sigma_n^2 + 1}}
\tag{A11}
$$

If we recall that $\sigma_n = \frac{|f_n|}{c_n} = \frac{1}{\lambda_n c_n}$ is the standard deviation of the "voice Gaussian" for spatial frequency $f_n$ and wavelength $\lambda_n$, we can rearrange Eqn. A11 for wavelength as

$$
\frac{a_{\text{out}}}{a} = \prod_{n=1}^{N} \frac{s_n}{\sqrt{s_n^2 + c_n^2\lambda_n^2}}
\tag{A12}
$$

we arrive at the relation we specified in Eqn. 6.

## A2   Effective amplitude attenuation for wave packets of varying sizes

Now that we have an estimate of the amplitude attenuation that might be expected for Gaussian wave packets in the S-transform, we can consider what the effects of this attenuation might be for higher dimensional S-transforms.

We can see from Eqn. A12 that the key factor in estimating the attenuation for a given wave packet is the number of observable wave cycles in each dimension. Here, for Gaussian wave packets, we can define this is as the ratio between the wavelength $\lambda_n$ and two standard deviations $2s_n$ in each dimension. If we consider the number of observable wave cycles in each dimension of an $N$-dimensional, unit amplitude, Gaussian wave packet to be $\beta_n = \frac{2s_n}{\lambda_n}$, that is, the number of wavelengths within two standard deviations, we can investigate the likely amplitude attenuation for the 1-, 2- and 3-dimensional S-transforms using the relation from Eqn. A12 as

$$
a_{\text{out}} = a \prod_{n=1}^{N} \frac{\beta_n}{\sqrt{\beta_n^2 + 4c_n^2}}
\tag{A13}
$$

Figure A1a shows the peak measured amplitude $a_{\text{out}}$ for Gaussian wave packets with varying $\beta_n$ in each dimension. Here we consider different scaling parameter values of $c_n$, namely 0.25, 1 and 4.

We find that for the 1-dimensional S-transform (blue curve), only 1 wave cycle per two standard deviations is required to measure amplitudes to within 90% of their input value for $c_n = 1$. Decreasing the scaling parameter $c_n$ can reduce this to

around 0.25 wave cycles, but this is at the expense of some spectral localisation, as discussed in Sect. 3.1. Increasing $c_n$ for improved spectral localisation results in much poorer amplitude measurement and localisation.

For higher dimensional S-transforms, the results are slightly more complicated. For example, if a wave packet had $\beta_1 = 1$ (that is, one wave cycle per two standard deviations) in one dimension, but only $\beta_2 = 0.5$ in the second dimension, the peak measurable amplitude for $c_n = 1$ would be $0.9 \times 0.7 \approx 0.63$ of the input value. If the wave packet also had a third dimension, with $\beta_3 = 0.75$, the peak measurable amplitude would be $0.9 \times 0.7 \times 0.85 \approx 0.535$ for $c_n = 1$. The curves shown on Fig. A1a for the 2DST (orange) and 3DST (purple) are if the wave packet was symmetrical in all dimensions, with the same number of wave cycles per two standard deviations in each.

Fig. A1b shows the expected peak measured amplitude $a_{\mathrm{out}}$ on isosurfaces for different numbers of wave cycles in each dimension for the 3DST using $c_n = 0.25$ for all dimensions. This is what we apply in this study. If more than 2 wave cycles are observable in each of the three dimensions, the peak measurable amplitude is over 90%. However, if any of these dimensions have lower values for $\frac{2s_n}{\lambda_n}$, the peak measurable amplitude falls away quite quickly. We observed this for some of our synthetic wave packets in Sect. 3.5. When wave packets were aligned strongly perpendicular to any particular axis, the wavelength became very long in that dimension, meaning $\frac{2s_n}{\lambda_n}$ was reduced and more attenuation was observed.

For the AIRS measurements used in this study, this effect can be quite significant. In the horizontal, we can usually expect at least several wave cycles to exist within a wave packet due to the relatively large physical width of the AIRS granules ($\sim 1800 \times 2400\,\mathrm{km}$) compared to the typical horizontal scale of most mid-frequency gravity waves in the atmosphere.

In the vertical however, our measurements only extend between $z = 10 - 70\,\mathrm{km}$, of which only the central $20 - 60\,\mathrm{km}$ are reliable. We also find average vertical wavelengths to be around $20\,\mathrm{km}$ from Fig. 7. As a result, wave packets will experience significant attenuation from this dimension, since realistically we are likely to observe far fewer wave cycles in the vertical. If we take the vertical to be the $3^{\mathrm{rd}}$ dimension, we can see that we are limited only to the lower section of the isosurfaces in Fig. A1b, below around 1-2 wave cycles, where peak measurable amplitudes are often below 60%.

If the precise nature of the wave field is known, that is, the exact form of any wave packet at any location in the input data is exactly defined, the attenuation could be estimated and corrected for by deriving an expression in the way that we did above. However, since we do not know the precise nature of the wave field (which is why we apply the S-transform in the first place), this is sadly not possible. In practice therefore, this highlights the need for the amplitude attenuation to be mitigated by applying new methods such as the "composite" approach we described in Sect. 3.4.1.

**Appendix B: Reducing the impact of AIRS retrieval noise**

In this study, several approaches have been applied to mitigate the unwanted effects of AIRS retrieval noise from our gravity-wave measurements. We acknowledge of course that some may propagate through, but this is the case with any study using observational measurements.

We first performed an analysis of the expected retrieval noise in the AIRS temperature measurements under various atmospheric conditions in Section 2.2 and Fig. 2a. Based on these results, we selected a noise threshold for wave amplitude of $1.5\,\mathrm{K}$

for all of our absolute measured wave amplitudes. To provide context for this value, Fig. B1 shows a histogram of all detrended AIRS temperature perturbations globally along the AIRS scan track at an altitude of $z = 40$ km during the month of June 2010.

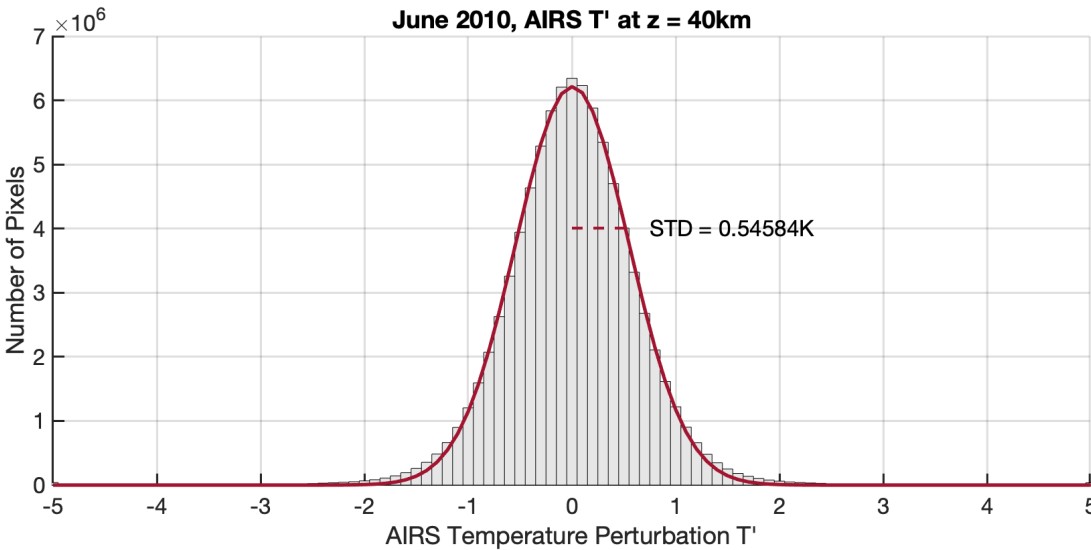

**Figure B1.** Histogram of all detrended AIRS temperature perturbations at z = 40 km globally during June 2010. A Gaussian fit to this distribution is shown by the red line, which has a standard deviation of around 0.55 K as shown.

The distribution of AIRS temperature perturbations falls as a rough Gaussian distribution centred around zero. A fitted Gaussian function for the distribution in Fig. B1 yields a standard deviation of around $\sigma = 0.55$ K. Our threshold of 1.5 K for 3DST-measured wave amplitudes, estimated from the values shown in Fig. 2a, corresponds to a cut-off of nearly $3\sigma$, well over the 99th percentile. This shows that our chosen threshold is a good cut-off for large portions of temperature perturbations that could be indistinguishable from noise.

A further consideration is that, generally speaking, retrieval noise may manifest as uncorrelated "speckle" noise in our measurements, as can be seen in regions of AIRS measurements without a clear gravity-wave signal in the examples in Fig. 1. This means that we should also be able to isolate and exclude this noise spectrally in the 3DST by not analysing for these very high frequencies that correspond to unwanted speckle noise.

Several steps are followed in this study to reduce the impact of such small-scale speckle noise by excluding it from our gravity-wave results. These steps are listed below and illustrated in Fig. B2, using the examples shown in Fig. 1.

We begin with the detrended and regridded AIRS temperature measurements along the AIRS scan track, as shown in the first column of Fig. B2(a-c) at an altitude of $z = 40$ km. The following steps are then applied:

1. A horizontal 3×3-pixel boxcar smoothing filter is applied to each vertical level of AIRS measurements, as shown in Figs. B2(d-f). This helps to suppress small-scale pixel-to-pixel variations that are unlikely to be physical or reliable (see Hindley et al., 2016, their Sect. 5.4).

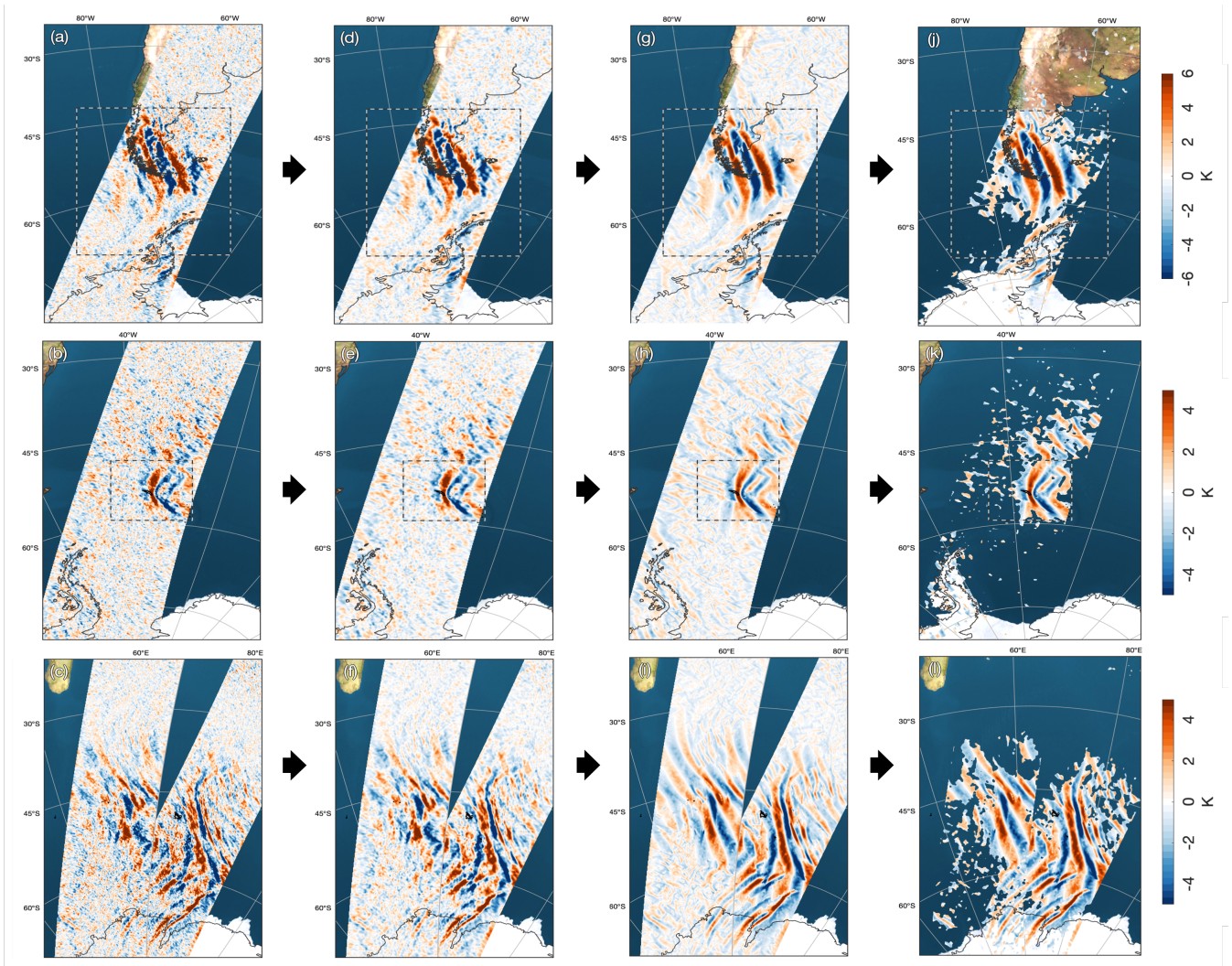

**Figure B2.** As Fig. 1, but showing our step-by-step approach to mitigating the effect of AIRS retrieval noise in our 3DST gravity-wave measurements. The first column (a-c) shows the original, detrended AIRS measurements at z = 40 km. The second column (d-f) shows the effect of the 3x3 horizontal boxcar smoothing. The third column (g-i) shows the real part of the complex 3DST output, which produces a reconstruction of the input data "as the 3DST sees it", using our tuning settings to ignore small-scale noise features with horizontal wavelengths $\lambda_H < 40$ km. Finally, the fourth column (j-l) is as the third column, except that regions where the absolute 3DST-measured wave amplitude is less than 1.5 K have been removed. It is only these remaining regions that are considered in our momentum flux, wavelength, intermittency and wavevector direction results in the main body of the paper.

2. We the only analyse the AIRS measurements for horizontal wavelengths $\lambda_H > 40$ km in the 3DST. This functionality of our 3DST application specifically excludes small-scale speckle noise features spectrally. Figures B2(g-i) shows the real part of our "collapsed" S-transform object $\Re[\mathcal{A}(x,y,z)]$ (see Sect. 3.2). This can be considered as a "reconstruction" of

the input data as it appears only within the selected spectral bounds that we have chosen for analysis in the 3DST. Close inspection of Figs. B2(g-i) shows that small-scale speckle noise is significantly reduced, and in its place only underlying features with larger horizontal wavelengths have been analysed. This output is a useful diagnostic tool for tuning the functionality of the 3DST method to the spectral characteristics of a specific dataset.

3. Finally, we apply our derived amplitude threshold of 1.5 K. Figs. B2(j-l) are as panels (g-i), but regions of AIRS measurements where the absolute 3DST-measured wave amplitude $|\mathcal{A}(x,y,z)| < 1.5$ K are removed. Only measured wave properties from the remaining regions are used in the wave occurrence, momentum flux, wavelength, intermittency and wavevector direction results in Sect. 5. Although some spurious regions do remain, the small spatial extent of these regions means that they do not significantly affect our results.

Our synthetic wave field analysis in Sect. 3.5 and Fig. 3 does not include simulated AIRS retrieval noise, although such noise was added as part of a very similar synthetic test of the 2DST in Hindley et al. (2016) who found that its inclusion did not significantly affect measured values. If we were to add retrieval noise to the synthetic wave field here, we would then simply follow the steps above to remove or exclude it from our results, so we omit such noise from the figure here for visual clarity.

*Acknowledgements.* NPH, CJW and NJM are supported by the UK Natural Environment Research Council (NERC) grant NE/R001391/1,
and TMG is supported by NERC grant NE/R001235/1. CJW is also supported by Royal Society University Research Fellowship UF160545. Reanalysis data are provided by the European Centre for Medium-Range Weather Forecasts (ECMWF). The authors also acknowledge the use of the Balena high-performance computing service at the University of Bath and the JURECA high-performance computing service at Forschungszentrum Jülich. The authors would like to acknowledge J. Perrett for reference comparisons to reanalysis winds, and E. Spence for much-appreciated mathematical assistance. Finally, we would like to thank Markus Rapp and one other anonymous reviewer for their
helpful and detailed reviews of this manuscript.

*Author contributions.* NPH developed and performed the 3-D S-transform analysis, with assistance from NDS. L Hoffmann and MJA developed the 3-D AIRS satellite retrieval and provided key technical support on its use here. NJM, CJW and TMG are responsible for the design and coordination of the NERC-funded DRAGON-WEX project that supported this work. LAH and MJA helped to focus the study with NPH during a residency collaboration. The manuscript and figures were produced by NPH with discussion, contribution and scientific
interpretation from all authors.

*Code and data availability.* The high-resolution AIRS temperature retrieval data can be made available to researchers upon request by contacting Lars Hoffmann at Forschungszentrum Jülich. Likewise, researchers interested in the $N$-dimensional Stockwell transform (NDST) code applied here should contact Neil Hindley at the University of Bath.

*Competing interests.* The authors declare that they have no competing interests.

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
