# Peer review of "Gravity waves in the winter stratosphere over the Southern Ocean: high-resolution satellite observations and 3-D spectral analysis"

_Atmospheric Chemistry and Physics, 2019_

## Referee Comment (RC1) · Anonymous Referee #1 · 15 Jun 2019

In this paper, an analysis of gravity waves in the Southern Hemisphere Winter stratosphere is presented. More precisely, the authors apply a 3-D Stockwell transform (3DST) analysis method to 3D AIRS temperature measurements from Hoffman and Alexander (2009), which enables them to derive different wave characteristics (such as amplitude, wavelength and propagation direction). A key result of the paper is to report a region of horizontal convergence of gravity wave momentum flux near 60° S during southern winter, thus confirming earlier findings. Horizontal propagation is, together with intermittency, one of the main topical questions regarding gravity wave parameterizations in climate models. Quantifying horizontal propagation in observations, as attempted in this paper, is hence highly valuable.

[Figure]

Overall, the paper is well presented and easy to read. However, I think some of the analyses are not sufficiently detailed. This partly comes from the fact that the authors attempt to accomplish two tasks, namely: 1) presenting their method and 2) describing the scientific results they obtain by applying it. Both aspects would be worth deeper investigation, but for this study I would recommend to investigate more thoroughly the performance of the method when applied to the AIRS data. Hence, although the large-scale picture of momentum flux convergence at 60°S is, as said above, valuable to the community, I would like the authors to carefully address my comments and suggestions below before the paper can be considered for final publication.

Main comments

1) The authors offer a thorough evaluation of the performance of their analysis method, and make an effort to address the amplitude reduction problem inherent to the 3DST. However, they disregard the impact of noise and observational filter in AIRS data on their gravity wave retrieval. This could be addressed in a similar manner as done in Sect. 3.5, but by adding typical noise from AIRS observations and applying the observational filter to their synthetic waves. I believe this is necessary for a rigorous description of the capabilities of the method.

2) The authors motivate their study (e.g. in the abstract lines 2 to 5) using the "cold-pole" bias, i.e. the too cold southern polar vortex and delayed vortex breakdown and stratospheric final warming which are simulated in climate models (MacLandress et al., 2012). However, in the stratosphere, the "cold-pole" bias is mainly an early spring-time feature (see, e.g., Fig. 2 of Garcia et al., 2017), while the analysis presented here spans the months of June, July and August 2010. Since the authors believe that daytime measurements are as valuable as nightime ones, I think using the months of September and October would be more appropriate to address their motivations.

3) Although they have access to a very valuable dataset, the authors do not come to any strong conclusion regarding the missing ingredient in parameterization which

should be introduced to solve the "cold pole" problem. This is particularly evident in the discussion (6.1 and 6.2), where the authors remain cautious and mainly point to the literature (e.g. expressions like "could be significant"). In particular, regarding the role of small islands, the authors could reproduce Fig. 11 of Jewtoukoff et al., 2015 (contribution to the zonal mean momentum flux of GW momentum flux above the islands and the ocean). I understand that the limited range of wavelengths accessible within the data hinders any definitive conclusion, but I believe that the authors could speculate using their observations.

Specific comments:

p 3 l 1-2: "unresolved in GCMs due to their size": unresolved and not accounted for in parameterizations

p3 line 10: Can secondary waves alone explain the lack of drag at 60°S, or do they require to be combined with one of the explanations above?

p5 line 1-2: What is this noise level?

p5 line 14: "interpolated": what interpolation is used?

p5 line 22: "a tapering function . . . is applied": Is this smoothing really necessary? How does it affect the retrieved vertical wavelength? (in relation to main comment 1)

p7 line 11-13: Could they also be orographic waves originating from South America or Africa and propagating horizontally?

p7 line 25: Could you briefly explain how the noise analysis of Hoffman and Alexander (2009) works?

p10 Eq. 1: The authors sometimes use absolute frequency (equation 1) and sometimes angular frequency (e.g. appendix A). They should be consistent for better readability. Similarly, they sometimes use $j$ (appendix 1) for the Euler number, sometimes $i$. This is a matter of taste, but I would also recommend using the scalar product notation

instead of matrix product with a transpose.

p 11 line 7: This approach of deriving the analytical signal is not strictly consistent with the definition of the 3DST given in Eq. 1. This does not affect the results, but the target quantity should be more clearly defined. Similarly, I believe there is a phase shift of $2\pi\tau.f = \tau.k$ between Eq. 1 and the method description p11 or Eq. A7 in the appendix. Please check and correct if needed.

p11 line 11: "FFT-1Ha(alpha) * Wn (...)"

p 13 section 3.4.2: It seems to me that you are converging towards the analytical signal for your amplitude retrieval. Why not use the full analytical signal (rather than a filtered version)?

p 16 and Figure 3: How does it change when noise and observational filter effects are included?

P17 l 33: How is it "windowed"? P20 line 30: "it could also result from the reduction in vertical resolution with altitude": an evaluation of the impact of the observational filter would help clarify. See also Main comment 1

p21 : The two case studies help illuminate the potential and limits of the AIRS measurements and the analysis method. Again, this point would be clearer if an analysis of the impact of the observational filter were included.

p 23 l 15 and 17-18: Motivating your observations by the cold-pole problem is not consistent with using June, July, August (see also main comment 3).

p24-25, Sect. 5.1: You mainly compare your results to other AIRS observations, do you know of any reference with other instruments (HIRDLS, SABER)? I understand that the waves observable by different platforms correspond to different parts of the GW spectrum, but this would be of interest to the reader.

p27 lines 11-13: Interestingly, the authors hypothesize a change in the dominant gravity

wave source between June and August (from orographic to non-orographic). Any idea why this would be the case?

p 27 lines 16-18: comparison with Ern et al.: you explain part of the discrepancy with their study (25

p 27-28: How significant are the differences in vertical and horizontal wavelengths?

p28 lines 17-18: "Generally ..." Longer vertical wavelengths also seem to correlate with shorter horizontal ones, correct? I find that the color scale makes it difficult to use the figure quantitatively.

p28 lines 22-28: This could be considered more carefully in Sect. 2 (see main comment 1).

p30 Table 2: It would also be interesting to separate the regions in the Southern Ocean with and without island. Having the mean in each region besides the zonal mean would also be helpful.

p 32 Sect. 5.4: Intermittency: Besides the Gini coefficient and the 90 % quantile, it is common to estimate intermittency using PDFs of momentum fluxes (Plougonven et al., 2012; Jewtoukoff et al., 2015, Holt et al., 2017). It would be interesting to show such pdfs for the different months and regions.

p33: A number of previous studies have suggested that intermittency is generally larger over orography than over the ocean. An exception in this dataset are orographic waves above the Kerguelen, which have lower intermittency and do not not stand out as more intermittent relative to the ocean. Any idea why?

p34 line 12: "could occur in reality" → "occurs for the waves observed in AIRS". I think the authors can be more specific and affirmative in their formulation.

p34 sect. 6.2: The authors mainly report references here. They could be more quantitative regarding the role of small islands by estimating their relative contribution in Table

2.

p36 line 23: "given the size, intensity, spatial extent": Could the authors be quantitative here and refer again to the figure? Why do your observations point to secondary waves and not primary ones? I do not doubt the importance of secondary waves, but I do not see how you can infer that from your observations. If this is pure speculation, you should state it clearly; otherwise, clarify your arguments.

p37 line 2: "under-represent": or under parameterize?

Typos:

p4 line 19: "sensitive to gravity waves"

p14 line 9: "as generalised" → "as general" ?

p18 l19: "explained in section 6" → section 6.3

p28 line 29: "it may be possible" → "we will attempt"

---

## Referee Comment (RC2) · Anonymous Referee #2 · 22 Jul 2019

The authors present a significant extension of the 3D-Stockwell transform (3DST) and then apply the 3DST to Southern hemisphere AIRS observations to determine gravity wave parameters like amplitudes, wavelengths, momentum fluxes and intermittency. With these parameters resolved in 3d, the authors discuss the problem of "missing gravity wave drag" at 60°S as introduced by McLandress et al. (2012).

In all, this is a very comprehensive manuscript that will ultimately make an important contribution to the scientific literature on the role of gravity wave dynamics in Southern hemisphere climate. I recommend publication in ACP once the following mostly minor comments are properly addressed:

[Figure]

1) When introducing satellite observations of gravity waves (e.g., page 3, lines 15-22) the corresponding observational filters and the accessible spatial scales should be mentioned.

2) Please specifically show the AIRS observational filter in Section 2 where the AIRS measurements are introduced. On page 4, line 20 reference is made to Ern et al. (2017) but no numbers are actually mentioned. I consider it critical for this paper to point out what these numbers are. Also statements like "by measuring these waves in AIRS observations we can provide constraints on a large part of the momentum budget" must be quantified! How large is this part? Is this statement based on hard facts? If yes, please present them with suitable references.

3) Page 5, line 8: Don't you know how many observations are made during day- and nighttime? Why is it only likely?

4) Page 7, line 31: Why not explicitly mention the mean/median noise error value? Since all these numbers are available why not compute it exactly?

5) Page 9, Section 3.1: I was initially irritated that the authors introduce this section as an extension of the 2D-Stockwell transform in Hindley et al (2016) and not of their first attempt of a 3D-Stockwell transform as described in Wright et al., ACP 2017. The latter is mentioned at the end of the section, but should be mentioned at the beginning! Why is another extension needed? What is new here? That should be pointed out from the start.

6) Also, I am wondering why the authors bother to develop the 3D Stockwell transform when there is such an overwhelmingly lare body of literature on the application of wavelets to geophysical data sets. Of course, the authors are free in their choice of a suitable method, but a few words on why not using wavelets would be appreciated.

7) Page 15, Figure 3 and related text: it is a very good approach to test the newly developed 3DST on synthetic data. However, I am wondering how realistic the synthetic

data set is. For example, the different wave packets all appear to be well separated in physical space. Is this what we expect to find in the atmosphere? Wouldn't it be more appropriate to do more sensitivity tests to find out how well the 3DST is performing?

8) The assumption of exponential altitude growth with altitude (page 18, line 1 and below) is certainly not generally applicable and that should be stated here. For example the paper by Kruse and Smith (JAS 2016; DOI: 10.1175/JAS-D-16-0173.1) demonstrates a very different behaviour in the vicinity of a wind minimum over New Zealand. It really depends on the local wind conditions and also on the initial wave forcing as found by Kaifler et al. GRL 2015 and Fritts et al., BAMS 2016 whether or not the waves propagate without breaking or not.

9) When discussing the possibility of upward versus downward propagating gravity wave reference could also be made to the paper by Kaifler et al. JASTP 2017 (http://dx.doi.org/10.1016/j.jastp.2017.03.003) which shows strong indications of downward propagating waves in lidar observations.

10) Page 20, line 29/30: Please indicate whether or not the increase of vertical wavelength is consistent with wind profiles from reanalysis data. Thanks!

11) Page 23, line 4/5: is this observed propagation pattern physically plausible?

12) Page 20-22: The authors here hint to a potential bias of their observation sbut never really discuss whether or not that strongly impacts their results. Some more discussion would be appreciated.

13) Page 34, line 10-12: These are very interesting results! Can this analysis be repeated at other altitudes below to see whether the momentum flux can actually be traced to where it is initiated at lower altitudes? That would be extremely valuabe to support the "refraction into the polar vortex"-hypothesis.

14) Page 37, line 24-26: Isn't it just the other way round (see table 2)? Sorry, if I am confused.

---

## Author Comment (AC1) · 13 Sep 2019

**Authors' Response to Reviewer's Comments on of acp-2019-371: "Gravity waves in the winter stratosphere over the Southern Ocean: high-resolution satellite observations and 3-D spectral analysis"**

by N. P. Hindley and co-authors

September 13, 2019

**Overview**

We would like to thank the reviewers for their hard work and useful comments on our submitted manuscript. The paper is substantially improved for as a result of their input. The level of detail and focus on some of the comments is very high, so we thank them again.

**Response to Reviewer #1**

**Main Comment 1**

The authors offer a thorough evaluation of the performance of their analysis method, and make an effort to address the amplitude reduction problem inherent to the 3DST. However, they disregard the impact of noise and observational filter in AIRS data on their gravity wave retrieval. This could be addressed in a similar manner as done in Sect. 3.5, but by adding typical noise from AIRS observations and applying the observational filter to their synthetic waves. I believe this is necessary for a rigorous description of the capabilities of the method.

The reviewer is right to question the effect of noise and observational filter on our results. However we do not think it is fair to say that we "disregard the impact of noise and observational filter in AIRS data on their gravity wave retrieval". We have gone to great length to understand, quantify and mitigate the effects of noise in our data and methods, following approaches that are standard practice in the literature.

To address the reviewer's concerns, we have added a new Appendix to the paper where our step-by-step approach to mitigating the impact of retrieval noise is described and illustrated in more detail, using the AIRS examples in Fig. 1.

We have also performed a sensitivity assessment to produce an additional figure showing the approximate observational filter of our gravity wave retrieval, now included in Fig. 2 in Sect. 2.2. We have expanded the discussion around the impact of the observational filter here. Although our approach is not perfect, its inclusion has greatly improved the paper.

With these improvements, our approach here now goes well beyond the current level of noise and wavelength assessments used in other studies for 3D data and methods, such as for example in the studies of Lehmann et al. (2012), Ern et al. (2017), Wright et al. (2017), Schoon and Zülicke (2018), some of which make no sensitivity assessment at all, so the present study is a significant step forward in that regard.

Regarding the synthetic wave field, this test is not designed to be realistic - we have real AIRS measurements for that - but instead to provide a variety of examples wave packets with different shapes, sizes, wavelengths and orientations such that we can fully assess the performance of the 3DST. If we applied the AIRS observational filter to the synthetic wave field

(not straightforward, a full retrieval on the synthetic field would have to be done which is beyond the scope of the study) we would only prove that our method can measure waves simply within the observational filter of AIRS. The two case studies analysed in Section 4 are sufficient to show this, including the results of previous studies such as in Fig. 4 in Wright et al. (2017), and it also doesn't provide a particularly broad spectral test.

If we added simulated AIRS noise to the synthetic waves, we would simply take steps to remove or exclude it by applying the methods listed in the new Appendix section to exclude noise features from our analysis, so its inclusion is not useful. These steps, particularly the spectral exclusion of noisy features, enables our 3DST application be quite resilient to retrieval noise, as shown in a new figure in the Appendix. Further, a near-identical test of the 2DST on a synthetic wave field was performed in Hindley et al. (2016), where it was found that the addition of AIRS-like speckle noise had no significant effect on measured properties, so we do not include it here for visual clarity. We have updated the text to reflect these points.

**Main Comment 2**

The authors motivate their study (e.g. in the abstract lines 2 to 5) using the "cold- pole" bias, i.e. the too cold southern polar vortex and delayed vortex breakdown and stratospheric final warming which are simulated in climate models (MacLandress et al., 2012). However, in the stratosphere, the "cold-pole" bias is mainly an early spring- time feature (see, e.g., Fig. 2 of Garcia et al., 2017), while the analysis presented here spans the months of June, July and August 2010. Since the authors believe that daytime measurements are as valuable as nighttime ones, I think using the months of September and October would be more appropriate to address their motivations.

Very good point. The description of the cold-pole problem was intended only to give focus and direction to the goals of the study - that 3D observations of gravity waves in the southern hemisphere are needed for flux estimates and are sorely lacking. We have adjusted the text in the abstract and introduction to reflect this.

It is still useful however to investigate on gravity-wave activity in the southern hemisphere during June-August. Figure R1 shows gravity-wave occurrence frequency for each month (the same as in Fig. 6 of the paper) for 11 years of AIRS data from 2008-2018, analysed using the 3DST in the same way as described in the paper. We can see that the highest wave activity in the region is observed during June-August, with significant but lower levels of activity in May and September. This figure is part of future work that is going towards a multi-year climatology study currently in preparation. The consideration of gravity waves in September and October will take place in this future study.

**Main Comment 3**

Although they have access to a very valuable dataset, the authors do not come to any strong conclusion regarding the missing ingredient in parameterization which should be introduced to solve the "cold pole" problem. This is particularly evident in the discussion (6.1 and 6.2), where the authors remain cautious and mainly point to the literature (e.g. expressions like "could be significant"). In particular, regarding the role of small islands, the authors could reproduce Fig. 11 of Jewtoukoff et al., 2015 (contribution to the zonal mean momentum flux of GW momentum flux above the islands and the ocean). I understand that the limited range of wavelengths accessible within the data hinders any definitive conclusion, but I believe that the authors could speculate using their observations.

This is an excellent suggestion regarding reproducing the small island analysis of Jewtoukoff et al. (2015), and one that is coming in a following study. We plan to produce something like Table 1 of Hoffmann et al. (2016), only with fractions of GWMF.

However to produce realistic estimates of the fractions of flux from small islands a non-trivial approach is needed. We must account for some islands having increased "background" fluxes simply by lying within the belt of enhanced GW activity and others not. For this reason we cannot simply take a box over the islands like we did in Table 2 or follow the approach of Jewtoukoff et al. (2015). As evidence of this, in the example in Fig. 1c we can see a huge wave over Kerguelen that is

almost certainly not an island mountain wave like in Fig. 1b. If we took a box over Kerguelen, we would just capture parts of waves like this and incorrectly assign their fluxes to small island sources.

A more sophisticated approach involving accurate background assessments to constrain the fraction of flux from small islands is thus beyond the scope of this study and is planned for a future study.

**Responses to Specific Comments**

- 1. p 3 l 1-2: "unresolved in GCMs due to their size": unresolved and not accounted for in parameterizations Fixed, thanks.
- 2. p3 line 10: Can secondary waves alone explain the lack of drag at 60S, or do they require to be combined with one of the explanations above?

The current literature on secondary gravity waves is in its infancy, so it is not possible to say yet. Understanding the link (if there is one) between intense mountain wave breaking over the southern Andes and the belt of enhanced GW activity at 60S is a key topic of research at the moment. It could be that the intense disturbances resulting from mountain wave breaking could trigger a cascade of secondary waves in a range of directions at various atmospheric layers. Some of these secondary wave packets, or even primary mountain wave packets, with low intrinsic phase speeds could find themselves taken far to the east by the strong eastward winds of the vortex before breaking and dissipating. As we said though, it is very difficult to comment on how likely this mechanism is at this stage so currently we only have speculation.

3. p5 line 1-2: What is this noise level?

We were referring to daytime noise levels that can be seen by the dashed lines in Fig. 2a for different latitudes and seasons. We've added a reference to the figure to help.

4. p5 line 14: "interpolated": what interpolation is used?

Standard linear interpolation of the values at the nearest grid points in each dimension. 'Regridded' would have been a better word, we've fixed this now.

5. p5 line 22: "a tapering function . . . is applied": Is this smoothing really necessary? How does it affect the retrieved vertical wavelength? (in relation to main comment 1)

We apologise, our phrasing was confusing. We do not apply a smoothing filter, we simply multiply each vertical column in the 3D AIRS measurement volume with a Tukey window that is equal to 1 between around z = 20 - 60 km and zero elsewhere, with a smooth half-bell taper from zero to one and vice versa. This removes spurious high-noise perturbations above and below the usable window (see Fig. 2a), and effectively "localises" the perturbations into our usable window. The effect on measured wavelengths is that is reduces our ability to measure vertical wavelengths longer than around 40 km (see the new Fig. 2c). We've updated the text to make this clearer.

6. p7 line 11-13: Could they also be orographic waves originating from South America or Africa and propagating horizontally?

Possibly, but here we don't think so - we have run animations looking at whole months of AIRS ascending and descending node scans globally, and we don't see clear evidence of intense mountain wave activity forming over South Africa that could extend out or become detached and float out over the oceans, but it is not impossible.

7. p7 line 25: Could you briefly explain how the noise analysis of Hoffmann and Alexander (2009) works?

Yes, no problem. We've included a short description in the paper. The noise analysis used to estimate the AIRS noise values in Fig 2a follows standard optimal estimation retrieval theory (Rodgers, 2000). The measurement covariance

matrix characterising the noise of the AIRS radiance observations is mapped into temperature errors by means of the gain matrix calculated in the retrieval process. Noise estimate for the individual AIRS channels used in the error analysis have been taken from version 6 of the AIRS channel property files provided by the AIRS instrument team at NASA.

8. p10 Eq. 1: The authors sometimes use absolute frequency (equation 1) and sometimes angular frequency (e.g. appendix A). They should be consistent for better read- ability. Similarly, they sometimes use j (appendix 1) for the Euler number, sometimes i. This is a matter of taste, but I would also recommend using the scalar product notation instead of matrix product with a transpose.

Ok, good suggestions. We've fixed the use of j in the appendix, and replaced the use of the transpose  $k^{\top}x$  with the scalar product  $k \cdot x$  throughout.

We used spatial frequency in the main body of the paper because it felt a little clearer, and is closer to the original S-transform definition in Stockwell et al. (1996). Further, since we use FFT algorithms, the output frequencies come out as spatial frequencies. We used angular frequencies (and stated so) in the appendix because it was cleaner than lugging factors of  $2\pi$  around through the derivation. We'd prefer to keep it this way, and since the main body and the appendix are quite separate we don't think it affects the readability too much.

9. p 11 line 7: This approach of deriving the analytical signal is not strictly consistent with the definition of the 3DST given in Eq. 1. This does not affect the results, but the target quantity should be more clearly defined. Similarly, I believe there is a phase shift of  $2\pi\tau \cdot k = \tau \cdot k$  between Eq. 1 and the method description p11 or Eq. A7 in the appendix. Please check and correct if needed.

The reviewer is absolutely right here, and it's great that they noticed this.

After some investigation, it seems this discrepancy arises because the code that we have developed to compute the NDST uses the same (general) method that can be found in Bob Stockwell's old matlab code for the 1-D S-transform, which at time of writing can still be found here: https://uk.mathworks.com/matlabcentral/fileexchange/51808-time-frequency-distribution-of-a-signal-using-s-transform-stockwell-transform. This method, and our own application, follows the general steps that we listed on page 11. Interestingly, when this code is computed however, the S-transform that comes out has a phase shift when compared to Stockwell's original definition in Stockwell et al. (1996) shown in Eqn. 1, as the reviewer rightly points out. It does not affect the results here, since we don't use the phase explicitly, but it could be important to consider in future applications. We have included the phase shift this in the text.

We wanted to be consistent with the original definition of the S-transform in Eqn. 1 to give readers something they might recognise before we extend it to N dimensions.

10. *p11 line 11: "FFT-1Ha(alpha) \* Wn (...)"*

Corrected, thanks.

11. p 13 section 3.4.2: It seems to me that you are converging towards the analytical signal for your amplitude retrieval. Why not use the full analytical signal (rather than a filtered version)?

Yes, we are using a filtered version of the analytic signal to make an estimate of wave amplitude at each location. The reason we filter it is so that the dominant measured amplitudes relate to the measured frequencies that chosen in the S-transform and not unwanted frequencies, such as those that correspond to noise.

12. p 16 and Figure 3: How does it change when noise and observational filter effects are included?

See response to Main Comment 1.

The synthetic wave field is not designed to be realistic - we have real AIRS data for that - but we found that small-scale speckle noise like the kind we see in AIRS measurement does not affect the measured values significantly, so we did

not include it. If it did, we would simply take steps like those shown in the new Appendix to reduce their impact, so it is not useful to include it here. A near-identical analysis was done in Hindley et al. (2016), who found that simulated noise did not affect measured values significantly.

Applying the observational filter would make the test less thorough, since we would only be demonstrating that we can measure waves within the observational filter of AIRS, rather assessing the performance of the 3DST on the more generalised variety of wave packets shown in Fig. 3, including some cases where it struggles.

13. P17 I 33: How is it "windowed"? P20 line 30: "it could also result from the reduction in vertical resolution with altitude": an evaluation of the impact of the observational filter would help clarify. See also Main comment 1

See response to comment above on the vertical localising window.

We have now included a sensitivity assessment for the observational filter of the gravity-wave retrieval (see response Main Comment 1), which shows that the effect of this limited vertical window is to reduce sensitivity to vertical wavelengths longer than around 40 km, as shown in the new Fig. 2c. This has now been clarified in the text.

14. p21: The two case studies help illuminate the potential and limits of the AIRS measurements and the analysis method. Again, this point would be clearer if an analysis of the impact of the observational filter were included.

See response to Main Comment 1. An estimated observational filter diagram has been included in Fig. 2c. The impact of the observational filter is that we are preferentially sensitive to waves with wavelengths within this filter.

15. p23 l15 and 17-18: Motivating your observations by the cold-pole problem is not consistent with using June, July, August (see also main comment 3).

See response to Main Comment 3.

16. p24-25, Sect. 5.1: You mainly compare your results to other AIRS observations, do you know of any reference with other instruments (HIRDLS, SABER)? I understand that the waves observable by different platforms correspond to different parts of the GW spectrum, but this would be of interest to the reader.

Thanks, we've added a discussion of and reference to Ern et al. (2018) in Sect. 5.3 which shows GWMF results from HIRDLS and SABER very nicely.

17. p27 lines 11-13: Interestingly, the authors hypothesize a change in the dominant gravity wave source between June and August (from orographic to non-orographic). Any idea why this would be the case?

Our main consideration for this was the non-uniformity in the zonal distribution of wave activity around 60S during June. Here, and in several previous studies (e.g. Hendricks et al., 2014; Hindley et al., 2015), wave activity is highest over the southern Andes and Antarctic Peninsula, then gradually decreases eastward around Antarctica to reach a minimum just before the southern Andes again. This distribution could be an indication of a connection between wave activity over the southern Andes and the belt of increased wave activity - but of course it is difficult to this investigate further, as a credible physical mechanism is needed (i.e. secondary waves or similar), which would likely require dedicated modelling or ray-tracing studies to investigate. Some previous observational studies did not consider this aspect Hendricks et al. (e.g. 2014), despite the observed distribution, so we mention it here for discussion.

During August however, the wave activity becomes much more zonally uniform. This could suggest the cessation of any connection between Andean mountain waves and the observed belt of GW activity, such that non-orographic sources from jets, fronts, storms, instabilities etc. become more dominant. Without further evidence however, this is just speculation, so we don't go further into detail in the paper.

We have clarified our language in the text to reflect this.

18. p 27 lines 16-18: comparison with Ern et al.: you explain part of the discrepancy with their study (25

Sorry, the comment seems to be incomplete. If it helps, we are in regular contact with Manfred Ern and the group at Forschungszentrum Jülich about the various strengths and weaknesses and that arise from the use of two different spectral analysis methods on 3D AIRS data. A comprehensive comparison study between the 3DST and the S3D method is currently underway.

19. p 27-28: How significant are the differences in vertical and horizontal wavelengths?

Plotting monthly means of properties like wavelengths tends to lead to quite generalised maps, due to the wide variation in measured values.

Our 3DST analysis tends to measure very long horizontal wavelengths ( $\lambda_H >> 1000$  km, i.e. a flat horizontal field) in regions of very little wave activity (see white regions in Figs 7(d,e,f)). Even once the noise threshold is applied, some small patches of these regions persist and aggregate in the monthly means, which is what we are seeing in the white regions. We have update the text to describe this better.

In the coloured regions, the nature of the wide variety of measured wavelengths means that monthly averages tend to a similar mean value for most regions - perhaps a median would have been better. For example, in the example in Fig. 1c, horizontal wavelengths of small mountain waves over the island of Kerguelen can easily be high biased in a monthly-mean by averaging with larger-scale waves like the one in the example. Despite this, general variation in measured wavelengths on a monthly scale is still observable.

We have thoroughly rewritten parts of Sect. 5.2 to make the reasons for the observed distribution of wavelengths clearer.

20. p28 lines 17-18: "Generally ..." Longer vertical wavelengths also seem to correlate with shorter horizontal ones, correct? I find that the color scale makes it difficult to use the figure quantitatively.

We apologise, that sentence was misleading and has been removed.

- 21. p28 lines 22-28: This could be considered more carefully in Sect. 2 (see main comment 1). See response to main comment 1.
- 22. p30 Table 2: It would also be interesting to separate the regions in the Southern Ocean with and without island. Having the mean in each region besides the zonal mean would also be helpful.

See response to Main Comment 3.

We thought about adding a value for the mean flux in each region in Table 2 but it might be confusing arithmetically. The value over the whole southern Andes region (around -4.55 mPa for June), plus the mean value over the Southern Ocean region (around -0.56 mPa for June) does not equal the mean value over the whole latitude band (-0.96 mPa). These values are also counter-intuitively quite different from the fractions of total flux due to the much larger area of the Southern Ocean (around 70% of total flux) and the increased number of rare, large amplitude events over the southern Andes (see PDFs now included in Fig. 9).

This is why we chose to show fractional values of the *total* zonal momentum flux in the latitude band, which is much simpler conceptually and we argue more useful. A large and thorough breakdown of the mean fluxes over these regions from AIRS measurements, and over small islands, is coming in a future study.

23. p 32 Sect. 5.4: Intermittency: Besides the Gini coefficient and the 90% quantile, it is common to estimate intermittency using PDFs of momentum fluxes (Plougonven et al., 2012; Jewtoukoff et al., 2015, Holt et al., 2017). It would be interesting to show such pdfs for the different months and regions. This is a great suggestion, and we have included the PDFs for each month in Fig. 9 using the regions as defined in Table 2. As the reviewer may have suspected, there are significant differences between the orographic and non-orographic regions during each month.

We have added a short discussion of this in the manuscript.

24. p33: A number of previous studies have suggested that intermittency is generally larger over orography than over the ocean. An exception in this dataset are orographic waves above the Kerguelen, which have lower intermittency and do not not stand out as more intermittent relative to the ocean. Any idea why?

Yes, we agree. We noticed this too but we weren't sure until our latest multi-year analysis was done. In Fig. R1 we can see that Kerguelen is located under a quite striking spot of increased wave occurrence during June, July and August. This suggests that waves are observed over the island very often in AIRS measurements, yielding a low value for intermittency.

We have expanded our discussion of this in the paper.

- 25. p34 line 12: "could occur in reality"  $\rightarrow$  "occurs for the waves observed in AIRS". I think the authors can be more specific and affirmative in their formulation.
  - Corrected, thanks.
- 26. p34 sect. 6.2: The authors mainly report references here. They could be more quantitative regarding the role of small islands by estimating their relative contribution in Table 2.

See response to Main Comment 3 and earlier comment above. This is non-trivial and planned in a subsequent study.

27. p36 line 23: "given the size, intensity, spatial extent": Could the authors be quantitative here and refer again to the figure? Why do your observations point to secondary waves and not primary ones? I do not doubt the importance of secondary waves, but I do not see how you can infer that from your observations. If this is pure speculation, you should state it clearly; otherwise, clarify your arguments.

We have updated the paragraph to reflect the reviewer's recommendation. Our logic was that since mountain waves generally form directly over the mountains and not far downwind, what was the reason for the 'spreading' of enhanced GW activity that extended out eastwards immediately downwind until about 15°W? Could they be mountain wave packets advected downwind via the mechanism of Sato et al. (2012)?. Alternatively, Preusse et al. (2014) found that resolved waves in this region had lowest traceable altitudes (LTAs) that were quite high (around 15 km), which would suggest however that they were non-orographic. However, the reviewer is right I suppose it was speculation, and we have stated so now, but it is a growing suspicion we have that secondary waves could be quite a significant fraction of missing waves in models, possibly due to breaking processes being damped/not fully simulated.

28. p37 line 2: "under-represent": or under parameterize?

Fixed, thanks.

**Typos**

All fixed, thanks.

**Response to Reviewer #2**

**Response to Specific Comments**

1. When introducing satellite observations of gravity waves (e.g., page 3, lines 15- 22) the corresponding observational filters and the accessible spatial scales should be mentioned.

Thanks, we have added this.

2. Please specifically show the AIRS observational filter in Section 2 where the AIRS measurements are introduced. On page 4, line 20 reference is made to Ern et al. (2017) but no numbers are actually mentioned. I consider it critical for this paper to point out what these numbers are. Also statements like "by measuring these waves in AIRS observations we can provide constraints on a large part of the momentum budget" must be quantified! How large is this part? Is this statement based on hard facts? If yes, please present them with suitable references.

We have added some values into the text of the approximate observational filter of the 3D AIRS measurements and rephrased the text to make this clearer at the point the reviewer suggested. We have also performed a sensitivity assessment and added a new observational filter diagram for the 3D AIRS measurements in Fig. 2c, which provides more information on the values involved. See response to Main Comment 1 from Reviewer #1.

We have removed the sentence about the fraction of the total GWMF budget.

- Page 5, line 8: Don't you know how many observations are made during day- and nighttime? Why is it only likely? Poleward of 30°S during June-August, around 70-75% of AIRS observations occurred under nighttime conditions. We have added these numbers in the text.
- 4. Page 9, Section 3.1: I was initially irritated that the authors introduce this section as an extension of the 2D-Stockwell transform in Hindley et al (2016) and not of their first attempt of a 3D-Stockwell transform as described in Wright et al., ACP 2017. The latter is mentioned at the end of the section, but should be mentioned at the beginning! Why is another extension needed? What is new here? That should be pointed out from the start.

We apologise if it was confusing, we have now revised the text to make the timeline of code developments and studies clearer. Both this study *and* the study of Wright et al. (2017) are extensions of the 2DST of Hindley et al. (2016), so the statement was correct, but it has been removed now anyway.

Two things are new here. Firstly, this is the first "extended" (we meant geographically speaking) application of the 3DST to 3D AIRS data over the whole Southern Ocean, rather than the regional study over the southern Andes in Wright et al. (2017). Secondly, the S-transform code has been completely rewritten here since the study of Wright et al. (2017) using the improved methods and techniques that are described in Sect. 3, and it has also been redesigned to handle 1-, 2-, 3- and 4-D datasets in a generalised manner. It's an entirely new package, and it is tested and validated here in a way that the previous version in Wright et al. (2017) was not.

We have adjusted the text to make this clearer.

5. Also, I am wondering why the authors bother to develop the 3D Stockwell trans- form when there is such an overwhelmingly large body of literature on the application of wavelets to geophysical data sets. Of course, the authors are free in their choice of a suitable method, but a few words on why not using wavelets would be appreciated.

A good question, and wavelets are indeed a legitimate choice for this purpose. There are two reasons why we use the S-transform here, and we've included this in the text.

Firstly, Continuous Wavelet Transforms (CWTs) are generally quite computationally intensive to run, and higherdimensional CWTs become prohibitively expensive using current computing technology. The S-transform can make use of fast (discrete) Fourier transform (DFT) algorithms, so higher-dimensional forms are relatively fast to compute. This makes is more practical for large-scale, multi-dimensional data analysis in the geosciences.

Secondly, CWTs do not *strictly* provide a measure of wave amplitude, but of a "magnitude" of the transform at a given time and scale. It is possible to derive an approximation of associated spectral power from these magnitude coefficients of the CWT from which one could make an estimate of wave amplitude, but since the spectrum is oversampled such estimate might not always relate exactly to the input signal physically. Also, for some wavelets even the conversion from scale to frequency is not especially accurate. The S-transform uses Fourier algorithms, the coefficients of which are directly related to wave amplitudes and are easier to work with.

The S-transform is actually very closely related to the wavelet transform, so much so that seems to be some discussion in the literature about whether it is or is not a wavelet transform after all (Stockwell, 2007; Gibson et al., 2006). It seems to depend on how one chooses to define an admissible wavelet for a wavelet transform.

I'm a big fan of wavelets (Hindley et al., 2015, Sect. 3), and would very much like to make use of the continuous sampling features of them, but I am yet to find a computationally-inexpensive higher-dimensional implementation that can yield reliable wave amplitude measurements faster than the S-transform code we use here.

6. Page 15, Figure 3 and related text: it is a very good approach to test the newly developed 3DST on synthetic data. However, I am wondering how realistic the synthetic data set is. For example, the different wave packets all appear to be well separated in physical space. Is this what we expect to find in the atmosphere? Wouldn't it be more appropriate to do more sensitivity tests to find out how well the 3DST is performing?

See response to Main Comment 1 from Reviewer #1.

The synthetic wave field is not designed to be exactly realistic - we have real data for that - it is designed to provide a wider variety of wave packets so that we can assess how our performs in all circumstances, including some where it struggles. In that way, it does act as a sensitivity test.

The synthetic wave packets do all overlap with each other, we have just chosen to draw the isosurfaces at one FWHM of each packet's envelope, so they appear well-separated but actually continue out to interfere with each other. As it turns out, in most AIRS observations that we have seen over the southern hemisphere during winter, most clear wave events really do seem to appear as big distinct wave packets like those seen in Fig. 1, so the synthetic wave field is actually a bit more brutal than we observed in most AIRS measurements.

Finally (see also response to Reviewer #1), we have now included a sensitivity assessment for our AIRS gravity-wave retrieval in Fig. 2c.

7. The assumption of exponential altitude growth with altitude (page 18, line 1 and below) is certainly not generally applicable and that should be stated here. For example the paper by Kruse and Smith (JAS 2016; DOI: 10.1175/JAS-D-16-0173.1) demonstrates a very different behaviour in the vicinity of a wind minimum over New Zealand. It really depends on the local wind conditions and also on the initial wave forcing as found by Kaifler et al. GRL 2015 and Fritts et al., BAMS 2016 whether or not the waves propagate without breaking or not.

Okay, thanks we have included this.

The use of the factor of  $\kappa(z) = e^{-\frac{z-40}{2H}}$  on wave amplitudes was only designed reduce the chance of large wave amplitudes at higher altitude dominating the spectrum of the whole measurement volume. This factor is immediately removed anyway from the 3DST-measured amplitudes afterwards, so it does not propagate any further into our results in any case. We have now mentioned that it should not be assumed in the general case.

Our assumption was based upon Fig. 10a of Wright et al. (2015), where exponential growth was seen quite clearly in the mid-stratosphere for HIRDLS measurements of GWs over the southern Andes during winter, although this was for quite a large average of data!

 When discussing the possibility of upward versus downward propagating gravity wave reference could also be made to the paper by Kaifler et al. JASTP 2017 (http://dx.doi.org/10.1016/j.jastp.2017.03.003) which shows strong indications of down- ward propagating waves in lidar observations.

Thanks, included.

9. Page 20, line 29/30: Please indicate whether or not the increase of vertical wavelength is consistent with wind profiles from reanalysis data. Thanks!

We have analysed zonal wind speeds over the region from ERA5 reanalyses for each day June 2010, including those on which the case studies are shown. We found that wind speeds gradually increased with height from the surface to around 50 km, where speeds reached around 70 m/s. Above 50 km, wind speeds remained constant for around 5 km before decreasing with altitude to around 30 m/s at 80 km at the model top.

Vertical wavelengths for quasi-stationary mountain waves would thus be expected to increase smoothly as  $\lambda_Z \approx \frac{2\pi U}{N}$  where  $\overline{U}$  is the mean wind speed and N is the Brunt-Väisälä frequency (Eckermann and Preusse, 1999).

Above 50 km however, no further increase in  $\lambda_Z$  is expected as the wind has stopped increasing.

We have added this to the paper.

10. Page 23, line 4/5: is this observed propagation pattern physically plausible?

Yes, this is a characteristic mountain wave pattern from an isolated source. Wave fronts are aligned in a "bow-wave" shape (e.g. Alexander et al., 2009). Northward (southward) parts are found to be directed northward (southward) but they also propagating into the wind, so they appear stationary in this shape.

We have revised the text to make this clearer.

11. Page 20-22: The authors here hint to a potential bias of their observation sbut never really discuss whether or not that strongly impacts their results. Some more discussion would be appreciated.

Apologies, the phrasing was poor. It's not a bias in the method but reduced precision for relatively long vertical wavelengths as a result of using DFT algorithms over a relatively small vertical window - as is the case for any study that uses such spectral techniques like this. This doesn't lead to a low-bias for wavelength measurements in our method but there is some reduced precision. This was discussed in Sect. 3.5.

We checked that our measurements were valid by repeating the analysis but also zero-padding the data at either end in the vertical. We found that the same vertical wavelength numbers came out for different regions, albeit at higher precision between the different regions. Extensive testing has gone into this over the years with this and we are confident that this does not significantly affect our results.

We have revised the text to remove the confusion.

12. Page 34, line 10-12: These are very interesting results! Can this analysis be repeated at other altitudes below to see whether the momentum flux can actually be traced to where it is initiated at lower altitudes? That would be extremely valuabe to support the "refraction into the polar vortex"-hypothesis.

This is an excellent suggestion, and we currently have a student using ray-tracing of these results to address exactly this question! This should lead a to publication soon.

13. Page 37, line 24-26: Isn't it just the other way round (see table 2)? Sorry, if I am confused. The reviewer is right, the wording here is confusing. We have revised the text to make it clearer.

**References**

[revised manuscript text omitted]